

# Composing topological domain walls and anyon mobility

Peter Huston[1], Fiona Burnell[2], Corey Jones[3] and David Penneys[4]

**1** Department of Mathematics, Vanderbilt University, Nashville, TN 37212, USA
**2** School of Physics and Astronomy, University of Minnesota, Minneapolis, MN 55455, USA
**3** Department of Mathematics, North Carolina State University,
Raleigh, North Carolina 27695, USA
**4** Department of Mathematics, The Ohio State University, Columbus, OH 43210, USA

## Abstract

Topological domain walls separating 2+1 dimensional topologically ordered phases can be understood in terms of Witt equivalences between the UMTCs describing anyons in the bulk topological orders. However, this picture does not provide a framework for decomposing stacks of multiple domain walls into superselection sectors — i.e., into fundamental domain wall types that cannot be mixed by any local operators. Such a decomposition can be understood using an alternate framework in the case that the topological order is anomaly-free, in the sense that it can be realized by a commuting projector lattice model. By placing these Witt equivalences in the context of a 3-category of potentially anomalous (2+1)D topological orders, we develop a framework for computing the decomposition of parallel topological domain walls into indecomposable superselection sectors, extending the previous understanding to topological orders with non-trivial anomaly. We characterize the superselection sectors in terms of domain wall particle mobility, which we formalize in terms of tunnelling operators. The mathematical model for the 3-category of topological orders is the 3-category of fusion categories enriched over a fixed unitary modular tensor category.

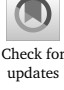

# 1 Introduction

The study of defects in topologically ordered phases of matter has many important physical applications, from engineering non-abelian anyons for quantum information applications [23, 30, 33, 100, 113] to classification of phases [1, 52, 121]. In 2+1 dimensions, an important class of defects are *topological domain walls* separating two topologically ordered regions [59, 79, 86, 97, 105].

The classification of such domain walls is well understood from several different perspectives. A topological domain wall separating topological orders described by the *unitary modular tensor categories* (UMTCs) $\mathcal{C}$ and $\mathcal{D}$ is defined by a *Witt equivalence* between $\mathcal{C}$ and $\mathcal{D}$, which describes the point defects that can be localized to the domain wall in a manner that is consistent with the fusion and braiding rules of the anyons that can be brought to the wall from either of the adjacent bulk regions [79]. Choosing such a Witt equivalence is equivalent to

| 2D bulk | UMTC |
| 1D topological domain wall | Witt equivalence |

Figure 1: Standard description of topological order in terms of localized excitations, cf. [79,85]; contrast with Figures 2 and 3 below. A Witt equivalence [42] $\mathcal{C} \to \mathcal{D}$ is a unitary multifusion category $\mathcal{X}$ with a choice of braided equivalence $Z(\mathcal{X}) \cong \mathcal{C} \boxtimes \overline{\mathcal{D}}$; see § 2.4. When we write $n$D above, we mean the spatial dimension. We also use this abbreviation in Figures 2 and 3.

specifying a Lagrangian algebra in $\mathcal{C} \boxtimes \overline{\mathcal{D}}$,[1] i.e. a gapped boundary between $\mathcal{C} \boxtimes \overline{\mathcal{D}}$ and the vacuum [85,97]. Another useful perspective is to study *particle mobility* across domain walls. This has been explored extensively for *invertible* domain walls $\mathcal{M}$ [2,7,13,17,22,23,30,33,95,126], in which a quasiparticle with topological charge $a$ entering the domain wall exits on the other side with topological charge $\Phi_{\mathcal{M}}(a)$, where $\Phi_{\mathcal{M}}$ is a braided equivalence between the UMTCs on either side.

However, the characterizations described above leave unaddressed an important question: what happens when we "compose" parallel domain walls by horizontally stacking them? This has been studied extensively in the case of non-chiral theories [6, 9, 10], where it has been shown that a composite of two parallel indecomposable domain walls can decompose as the direct sum of multiple superselection sectors, which need not be equivalent to one another. In particular, particle mobility need not be the same in different superselection sectors. In this setting, extensive use is made of the higher 3-categorical structure of fusion categories, which classify topological defects according to the cobordism hypothesis [12, 101].

In this paper, we develop the tools necessary to extend the study of composite domain walls to the chiral setting. We describe *tunneling operators*, which bring anyons from one side of a domain wall to the other, and explain how the structure of the *space of tunneling operators* gives a natural description of a general domain wall from the perspective of anyon mobility. We show how composites of certain tunneling operators across a composite domain wall can be used to determine the decomposition of the composite wall into distinct superselection sectors. We then describe how to identify the indecomposable domain walls in each superselection sector by computing sets of tunneling operators for the various anyon types, and we carry out the computations in several examples. Parallel results in the context of conformal field theory have been previously obtained in [58].

While this work contains some results that are primarily of mathematical interest, which we phrase in the language of higher category theory, the statements of the main results do not require this technology, and are interesting and accessible to physicists with a background in topological order. In particular, we separate out some remarks which provide context for those interested in higher categories of topological orders but which could be safely skipped over via the label "Remark (Mathematical)."

In order to study these questions, we adopt a new mathematical perspective on (2+1)D topological orders. The standard characterization of a (2+1)D topological order is by its UMTC of localized excitations, resulting in the correspondence between defects and mathematical data in Figure 1.

A natural attempt to put this characterization in mathematical terms would be to describe (2+1)D topological order using the Morita 4-category UBFC of unitary braided fusion categories.[2] In this 4-category, 0-morphisms are unitary braided fusion categories, which correspond to bulk topological orders, and 1-morphisms are bimodule multifusion categories, which

---

[1] $\overline{\mathcal{D}}$ is the UMTC with the reverse braiding of $\mathcal{D}$.

[2] To be more precise, we take the 1-truncation of the 4-subcategory UMTC of UBFC whose objects are UMTCs and whose higher morphisms are all invertible.

| 2D bulk | UFC |
| --- | --- |
| 1D domain wall | bimodule category |
| 0D point defect | bimodule functor |
| local operators[4] | bimodule natural transformations |

Figure 2: Description of anomaly free topological order in terms of ingredients for commuting projector model, cf. [79, 93]; compare with Figure 3 below.

correspond to codimension 1 topological defects, i.e. domain walls. However, 2-morphisms are bimodule categories (with compatible actions of the relevant braided fusion categories on each side), which do not correspond to codimension 2 topological defects. In particular, the anyons themselves do not appear as 2-morphisms. Moreover, 3-morphisms, which are bimodule functors, do not form a vector space, so linear algebraic data such as $F$-symbols do not appear at this categorical level. (We refer the reader to Remark 2.4 for a more detailed discussion of UBFC.)

Consequently, details such as how one can concatenate tunneling operators across parallel domain walls or use local operators to distinguish superselection sectors of the composite of two walls cannot be explained naturally from this perspective (e.g. via the graphical calculus of UBFC). In particular, since the composite of two Witt equivalences is again a Witt equivalence, the decomposition of a composite domain wall into superselection sectors is *not* a direct sum decomposition of 1-morphisms in UBFC.

These difficulties illustrate the necessity of placing the tensor categories of excitations listed in Figure 1 into the context of a 3-category of (2+1)D topological orders. In the anomaly-free[3] setting, this context is well-understood: a unitary fusion category (UFC) $\mathcal{X}$ can be used to construct a Levin-Wen string-net model [103] with (2+1)D topological order, where the localized excitations are given by $\mathrm{End}_{\mathcal{X}-\mathcal{X}}(\mathcal{X}) \cong Z(\mathcal{X})$ [76, 80, 103]. Moreover, unitary fusion categories form a 3-category UFC describing all levels of anomaly-free (2+1)D topological order, as summarized in Figure 2.

From this perspective, it is clear how to decompose parallel domain walls into superselection sectors by decomposing the relative tensor product of bimodule categories into indecomposable summands [9, 10]. Moreover, since 0D point defects between a domain wall and itself are wall excitations, and 0D point defects between the trivial domain wall and itself are localized excitations in the 2D bulk, this perspective naturally produces the tensor categories of localized excitations in Figure 1 as endomorphisms of 1-morphisms [79].

We adapt these techniques to the anomalous setting by introducing a new perspective on (2+1)D topological order afforded by *enriched fusion categories*. (2+1)D topologically ordered phases typically carry an anomaly described by an invertible (3+1)D topological quantum field theory [71]. Such anomalies correspond to Witt classes of UMTCs [15].

To describe a given topological order, we therefore first choose a representative UMTC $\mathcal{A}$ of the Witt class of the anomaly. An $\mathcal{A}$-enriched UFC is a UFC $\mathcal{X}$ equipped with a (fully faithful) unitary braided tensor functor $F : \mathcal{A} \to Z(\mathcal{X})$,[5] the Drinfeld center of $\mathcal{X}$.[6] In fact, $\mathcal{A}$-enriched

---

[3]Here, the anomaly refers to an obstruction to being realizable with a commuting projector local Hamiltonian, or equivalently, an obstruction to the low energy effective topological quantum field theory (TQFT) being fully extended. We refer the reader to § 2.1 for a further discussion of the anomaly.

[4]In this paper, by a local operator in a topologically ordered system, we mean a topological local operator, which in general is only a quasilocal operator (i.e., can be approximated by local operators) which corresponds to an intertwining operator between superselection sectors of the low energy effective quantum field theory describing the emergent topological order. In the commuting projector lattice models we will describe, such operators will actually be local.

[5]Such pairs $(\mathcal{X}, F)$ were called *module tensor categories* for $\mathcal{A}$ in [67, 90], and the later articles [74, 92, 93, 108, 109] motivate the name $\mathcal{A}$-enriched fusion category.

[6]The Drinfeld center $Z(\mathcal{X})$ of a UFC $\mathcal{X}$ is constructed by looking at objects equipped with half-braidings. See

| 2D bulk | $\mathcal{A}$-enriched fusion category |
|---|---|
| 1D domain wall | $\mathcal{A}$-enriched bimodule category |
| 0D point defect | $\mathcal{A}$-centered bimodule functor |
| local operators | bimodule natural transformations |

Figure 3: Description of topological order in terms of ingredients for commuting projector model afforded by $\mathcal{A}$-enriched fusion categories.

fusion categories form a linear 3-category denoted $\mathrm{UFC}^{\mathcal{A}}$.

As for the string net models previously described, from this perspective, the UMTC of bulk excitations in Figure 1 arises as $\mathrm{End}^{\mathcal{A}}_{\mathcal{X}-\mathcal{X}}(\mathcal{X})$; i.e. by taking the enriched center/Müger centralizer $Z^{\mathcal{A}}(\mathcal{X})$ [91, 112]. More generally, the tensor category of excitations localized to a domain wall is similarly a category of endomorphisms of the corresponding $\mathcal{A}$-enriched bimodule category. We explore this in detail in Construction 2.8 and Example 2.9 below. Since the $\mathcal{A}$-enriched center functor is fully faithful on the 1-truncation of $\mathrm{UFC}^{\mathcal{A}}$ [90] (see also § 3.2 below), we can use both perspectives on topological order side-by-side, and describe phenomena such as anyon condensation using the usual language of condensable algebras in a UMTC (see Appendix B for more details).

Instead of putting these bulk excitations front and center, however, our perspective should be viewed as describing topological orders in terms of the data which can be used to write down a (3+1)D commuting projector lattice model in which the desired topological order appears on the boundary. The role of the bulk is to trivialize the anomaly relative to $\mathcal{A}$, thereby enabling such a commuting projector realization. In § 2.2, we show by explicit construction how an $\mathcal{A}$-enriched fusion category $(\mathcal{X}, F)$ is exactly the necessary data to write down a commuting projector boundary of the Walker-Wang model [125] with bulk $\mathcal{A}$. This is parallel to how an honest fusion category is the necessary data to construct a Levin-Wen string net model [103]; in fact, string-net models occur as the special case $\mathcal{A} = \mathsf{Hilb}$.[7]

In this setting, we get the categorical description of topological order in Figure 3. In this framework, (2+1)D topological orders form a linear 3-category, where local operators at the top level can be used to describe tunneling operators, the decomposition of composite domain walls, and the spaces of ground states when a domain wall is placed along the equator of a sphere.

Applying our anyon mobility perspective on domain walls, we discuss at some length a particularly interesting class of composite domain walls, obtained by beginning with a $\mathcal{C}$ bulk region and condensing a condensate $A \in \mathcal{C}$ in the complement of a strip.[8] In other words, by composing two condensation boundaries between a $\mathcal{C}$ bulk and the $\mathcal{C}^{\mathrm{loc}}_A$ bulk obtained when $A$ is condensed, we obtain a domain wall between two $\mathcal{C}^{\mathrm{loc}}_A$ bulk regions. We will see that the superselection sectors of the composite domain wall are related to the topological ground state degeneracy within the strip of $\mathcal{C}$ bulk, when appropriate boundary conditions are imposed.

In particular, when the condensing anyons form a copy of the regular representation $\mathbb{C}^G$ of $G$ for a finite group $G$, then in the absence of excitations in the strip, different topological ground state sectors are associated with invertible boundaries carrying out different symmetry actions on the anyons in question. In this case, the boundaries represent $G$-crossed braided defects, the tunneling operators describe the corresponding braiding operation in the $G$-crossed braided category, and anyon condensation is associated with de-equivariantization of the cat-

---

[111] or [99, §4] for an introductory discussion of such UMTCs.

[7]In this paper, $\mathsf{Hilb}$ refers to the symmetric monoidal category of *finite dimensional* Hilbert spaces.

[8]Anyon condensation involves a choice of condensate $A$, which is identified with the vacuum where $A$ is condensed. Anyons in $A$ are then precisely those which can become condensed at the domain wall, or equivalently, those which can pass *across* the domain wall to become the vacuum.

egorical symmetry $G$. We emphasize that this choice of condensate is very special: for more general condensates, the superselection sectors of the composite boundary need not be invertible, meaning that some anyons cannot cross between the two $C_A^{\text{loc}}$ regions.

## 1.1 Outline

The structure of our paper is as follows. In § 2, we begin by explaining the description of (2+1)D topological orders and (1+1)D domain walls between them in terms of enriched fusion categories, including the passage back and forth between enriched fusion categories and the usual description in terms of categories of localized excitations. In § 3, we review the description of domain walls in terms of condensable algebras, including a detailed descriptions of how an arbitrary indecomposable domain wall factorizes as the composite of parallel invertible and condensation domain walls, as well as the mathematical operations involved in composing domain walls. We then build a description of how the composition of two parallel indecomposable domain walls splits into superselection sectors under the action of local operators. In § 4, we introduce sets of tunneling operators, and explain how indecomposable domain walls can be characterized by their tunneling operators. We then investigate the relationship between tunneling operators and composition of parallel domain walls, revealing how tunneling operators for a composite domain wall split up across the superselection sectors, allowing one to identify the resulting domain wall in each sector. Finally, in § 5, we work out the decompositions of several composite domain walls into superselection sectors, including non-Abelian examples and an example with nontrivial anomaly.

We include several appendices which contain well understood mathematical background material. Appendix A explains the basics of fusion categories and UMTCs, and Appendix B gives a review of condensable algebras. Appendix C.1 discusses $\mathcal{D}(G) := Z(\text{Hilb}(G))$ in detail, and § C.3 does the explicit example of the dihedral group $G = D_{2n}$.

## 1.2 Glossary

We end this introduction with a brief dictionary summarizing the correspondence between mathematical terminology and notation and the physical concepts related to topological order, which appears as Figure 4 below. The descriptions here are abbreviated, and this table should be interpreted as an expansion of [79, Table 1]. Note that the operations ⊡ and ⊠ are all usually denoted by ⊠ in the literature.

## 2 Enriched UFCs and domain walls between (2+1)D topologically ordered phases

In this section, we extend the 3-categorical description of anomaly-free topological order from [79] to the anomalous setting using enriched UFCs. That is, we replace the 3-category UFC of unitary fusion categories with the 3-category UFC$^A$ of UFCs enriched over a fixed UMTC $\mathcal{A}$ representing the anomaly; in the case $\mathcal{A} \cong \text{Hilb}$, we recover UFC. We also explain how taking the enriched center can be used to translate between the enriched setting and the description of bulk topological orders and domain walls via UMTCs and Witt-equivalences. In this way, our description contains all the information present in the UMTC/Witt-equivalence picture, but we show that it also contains additional structure which sheds light on domain wall composition.

In § 2.1 and 2.3, we explain enriched UFCs in further detail, as well as how modular categories describe topological order from the viewpoint of enriched UFCs. In § 2.2, we show

| Details of boundary for Walker-Wang model | Algebraic structure |
|---|---|
| (3+1)D bulk invertible TQFT anomaly | UMTC $\mathcal{A}$ |
| edge labels of bulk | simple objects in $\mathcal{A}$ |
| (2+1)D boundary theory | $\mathcal{A}$-enriched UFC $\mathcal{X}$, which is an object in the 3-category $\text{UFC}^{\mathcal{A}} \subset \text{UmFC}^{\mathcal{A}} := \text{UBFC}(\mathcal{A} \to \text{Hilb})$ |
| edge labels of boundary | simple objects in $\mathcal{X}$ |
| anyonic boundary excitations | enriched center/Müger centralizer $Z^{\mathcal{A}}(\mathcal{X}) = \mathcal{A}' \subset Z(\mathcal{X})$ |
| change of representative of anomaly | composition with Witt-equivalence ${}_{\mathcal{B}}\mathcal{W}_{\mathcal{A}}$ in UBFC by ${}_{\mathcal{B}}\mathcal{W}_{\mathcal{A}} \boxdot_{\mathcal{A}} - : \text{UmFC}^{\mathcal{A}} \to \text{UmFC}^{\mathcal{B}}$ |
| (1+1)D topological domain wall | $\mathcal{A}$-centered $\mathcal{X} - \mathcal{Y}$ bimodule $\mathcal{M}$ |
| edge labels of domain wall | simple objects in $\mathcal{M}$ |
| excitations on domain wall | objects in the category $\text{End}^{\mathcal{A}}_{\mathcal{X}-\mathcal{Y}}(\mathcal{M})$ of $\mathcal{A}$-centered $\mathcal{X} - \mathcal{Y}$ bimodule functors |
| condensate | condensable algebra $A \in Z^{\mathcal{A}}(\mathcal{X})$ |
| anyons in condensed region | simple objects in $Z^{\mathcal{A}}(\mathcal{X})^{\text{loc}}_A$ |
| excitations at boundary of condensed region | simple objects in $Z^{\mathcal{A}}(\mathcal{X})_A$ |
| edge labels for condensed region | simple objects in $\mathcal{X}_A$ |
| classification of topological domain walls | Lagrangian algebras $L(A, \overline{B}, \Phi) \in Z^{\mathcal{A}}(\mathcal{X}) \boxtimes \overline{Z^{\mathcal{A}}(\mathcal{Y})}$ |
| fusion of domain walls | relative Deligne product ${}_{\mathcal{X}}\mathcal{M} \boxdot_{\mathcal{Y}} \mathcal{N}_{\mathcal{Z}}$ |
| summands of composite domain wall | minimal projections in $Z^{\mathcal{A}}(\mathcal{Y})(B_1 \to B_2)$ where ${}_{\mathcal{X}}\mathcal{M}_{\mathcal{Y}} \longleftrightarrow L(A, \overline{B_1}, \Phi) \in Z^{\mathcal{A}}(\mathcal{X}) \boxtimes \overline{Z^{\mathcal{A}}(\mathcal{Y})}$ and ${}_{\mathcal{Y}}\mathcal{N}_{\mathcal{Z}} \longleftrightarrow L(B_2, \overline{C}, \Psi) \in Z^{\mathcal{A}}(\mathcal{Y}) \boxtimes \overline{Z^{\mathcal{A}}(\mathcal{Z})}$ |
| point defect | $\mathcal{A}$-centered $\mathcal{X} - \mathcal{Y}$ bimodule functor $F : \mathcal{M} \to \mathcal{N}$ |
| local operator | $\mathcal{X} - \mathcal{Y}$ bimodule natural transformation |
| fusion channel to transport anyon $c$ through domain wall to become $d$ | tunneling operator in $\text{Hom}_{\text{UFC}^{\mathcal{A}}}\left( \begin{array}{c} \text{[figure]} \end{array} \longrightarrow \begin{array}{c} \text{[figure]} \end{array} \right)$ |

Figure 4: Glossary for details of boundary for Walker-Wang models and algebraic higher categorical structure from $\text{UFC}^{\mathcal{A}}$, expanding on [79, Fig. 1].

how an enriched UFC gives rise to a lattice model for a chiral (2+1)D topological order on the boundary of a Walker-Wang model, including a concrete example. In § 2.4, we introduce enriched bimodules between enriched UFCs as the data which determine a (1+1)D domain wall. We also describe how to go back and forth between enriched UFCs and enriched bimodules and the UMTCs and Witt equivalences which describe bulk and wall excitations.

## 2.1 Topological orders and enriched fusion categories

In [79], the authors explain how each level of morphism in UFC labels an aspect of anomaly-free (2+1)D topological order, a correspondence which is summarized in Figure 2, and more extensively in [79, Table 1]. A unitary fusion category $\mathcal{X}$ is the input for the well-known Levin-Wen string net model [103], which is a commuting projector model of $Z(\mathcal{X})$ topological order. The 1-morphisms in $\text{UFC}(\mathcal{X} \to \mathcal{Y})$ are unitary $\mathcal{X} - \mathcal{Y}$ bimodule categories ${}_{\mathcal{X}}\mathcal{M}_{\mathcal{Y}}$, which determine commuting projector models for (1+1)D topological domain walls between the Levin-Wen models determined by $\mathcal{X}$ and $\mathcal{Y}$. (This is in contrast to the Witt equivalence bimodule categories mentioned previously, which are bimodules between the two UMTCs representing the topological orders.)

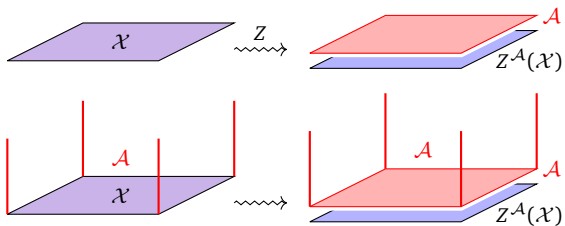

Figure 5: Since $Z(\mathcal{X}) \cong Z^{\mathcal{A}}(\mathcal{X}) \boxtimes \mathcal{A}$, attaching an $\mathcal{A}$-Walker-Wang bulk to $\mathcal{X}$ trivializes the $\mathcal{A}$-layer of topological order, leaving only $Z^{\mathcal{A}}(\mathcal{X})$.

In general, (2+1)D topologically ordered phases carry an *anomaly* [87]. This can be seen in the fact that the underlying mapping class group representations of surfaces appearing in the associated topological quantum field theory are projective, rather than honest. The anomaly is characterized by an invertible (3+1)D topological quantum field theory (TQFT) such that the original (2+1)D theory can be realized as a topological boundary [71, § III.B]. A concrete realization of this is the Walker-Wang construction [125], which takes as input a UMTC; the corresponding model has an invertible bulk, and can be cut off to realize the topological order associated with the corresponding UMTC on its boundary [123].

From a physical perspective, it may seem odd that we are claiming that a (3+1)D theory with boundary is describing a (2+1)D universality class. However, this can be understood conjecturally via [119, § 4]. The idea is that (3+1)D Walker-Wang models built from UMTCs can conjecturally be disentangled to a trivial phase by a quantum cellular automata (QCA). In this sense, we can consider the (3+1)D phase to be trivial, and a topological boundary of a (UMTC) Walker-Wang model is in the same universality class of a purely (2+1)D theory.

**Ansatz 2.1.** *The topological order of a (2+1)D topologically ordered system with anomaly described by the UMTC $\mathcal{A}$ is described by an $\mathcal{A}$-enriched UFC $(\mathcal{X}, F)$. The low energy excitations of this system are described by the enriched center $Z^{\mathcal{A}}(\mathcal{X})$.*

Here, an *$\mathcal{A}$-enriched unitary (multi)fusion category* consists of a pair $(\mathcal{X}, F)$, where $\mathcal{X}$ is a unitary (multi)fusion category and $F : \mathcal{A} \to Z(\mathcal{X})$ is a braided unitary tensor functor that takes anyon types (or more generally, objects) in $\mathcal{A}$ to anyon types (objects) in the Drinfeld center of $\mathcal{X}$. In what follows, we will frequently suppress $F$. The enriched center of $(\mathcal{X}, F)$ is the *Müger centralizer* $Z^{\mathcal{A}}(\mathcal{X}) := F(\mathcal{A})' \subset Z(\mathcal{X})$ [91, 112]. That is, $Z^{\mathcal{A}}(\mathcal{X})$ are those anyons in the usual Drinfeld center $Z(\mathcal{X})$ that braid trivially with (are centralized by) the image of $\mathcal{A}$.

Since $\mathcal{A}$ is nondegenerate and $F$ is fully faithful, the Drinfeld center $Z(\mathcal{X})$ can be factored: $Z(\mathcal{X}) \cong \mathcal{A} \boxtimes Z^{\mathcal{A}}(\mathcal{X})$ [112]. Physically, this means that $Z(\mathcal{X})$ describes two decoupled (2+1)D layers, one with $\mathcal{A}$ topological order and one with $Z^{\mathcal{A}}(\mathcal{X})$ topological order. By attaching an invertible (3+1)D topological order to the $\mathcal{A}$-layer, we can trivialize its (2+1)D topological order, leaving only the $Z^{\mathcal{A}}(\mathcal{X})$ topological order of interest (Fig. 5).

As we will see in § 2.4, the UMTC $Z^{\mathcal{A}}(\mathcal{X})$ of local excitations completely determines (up to Morita equivalence) the $\mathcal{A}$-enriched fusion category $\mathcal{X}$, so one can equivalently use the UMTC $Z^{\mathcal{A}}(\mathcal{X})$ to specify the topological order. However, Remark 3.13 will show that considering $\mathcal{A}$-enriched fusion categories (and bimodules between them) helps in developing a more complete understanding of defects between regions with (2+1)D topological order.

**Example 2.2.** In the case of trivial anomaly, i.e. $\mathcal{A} = \mathsf{Hilb}$, the enriched center is the ordinary Drinfeld center, and the above discussion agrees with the Levin-Wen description of localized excitations in string-net models [103].

**Example 2.3.** In the usual UMTC description of topological order, the UMTC $\mathcal{C}$ describing the (2+1)D topological order can be viewed as *self-enriched*, since $Z(\mathcal{C}) \cong \mathcal{C} \boxtimes \overline{\mathcal{C}}$. In this case, the UMTC representing the anomaly is $\mathcal{A} = \overline{\mathcal{C}}$, since $Z^{\overline{\mathcal{C}}}(\mathcal{C}) = \mathcal{C}$. We will see in the next section that the Walker-Wang model arises from this perspective [123, 125].

**Remark (Mathematical) 2.4.** A subset of the authors had long been troubled by the following 'off-by-one' inconsistency. It is expected (mostly from the sorts of pictures drawn in physical arguments) that (2+1)D topological orders together with topological domain walls and point defects should form a 3-category, possibly with some kind of symmetric monoidal product corresponding to stacking of phases.

However, it is generally agreed that UMTCs are the correct object to describe (2+1)D topological orders. These are naturally objects of the 4-category UBFC of unitary braided fusion categories [14, 63, 73] [74, §2.3], whose 1-morphisms are bimodule multifusion categories, 2-morphisms are compatible bimodule categories, 3-morphisms are compatible bimodule functors, and 4-morphisms are natural transformations. In addition, equivalence between objects in this 4-category is Witt equivalence by [74, Thm. 2.18], which is clearly the wrong equivalence relation for topological orders.

These inconsistencies are fixed by using enriched fusion categories to describe topological orders. Indeed, $\mathcal{A}$-enriched fusion categories form a 3-category $\mathrm{UFC}^{\mathcal{A}}$ which arises as the 3-subcategory of the Hom 3-category[9] $\mathrm{UBFC}(\mathcal{A} \to \mathrm{Hilb})$ whose objects are $\mathcal{A}$-enriched UFCs (as opposed to $\mathcal{A}$-enriched unitary multifusion categories (UmFCs)), where the UMTC $\mathcal{A}$ representing the anomaly is fixed.

$$\mathrm{UFC}^{\mathcal{A}} \subset \mathrm{UmFC}^{\mathcal{A}} = \mathrm{UBFC}(\mathcal{A} \to \mathrm{Hilb}).$$

The 3-category $\mathrm{UFC}^{\mathcal{A}}$, however, is not symmetric monoidal, as anomalies multiply. That is, given topological orders $(\mathcal{X}, F) : \mathcal{A} \to \mathrm{Hilb}$ and $(\mathcal{Y}, G) : \mathcal{B} \to \mathrm{Hilb}$, stacking (which is the natural tensor product in this category) gives us a topological order $(\mathcal{X} \boxtimes \mathcal{Y}, F \boxtimes G) : \mathcal{A} \boxtimes \mathcal{B} \to \mathrm{Hilb}$.

Note that UBFC includes into the Morita 4-category of fusion 2-categories via $\mathcal{C} \mapsto \mathrm{Mod}(\mathcal{C})$, and objects in this latter 4-category describe fully extended (3+1)D commuting projector lattice models of topological order [45]. The dimensional reduction appears because a (2+1)D topologically ordered phase occurs on the boundary of a (3+1)D invertible TQFT. While the natural pictures for these models are (3+1)D, since there are no bulk excitations in the Walker-Wang models associated to UMTCs [123], and we do not allow defects that extend into the bulk, we can just draw (2+1)D pictures of the boundary, which corresponds to looking at the 3-category $\mathrm{UFC}^{\mathcal{A}}$ and forgetting its origin as the hom-category $\mathrm{UBFC}(\mathcal{A} \to \mathrm{Hilb})$.

## 2.2 Walker-Wang type model for an enriched fusion category

To understand how the $\mathcal{A}$-enriched fusion category $(\mathcal{X}, F)$ realizes a given topological order, it is enlightening to examine the commuting projector models realizing our construction in more detail. Morally, our construction can be viewed as follows. Any Drinfeld center can be realized by a (2+1)D string net model, which is constructed from a UFC [79, 99, 103] The string net model associated to $\mathcal{X}$ (thought of as an ordinary fusion category) is a commuting projector model with anyons described by the UMTC $Z(\mathcal{X}) \cong Z^{\mathcal{A}}(\mathcal{X}) \boxtimes \mathcal{A}$. To trivialize the $\mathcal{A}$ layer, we attach this string net to the Walker-Wang model associated to $\mathcal{A}$ (see Fig. 5). This generalizes the topological boundary conditions for the Walker-Wang models considered in [123, 125], by gluing a suitable (2+1)D string net to the boundary; the resulting surface theory has the topological order $Z^{\mathcal{A}}(\mathcal{X})$.

---

[9]In this manuscript, if $\mathcal{C}$ is an $n$-category, then $\mathcal{C}(x \to y)$ denotes the $(n-1)$-category of morphisms from $x$ to $y$. For example, if $H$ and $K$ are finite dimensional Hilbert spaces, then $\mathrm{Hilb}(H \to K)$ is the linear 0-category, i.e. vector space, of linear operators from $H$ to $K$.

To make this construction explicit, we must specify how the bulk and boundary layers are attached. Here we describe in detail the simplest subset of these models, for which the fusion rules of $\mathcal{X}$ and $\mathcal{A}$ are multiplicity free, and that the composite $\mathcal{A} \to Z(\mathcal{X}) \to \mathcal{X}$ is fully faithful. This latter assumption means we can identify the anyons $a, b, c, \ldots$ in $\mathcal{A}$ with simple objects in $\mathcal{X}$, so that $\mathrm{Irr}(\mathcal{X})$ can be written as a disjoint union $\{a, b, c, \ldots\} \sqcup \{x, y, z, \ldots\}$ where $x, y, z, \ldots$ are the remaining simples in $\mathcal{X}$ not coming from $\mathcal{A}$. Moreover, given two anyons $a, b \in \mathrm{Irr}(\mathcal{A})$, the $\Omega$ tensor to describe the half-braiding for $F(a)$ with the image of $b \in \mathcal{X}$ is given by the $R$-matrix in $\mathcal{A}$. From these simplifications, the $R$-matrix for the $\mathcal{A}$-bulk and the $\Omega$-tensor for the $\mathcal{X}$-boundary string net, together with the choice of which subset of anyons in $Z(\mathcal{X})$ to identify with $\mathcal{A}$, is sufficient to fully describe the Hamiltonian. For the general case, we can add degrees of freedom to vertices as in [84], and the description of half-braidings requires more indices for the $\Omega$ tensors as in [99, (42,43)].

We begin with the usual brick-layer lattice for the Walker-Wang model, where red edges carry $\mathbb{C}^{\mathrm{Irr}(\mathcal{A})}$ spins labelled by anyons $a, b, c, \ldots$ in $\mathcal{A}$ and black edges carry $\mathbb{C}^{\mathrm{Irr}(\mathcal{X})}$ spins labelled by simple objects $\{a, b, c, \ldots,\} \sqcup \{x, y, z, \ldots\}$ of $\mathcal{X}$.

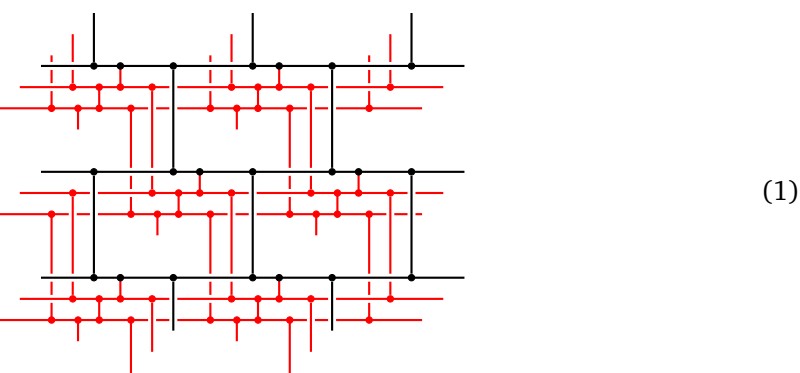

(1)

The Hamiltonian in the red $\mathcal{A}$-bulk is identical to the Walker-Wang Hamiltonian. There are vertex terms projecting to the subspace of admissible triples at that vertex, and the plaquette term uses the braiding of $\mathcal{A}$ to resolve the crossing.

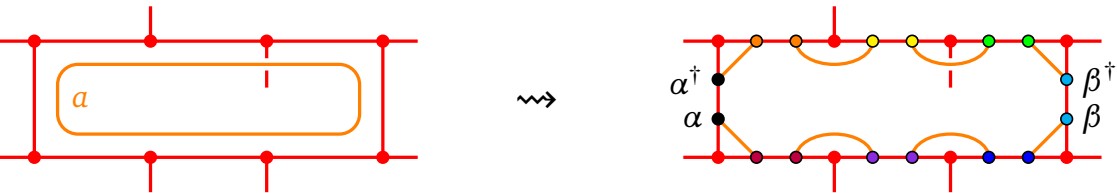

Here, we resolve the crossing by the formula

$$\left\langle \begin{array}{c} b \diagdown \diagup a \\ \diagtimes \end{array} \right| = \sum_y \overline{R}_c^{ba} \sqrt{\frac{d_c}{d_a d_b}} \left\langle \begin{array}{c} b \bullet a \\ c \\ a \bullet b \end{array} \right| \qquad (2)$$

We then use the $F$-symbols to resolve the diagram on the right hand side back to the original lattice, to obtain the matrix elements of the Hamiltonian.

The Hamiltonian on the black boundary has vertex and plaquette terms similar to the Levin-Wen string net model for $\mathcal{X}$, with two important changes. The vertices which have a red edge where the $\mathcal{A}$-bulk meets the $\mathcal{X}$ boundary must have $\mathbb{C}^{\mathrm{Irr}(\mathcal{A})}$ spins on the red edge and $\mathbb{C}^{\mathrm{Irr}(\mathcal{X})}$ on the black edges. By assumption, we can identify $\mathrm{Irr}(\mathcal{A})$ as a subset of $\mathrm{Irr}(\mathcal{X})$, so we use the usual Levin-Wen vertex term for $\mathcal{X}$ at these vertices. The plaquette term for the $\mathcal{X}$-boundary uses the half-braiding for the $\mathcal{A}$-anyons with $\mathcal{X}$ afforded by the (fully faithful)

central action $F : \mathcal{A} \to Z(\mathcal{X})$.

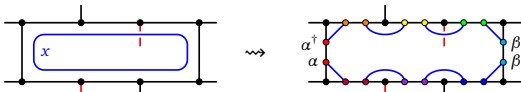

By our simplifying assumption, the anyons in $\mathcal{A}$ stay simple when we forget them down to $\mathcal{X}$, and so the $\Omega$-tensor (3) giving the $6j$-description of the half-braiding as in [99, (42,43)][10] can be substantially simplified. We resolve the crossing of the blue $x$-string from $\mathcal{X}$ with the red $a$-string from $\mathcal{A} \subset Z(\mathcal{X})$ by

$$\left\langle \; \overset{a}{\diagdown}\mkern-8mu\overset{x}{\diagup} \; \right| = \sum_y \overline{\Omega}_y^{a,x} \sqrt{\frac{d_y}{d_x d_a}} \left\langle \; \begin{matrix} a & x \\ & y \\ x & a \end{matrix} \; \right| . \tag{3}$$

To see the boundary excitations are indeed $Z^{\mathcal{A}}(\mathcal{X})$, we consider the category of excitations $Z(\mathcal{X})$ in the boundary string net model. Pairs of such excitations are created by quasiparticle string operators, as described in [99]. In order for the corresponding anyon to represent a point-like, deconfined excitation at the surface of our 3D model, we must be able to change the path of the anyon string operator arbitrarily away from its endpoints without creating additional excitations. Thus we must be able to move this path past the red links extending from the boundary to the bulk. This is exactly the condition that the anyon is centralized by $F(\mathcal{A})$, i.e., the excitations are given by $Z^{\mathcal{A}}(\mathcal{X})$.

**Remark 2.5.** In this section, our conventions for the $S, T$ matrices follow [7, (35) and (37)], which do not agree with those in [99, (58) and (62)]. The $S$-matrix in a UMTC $\mathcal{A}$ is given by

$$S_{a,b} := \frac{1}{D_{\mathcal{A}}} \cdot \; {}_a\!\!\bigcirc\mkern-12mu\bigcirc_{\!b} \; ,$$

where $D_{\mathcal{A}}$ is the square root of the global dimension of $\mathcal{A}$, and the $T$-matrix $T = \mathrm{diag}(\theta_a)_{a \in \mathrm{Irr}(\mathcal{A})}$ where

$$\theta_a := \frac{1}{d_a} \cdot \; {}_{\bar{a}}\!\!\bigcirc\mkern-4mu{}^a\mkern-4mu\bigcirc^{\bar{\bar{a}}}_{\bar{a}} \; .$$

### 2.2.1  Chiral example: $SU(2)_4$

To illustrate this construction in more detail, we show how to realize $SU(2)_4$ at the boundary of a Walker-Wang model with $SU(3)_1$-bulk. We reverse engineer this model by starting with $SU(2)_4$ and considering the domain wall (see § 2.4 below) coming from the conformal inclusion $SU(2)_4 \subset SU(3)_1$, which can be obtained by condensing the $\mathbb{Z}_2$ boson in $SU(2)_4$ to obtain $SU(3)_1$.

$$\boxed{SU(2)_4 \; \big| \; SU(3)_1}$$
$$\mathcal{TY}_{3,-}$$

The UMTC $SU(2)_4$ can be defined as the semsimple part of $\mathrm{Rep}(\mathcal{U}_q(\mathfrak{su}_2))$ at $q = \exp(2\pi i/12)$ [3, 16, 117], [83, § 6], or as the semisimple quotient by negligibles of the Temperley-Lieb-Jones category $\mathcal{TLJ}(s)$ with $s = \exp(\pi i/12)$. This latter skein-theoretic description has loop parameter

---

[10]We choose here a different convention for indices for our $\Omega$-tensors than in [99], as our assumptions of multiplicity free and the composite $\mathcal{A} \to Z(\mathcal{X}) \to \mathcal{X}$ being fully faithful let us use fewer indices. In particular, our index convention here is chosen to be as close as possible to the convention for the $R$-matrix. The two conventions are related by $\Omega_y^{a,x} = \Omega_a^{x,aay}$ from [99, (42)] where $a \in \mathrm{Irr}(\mathcal{A})$ and $x, y \in \mathrm{Irr}(\mathcal{X})$.

$$\bigcirc = -s^2 - s^{-2} = -\sqrt{3}$$

and braiding

$$\bigtimes := s \Big|\Big| + s^{-1} \;\overset{\smile}{\frown} \tag{4}$$

[53], [81, § 9], [124, §1.2], [49, §2]. Since the loop parameter for the strand is negative, the pivotal structure[11] on this braided fusion category is given by $\varphi_n := (-1)^n$ on the anyons $\{f_n\}$, which endows $SU(2)_4$ with the structure of a MTC. Under the dagger structure given by the conjugate linear extension of

$$(\frown)^\dagger := -\smile, \tag{5}$$

$SU(2)_4$ is a UMTC.[12]

The fusion and modular data of $SU(2)_4$ is as follows:

- anyons: $\{f_0 = 1, f_1, f_2, f_3, f_4 = g\}$

- fusion rules:

| $\otimes$ | $f_1$ | $f_2$ | $f_3$ | $g$ |
|-----------|-------|-------|-------|-----|
| $f_1$ | $1 + f_2$ | $f_1 + f_3$ | $f_2 + f_4$ | $f_3$ |
| $f_2$ | $f_1 + f_3$ | $1 + f_2 + g$ | $f_1 + f_3$ | $f_2$ |
| $f_3$ | $f_2 + f_4$ | $f_1 + f_3$ | $1 + f_2$ | $f_1$ |
| $g$ | $f_3$ | $f_2$ | $f_1$ | $1$ |

- quantum dimensions: $d_1 = d_g = 1$, $d_{f_1} = d_{f_3} = \sqrt{3}$, and $d_{f_2} = 2$

- associator/F-symbols: see [81, § 9.12] or [4, Appendix E]

- braiding/R-symbols: $\overset{a \quad b}{\underset{c}{\bigvee}} = R_c^{ab} \overset{a \quad b}{\underset{c}{\bigvee}}$ where

$$R_c^{ab} = (-1)^{\frac{a+b+c}{2}} s^{\frac{c(c+2)-a(a+2)-b(b+2)}{2}}$$

- S-matrix:[13]

$$\frac{1}{2\sqrt{3}} \begin{pmatrix} 1 & \sqrt{3} & 2 & \sqrt{3} & 1 \\ \sqrt{3} & \sqrt{3} & 0 & -\sqrt{3} & -\sqrt{3} \\ 2 & 0 & -2 & 0 & 2 \\ \sqrt{3} & -\sqrt{3} & 0 & \sqrt{3} & -\sqrt{3} \\ 1 & -\sqrt{3} & 2 & -\sqrt{3} & 1 \end{pmatrix}$$

[11]A *pivotal structure* on a fusion category is a trivialization of the double-dual functor, which consists of a scalar for each simple satisfying a coherence condition [48, § 4.7].

[12] It was determined in [53] when $\mathcal{TLJ}(s)$ is unitary. As stated in [124, § 1.4], $\mathcal{TLJ}(s)$ is unitary when $s = \pm i e^{\pm \frac{2\pi i}{24}}$. These 4 choices of $s$ give the 4 unitary braidings on the unitary Temperley-Lieb-Jones category which arises from subfactor theory [75] with dagger structure

$$(\frown)^\dagger := \smile.$$

We also have that $\mathcal{TLJ}(s)$ is unitary whenever $s = \pm e^{\pm \frac{2\pi i}{24}}$ with the dagger structure (5); these 4 choices of $s$ give the 4 unitary braidings on the underlying UFC of $SU(2)_4$.

[13]The $S$-matrix above was computed with the formula $S_{i,j} = [(i+1)(j+1)]_s$ where $[n]_s = \frac{s^{2n} - s^{-2n}}{s^2 - s^{-2}}$ for $s = \exp(\pi i/12)$. This formula is obtained from [124, p. 15] by including the pivotal structures $\varphi_i = (-1)^i$ for $f_i$ and $\varphi_j = (-1)^j$ for $f_j$.

- twists:[14] $1, e^{\pi i/4}, e^{2\pi i/3}, e^{-3\pi i/4}, 1$

It is helpful to use shorthand notation for the boson $f_4 =: g$ and the condensable algebra $1 + g =: A$. We refer the reader to Appendix B for background on condensable algebras and anyon condensation. The category of right $A$-modules in $SU(2)_4$, which describes excitations on the domain wall between $SU(2)_4$ and $SU(3)_1$ (see Example 3.2), is a $\mathbb{Z}/3$ Tambara-Yamagami unitary fusion category $\mathcal{TY}_{3,-}$,[15] and can be described as follows [122], [99, § VII E]:

- simple objects: $\{0, 1, 2, \sigma\}$

- fusion rules: $\mathbb{Z}/3$ for $\{0, 1, 2\}$ and $\sigma^2 = 0 + 1 + 2$

- quantum dimensions: $d_0 = d_1 = d_2 = 1$ and $d_\sigma = \sqrt{3}$

- associator/F-symbols: determined by the bicharacter $\langle a, b \rangle := \zeta^{-ab}$, where $\zeta := \exp(2\pi i/3)$ and a choice of sign:

$$F^{a\sigma b}_{\sigma\sigma\sigma} = F^{\sigma a \sigma}_{b\sigma\sigma} = \zeta^{-ab},$$
$$F^{\sigma\sigma\sigma}_{\sigma ab} = \frac{-1}{\sqrt{3}}\zeta^{ab}.$$

Since the generator $f_1$ of $SU(2)_4$ is pseudo-real, so is the non-invertible object $\sigma \in \mathcal{TY}_{3,-}$, which is reflected in the F-symbol $F^{\sigma\sigma\sigma}_{\sigma ab}$ above. (Observe $\sigma = f_1 + f_3$ in the category of right $A$-modules.)

The category of *local* right $A$-modules in $SU(2)_4$, corresponding to those wall excitation types which braid trivially with the condensate and thus remain deconfined, is $SU(3)_1$. This category is described by the following data, where again $\zeta = \exp(2\pi i/3)$ [116, 5.3.3]:

- anyons: $\{0, 1, 2\}$

- fusion rules: $\mathbb{Z}/3$

- quantum dimensions: $d_0 = d_1 = d_2 = 1$

- associator/F-symbols: trivial

- braiding/R-symbols:

$$R^{1,2}_0 = R^{2,1}_0 = \zeta^{-1},$$
$$R^{1,1}_2 = R^{2,2}_1 = \zeta.$$

(6)

- S-matrix: $\dfrac{1}{\sqrt{3}}\begin{pmatrix} 1 & 1 & 1 \\ 1 & \zeta & \zeta^2 \\ 1 & \zeta^2 & \zeta \end{pmatrix}$

- twists: $1, \zeta, \zeta$

---

[14]The twists above were computed with the formula $\theta_n = s^{n(n+2)}$ with $s = \exp(\pi i/12)$, which agrees with the formula from [83, §6], and can be proven by induction using the balance axiom and (4). Our formula differs from [81, § 9.7] by including the pivotal structure $\varphi_n = (-1)^n$ for $f_n$; see [67, (32) from Appendix A.2].

[15]There are four $\mathbb{Z}/3$ Tambara-Yamagami UFCs corresponding to a choice of bicharacter $\langle a, b \rangle = \zeta^{\pm ab}$ and a choice of sign $\pm$ corresponding to the Frobenius-Schur indicator of $\sigma$. For $SU(2)_4$, this sign must be $-1$, and the two UFCs corresponding to the $\pm$ bicharacters give monoidally opposite UFCs. Their centers differ by reversing the braiding; the UFC with $\zeta^{ab}$ bicharacter has center $\overline{SU(2)_4} \boxtimes SU(3)_1$, and the UFC with $\zeta^{-ab}$ bicharacter has center $SU(2)_4 \boxtimes \overline{SU(3)_1}$ as desired.

By [42, Cor. 3.30],

$$Z(\mathcal{TY}_{3,-}) \cong SU(2)_4 \boxtimes \overline{SU(3)_1}, \tag{7}$$

so setting $\mathcal{A} := \overline{SU(3)_1}$ and $\mathcal{X} := \mathcal{TY}_{3,-}$, we have $Z^{\mathcal{A}}(\mathcal{X}) = SU(2)_4$. In the lattice model for this example, in (1), every red edge has $\mathbb{C}^3$ spins labelled by $\mathbb{Z}/3 = \{0,1,2\}$, and every black edge has $\mathbb{C}^4$ spins corresponding to $\mathrm{Irr}(\mathcal{X}) = \{0,1,2,\sigma\}$. Vertices at the surface impose the fusion rules of $\mathcal{TY}_{3,-}$, with the labels $0,1,2$ in the bulk being treated as equivalent to labels $0,1,2$ in the boundary under fusion.

As $\mathcal{A}$ has only Abelian anyons, $d_0 = d_1 = d_2 = 1$, resolving the crossing (2) is as easy as $y = x + z$:

$$\left\langle \;\vcenter{\hbox{\includegraphics{}}}\; \right| = \overline{R}^{b,a}_{(a+b)} \left\langle \;\vcenter{\hbox{\includegraphics{}}}\; \right|,$$

where $\overline{R}$ is the reverse $R$-matrix of (6). Resolving (3) to describe the boundary plaquette terms is also easier since for any bulk edge label $a$, $d_a = 1$, and $d_x = d_y$ as $xa = y = ax$. In particular, $\Omega^{a,x}_y = R^{a,x}_y$ whenever $x,y \in \{0,1,2\}$, so the only extra data needed are the $1 \times 1$ unitary matrices $\Omega^{\sigma,\sigma}_a$:

$$\left\langle \;\vcenter{\hbox{\includegraphics{}}}\; \right| = \overline{\Omega}^{a,\sigma}_\sigma \left\langle \;\vcenter{\hbox{\includegraphics{}}}\; \right|.$$

These $\Omega$-symbols are given by

$$\Omega^{0,\sigma}_\sigma = 1, \quad \Omega^{1,\sigma}_\sigma = \zeta, \quad \Omega^{2,\sigma}_\sigma = \zeta.$$

### 2.2.2 The Drinfeld center of $\mathcal{TY}_{3,-}$

The centers of the Tambara-Yamagami UFCs were first computed in [70, § 3]. For completeness, we list here the 15 simple objects in $Z(\mathcal{TY}_{3,-})$ and the data of the $\Omega$-tensor using the conventions of [99];[16] the $\overline{\Omega}$-tensor is determined by the $\Omega$-tensor by [99, (48b) and (48c)]. Here, we use the original notation of [99] (see Footnote 10) as some of the objects in the center are direct sums of simples in $\mathcal{TY}_{3,-}$.

- invertibles $\alpha_{g,j}$ for each element $g \in \mathbb{Z}/3$ and $j = 0,1$, for a total of 6 abelian anyons. The underlying object of $\alpha_{g,j}$ is $g \in \mathcal{TY}_{3,-}$, so $\dim(\alpha_{g,j}) = 1$.

$$
\begin{array}{ll}
\underline{\alpha = \alpha_{0,j}:} & \Omega^{1,001}_\alpha = \Omega^{2,002}_\alpha = 1 \\
& \Omega^{\sigma,00\sigma}_\alpha = (-1)^j \\
\underline{\alpha = \alpha_{1,j}:} & \Omega^{1,112}_\alpha = \zeta^{-1} \\
& \Omega^{2,110}_\alpha = \zeta \\
& \Omega^{\sigma,11\sigma}_\alpha = (-1)^j \zeta^{-1} \\
\underline{\alpha = \alpha_{2,j}:} & \Omega^{1,220}_\alpha = \zeta \\
& \Omega^{2,221}_\alpha = \zeta^{-1} \\
& \Omega^{\sigma,22\sigma}_\alpha = (-1)^j \zeta^{-1}
\end{array}
$$

---

[16]The $\Omega$-tensor for the $\mathbb{Z}/3$ Tambara-Yamagami UFCs with bicharacter $\zeta^{ab}$ were computed by Chien-Hung Lin. The case with $+1$ Frobenius-Schur indicator appears in [99], and the case with $-1$ Frobenius-Schur indicator is commented out in the `arXiv` source of [99]. Taking complex conjugate gives the $\Omega$ tensors for the other two $\mathbb{Z}/3$ Tambara-Yamagami UFCs (see Footnote 15). Indeed, the complex conjugate UFC is equivalent to the opposite UFC by taking dagger, and the opposite is equivalent to the monoidal opposite by taking duals. We present here the $\Omega$-tensor for the bicharacter $\zeta^{-ab}$ and sign $-1$ by taking the complex conjugate of this commented out data with Chien-Hung Lin's permission.

- 1 simple $\gamma_{g,h}$ for each distinct pair of elements $g,h \in \mathbb{Z}/3$ whose underlying object is $g \oplus h$ in $\mathcal{TY}_{3,-}$, so $\dim(\gamma_{g,h}) = 2$. The $\Omega$-tensors are determined up to three $U(1)$ gauge phases $\phi_1, \phi_2, \phi_3$:

$\underline{\gamma = \gamma_{0,1}}$: $\Omega_\gamma^{1,001} = \zeta^{-1}$
$\qquad\qquad \Omega_\gamma^{2,002} = \zeta$
$\qquad\qquad \Omega_\gamma^{\sigma,00\sigma} = \Omega_\gamma^{\sigma,11\sigma} = 0$
$\qquad\qquad \Omega_\gamma^{1,112} = \Omega_\gamma^{2,110} = 1$
$\qquad\qquad \Omega_\gamma^{\sigma,01\sigma} = e^{i\phi_1}$
$\qquad\qquad \Omega_\gamma^{\sigma,10\sigma} = e^{-i\phi_1}$

$\underline{\gamma = \gamma_{0,2}}$: $\Omega_\gamma^{1,001} = \zeta$
$\qquad\qquad \Omega_\gamma^{2,002} = \zeta^{-1}$
$\qquad\qquad \Omega_\gamma^{3,00\sigma} = \Omega_\gamma^{3,22\sigma} = 0$
$\qquad\qquad \Omega_\gamma^{1,220} = \Omega_\gamma^{2,221} = 1$
$\qquad\qquad \Omega_\gamma^{\sigma,02\sigma} = e^{i\phi_2}$
$\qquad\qquad \Omega_\gamma^{\sigma,20\sigma} = e^{-i\phi_2}$

$\underline{\gamma = \gamma_{1,2}}$: $\Omega_\gamma^{1,112} = \Omega_\gamma^{2,221} = \zeta$
$\qquad\qquad \Omega_\gamma^{2,110} = \Omega_\gamma^{1,220} = \zeta^{-1}$
$\qquad\qquad \Omega_\gamma^{\sigma,11\sigma} = \Omega_\gamma^{\sigma,22\sigma} = 0$
$\qquad\qquad \Omega_\gamma^{\sigma,12\sigma} = \zeta^{-1}e^{i\phi_3}$
$\qquad\qquad \Omega_\gamma^{\sigma,21\sigma} = e^{-i\phi_3}$

- 2 simples $\delta_{g,j}$ for each $g \in \mathbb{Z}/3$ and $j = 0,1$ whose underlying object is $\sigma \in \mathcal{TY}_{3,-}$, so $\dim(\delta_{g,j}) = 1$.

$\underline{\delta = \delta_{0,j}}$: $\Omega_\delta^{1,\sigma\sigma\sigma} = \Omega_\delta^{2,\sigma\sigma\sigma} = \zeta$
$\qquad\qquad \Omega_\delta^{\sigma,\sigma\sigma 0} = (-1)^j e^{-\pi i/4}$
$\qquad\qquad \Omega_\delta^{\sigma,\sigma\sigma 1} = \Omega_\delta^{\sigma,\sigma\sigma 2} = -(-1)^j e^{-11\pi i/12}$

$\underline{\delta = \delta_{1,j}}$: $\Omega_\delta^{1,\sigma\sigma\sigma} = 1$
$\qquad\qquad \Omega_\delta^{2,\sigma\sigma\sigma} = \zeta^{-1}$
$\qquad\qquad \Omega_\delta^{\sigma,\sigma\sigma 0} = \Omega_\delta^{\sigma,\sigma\sigma 2} = -(-1)^j e^{-7\pi i/12}$
$\qquad\qquad \Omega_\delta^{\sigma,\sigma\sigma 1} = (-1)^j e^{-11\pi i/12}$

$\underline{\delta = \delta_{2,j}}$: $\Omega_\delta^{1,\sigma\sigma\sigma} = \zeta^{-1}$
$\qquad\qquad \Omega_\delta^{2,\sigma\sigma\sigma} = 1$
$\qquad\qquad \Omega_\delta^{\sigma,\sigma\sigma 0} = \Omega_\delta^{\sigma,\sigma\sigma 1} = -(-1)^j e^{-7\pi i/12}$
$\qquad\qquad \Omega_\delta^{\sigma,\sigma\sigma 2} = (-1)^j e^{-11\pi i/12}$

The $S, T$-matrices for $Z(\mathcal{TY}_{3,-})$ (see Remark 2.5) are given by [70, Thm. 3.6]. First, for each $g \in \mathbb{Z}/3$, we let $\omega_g$ be a square root of $(-1)^g \cdot i \cdot \exp(-g^2\pi i/3)$. We choose

$$\omega_0 = e^{\pi i/4}, \qquad \omega_1 = \omega_2 = e^{7\pi i/12}.$$

The twists are given by:

- $T(\alpha_{g,j}) = \langle g, g \rangle = \zeta^{-g^2}$ which is 1 if $g = 0$ and $\zeta^{-1}$ otherwise.

- $T(\gamma_{g,h}) = \langle g, h \rangle = \zeta^{-gh}$ which is 1 if $g = 0$ and $\zeta$ if $g = 1$ and $h = 2$.

- $T(\delta_{g,j}) = (-1)^j \omega_g$, giving the following twists:

$$e^{\pi i/4}, \quad e^{5\pi i/4}, \quad e^{7\pi i/12}, \quad e^{19i\pi/12}, \quad e^{7\pi i/12}, \quad e^{19\pi i/12}.$$

We give a table of the twists of all anyons in $Z(\mathcal{TY}_{3,-})$ in (8) below. The block $S$-matrix is given by:

| | $\alpha_{k,j}$ | $\gamma_{k,\ell}$ | $\delta_{k,j}$ |
|---|---|---|---|
| $\alpha_{g,i}$ | $\dfrac{1}{6}\overline{\langle g,k\rangle}^2$ | $\dfrac{1}{3}\overline{\langle g,k+\ell\rangle}$ | $\dfrac{(-1)^i}{2\sqrt{3}}\overline{\langle g,k\rangle}$ |
| $\gamma_{g,h}$ | $\dfrac{1}{3}\overline{\langle g+h,k\rangle}$ | $\dfrac{1}{3}\overline{\langle g,\ell\rangle\langle h,k\rangle + \langle g,k\rangle\langle h,\ell\rangle}$ | $0$ |
| $\delta_{g,i}$ | $\dfrac{(-1)^j}{2\sqrt{3}}\overline{\langle k,g\rangle}$ | $0$ | $\dfrac{(-1)^{i+j}\omega_g\omega_k}{6}\sum_\ell \overline{\langle \ell-(g+k),\ell\rangle}$ |

Ordering the anyons of $Z(\mathcal{TY}_{3,-})$ as follows (the $\phi$ such that the twist $\theta = e^{\phi\pi i}$ is matched for convenience)

| | $\alpha_{0,0}=1$ | $\alpha_{1,0}$ | $\alpha_{2,0}$ | $\delta_{0,0}$ | $\delta_{2,1}$ | $\delta_{1,1}$ | $\gamma_{1,2}$ | $\gamma_{0,2}$ | $\gamma_{0,1}$ | $\delta_{0,1}$ | $\delta_{2,0}$ | $\delta_{1,0}$ | $\alpha_{0,1}$ | $\alpha_{1,1}$ | $\alpha_{2,1}$ |
|---|---|---|---|---|---|---|---|---|---|---|---|---|---|---|---|
| $\phi$ | $0$ | $\dfrac{-2}{3}$ | $\dfrac{-2}{3}$ | $\dfrac{1}{4}$ | $\dfrac{19}{12}$ | $\dfrac{19}{12}$ | $\dfrac{2}{3}$ | $0$ | $0$ | $\dfrac{5}{4}$ | $\dfrac{7}{12}$ | $\dfrac{7}{12}$ | $0$ | $\dfrac{-2}{3}$ | $\dfrac{-2}{3}$ |

(8)

we see that the $S,T$-matrices of $Z(\mathcal{TY}_{3,-}) = Z(\mathcal{X})$ are exactly the tensor product of the $S,T$-matrices of $SU(2)_4 \cong Z^{\mathcal{A}}(\mathcal{X})$ and $\overline{SU(3)_1} = \mathcal{A}$ respectively. Here, the anyons in violet and red generate the copy of $\overline{SU(3)_1}$ in $Z(\mathcal{TY}_{3,-})$, and the anyons in violet and blue generate the copy of $SU(2)_4$.

## 2.3 Change of enrichment and 1-composition in UBFC

Comparing the model described above to the original construction of [125], it is evident that the choice of bulk is not unique. This reflects the fact that, by the cobordism hypothesis [12, 101], anomalies are characterized by a *Witt class* of UMTCs [15]. Here, the Witt class of a UMTC $\mathcal{A}$ is all UMTCs $\mathcal{B}$ such that $\mathcal{A}\boxtimes\overline{\mathcal{B}}$ is braided equivalent to the Drinfeld center [42] $Z(\mathcal{W})$ of a UFC $\mathcal{W}$; when $\mathcal{W}$ has such a braided tensor equivalence $\mathcal{A}\boxtimes\overline{\mathcal{B}} \to Z(\mathcal{W})$, we call it a *Witt equivalence* from $\mathcal{A}$ to $\mathcal{B}$.[17] In the previous section, for example, we saw that $\mathcal{TY}_{3,-}$ is a Witt equivalence between $SU(3)_1$ and $SU(2)_4$.

Physically, this means that Witt equivalent UMTC's determine invertible bulks which can realize the same set of boundary topological orders, and also that Witt equivalent UMTCs can appear as surface topological orders of the same invertible bulk. In the previous subsection, for example, the standard boundary conditions of [125] lead to $SU(3)_1$ surface topological order, and we could have obtained the same $SU(2)_4$ surface topological order from a bulk theory $\mathcal{A} = SU(2)_4$. This suggests that these two Walker-Wang models should, in some sense, be equivalent, since they can realize the same set of topological orders at their boundaries. Indeed, it is widely believed that two Walker-Wang models are related by a finite depth quantum circuit if, and only if, they are Witt equivalent [119]. This leads to the conjecture in [119, § 4] that the group of QCA modulo finite depth quantum circuits is isomorphic to the Witt group of UMTCs.

---

[17]Note that here, the bimodule tensor category $\mathcal{W}$ is not describing the excitations on a (1+1)D domain wall between (2+1)D bulks, but the data of a commuting projector model of a (2+1)D domain wall between (3+1)D bulks. One can, however, interpret this as a mapping from edge labels in $\mathcal{A}$ and $\mathcal{B}$ to anyons in the string-net model associated to $\mathcal{W}$.

Explicitly, we can change the choice of which bulk theory describes the anomaly by composing with an appropriate invertible 1-morphism $\mathcal{B} \to \mathcal{A}$ in the 4-category UBFC, giving an invertible 3-functor $\mathrm{UmFC}^{\mathcal{A}} \to \mathrm{UmFC}^{\mathcal{B}}$. This composition should be viewed as stacking a Witt-equivalence $\mathcal{W}$ in the 3D bulk on top of the 2D boundary:

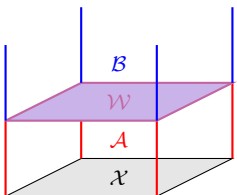

The $\mathcal{X}$-labelled boundary hosts the topological order $Z^{\mathcal{A}}(\mathcal{X})$. The surface labeled by $\mathcal{W}$ represents a bulk defect implementing the Witt equivalence between $\mathcal{A}$ and $\mathcal{B}$. If we collapse the $\mathcal{A}$-bulk region to the boundary, we can think of the parallel $\mathcal{X}$ and $\mathcal{W}$ sheets as a single (2+1)D topological boundary from the $\mathcal{B}$-bulk to vacuum, which supports the same UBFC of localized excitations as the original boundary labelled by $\mathcal{X}$. This boundary will be equivalent to the boundary determined by the $\mathcal{B}$-enriched multifusion category $\mathcal{W} \boxdot_{\mathcal{A}} \mathcal{X}$, which we will define below.

In order to understand this stacking operation, let us discuss the bulk defect labelled by $\mathcal{W}$ in more detail. In the Walker-Wang model, such a defect is obtained by inserting a layer of the string net constructed from the fusion category $\mathcal{W}$. To attach this layer to the Walker-Wang bulk, we adopt the same strategy as in § 2.2, using the functor $\overline{\mathcal{A}} \to Z(\mathcal{W})$ to attach the $\mathcal{A}$ Walker-Wang bulk from below. If $\mathcal{W}$ is a Witt equivalence (i.e., if $F$ is a braided equivalence), then this defect will be invertible by [74, Thm. 2.18]. Invertibility means that there is another boundary, namely $\mathcal{W}^{\mathrm{mp}}$,[18] which can be stacked with $\mathcal{W}$ to give the trivial boundary. This implies that in the absence of further defects, an invertible boundary cannot be detected using operators localized near the 2D defect; in particular, the defect plane does not have any topological order or anyons.

We now describe how to stack defects corresponding to Witt equivalences. As noted above, stacking with the invertible boundary $\mathcal{W}$ corresponds to composing with the 1-morphism $\mathcal{W}$ in UBFC. Mathematically, this stacking operation is given by the relative Deligne product $\boxdot_{\mathcal{A}}$.

**Warning 2.6.** The operation $\boxdot_{\mathcal{A}}$ is generally written $\boxtimes_{\mathcal{A}}$, since it is related to the (non-relative) Deligne product $\boxtimes$. However, in this paper, several operations which have different mathematical definitions and/or physical meanings, but are all conventionally denoted by $\boxtimes_{\mathcal{A}}$, will appear. We therefore introduce this unconventional notation as a disambiguation.

**Definition 2.7.** Given a UMTC $\mathcal{A} \in \mathrm{UBFC}$, we define the *canonical Lagrangian algebra*

$$K_{\mathcal{A}} := \bigoplus_{a \in \mathrm{Irr}(\mathcal{A})} a \boxtimes \overline{a} \in \mathcal{A} \boxtimes \overline{\mathcal{A}}. \tag{9}$$

Here, the term Lagrangian algebra refers to the fact that the category $(\mathcal{A} \boxtimes \overline{\mathcal{A}})^{\mathrm{loc}}_{K_{\mathcal{A}}}$ of local modules is just $\mathrm{Hilb}_{\mathrm{fd}}$ – in other words, condensing the anyons in $K_{\mathcal{A}}$ leads to a trivial topological order. See Appendix B for details. Note that, since the Deligne tensor product is symmetric, $\mathcal{A} \boxtimes \overline{\mathcal{A}} \cong \overline{\mathcal{A}} \boxtimes \mathcal{A}$, and this canonical equivalence takes $K_{\mathcal{A}}$ to $K_{\overline{\mathcal{A}}}$. As such, we denote both algebras by $K_{\mathcal{A}}$.

Given an $\mathcal{A} - \mathcal{B}$ bimodule fusion category $\mathcal{U}$ and a $\mathcal{B} - \mathcal{C}$ bimodule fusion category $\mathcal{V}$ where $\mathcal{A}, \mathcal{B}, \mathcal{C} \in \mathrm{UBFC}$ are UMTCs, following [74, §2.3] based on [14], we define the 1-composite

$$\mathcal{U} \boxdot_{\mathcal{B}} \mathcal{V} := (\mathcal{U} \boxtimes \mathcal{V})_{K_{\mathcal{B}}}. \tag{10}$$

---

[18]Given a (multi)fusion category $\mathcal{X}$, $\mathcal{X}^{\mathrm{mp}}$ is the (multi)fusion category obtained from $\mathcal{X}$ by reversing the monoidal product.

In other words, $\mathcal{U} \boxdot_{\mathcal{B}} \mathcal{V}$ is given by the category of $K_{\mathcal{B}}$-modules in $\mathcal{U} \boxtimes \mathcal{V}$ (see Appendix B). Here, $K_{\mathcal{B}} \in \overline{\mathcal{B}} \boxtimes \mathcal{B} \subset Z(\mathcal{U} \boxtimes \mathcal{V})$ is given by the analog of (9) for $\overline{\mathcal{B}}$. The composite defect $\mathcal{U} \boxdot_{\mathcal{B}} \mathcal{V}$ is an $\mathcal{A} - \mathcal{C}$ bimodule multifusion[19] category, describing a Witt equivalence between $\mathcal{A}$ and $\mathcal{C}$. See § 2.4 below for a definition of bimodule multifusion category (which appears with a different physical interpretation!).

This composition has a concrete realization in terms of Walker-Wang models. Consider a stack of three initially decoupled Walker-Wang layers, with bulks constructed from three Witt equivalent UMTC's $\mathcal{A}, \mathcal{B},$ and $\mathcal{C} \in$ UBFC respectively. We first attach a string-net associated with the fusion category $\mathcal{U}$ to the bottom of the $\mathcal{A}$ layer, as described in Sec. 2.2. Using an reflected procedure, we then attach a $\mathcal{V}$ string net to the top of the $\mathcal{C}$ layer. In the case that $\mathcal{U}, \mathcal{V}$ are Witt equivalences to $\mathcal{B}$ from $\mathcal{A}, \mathcal{C}$ respectively, we now have a Walker-Wang model with a $\mathcal{A}$ bulk and surface $\overline{\mathcal{B}}$ topological order at the boundary on the bottom, and a Walker-Wang model with a $\mathcal{C}$ bulk and surface $\mathcal{B}$ topological order at the boundary on the top. Finally, we must connect the $\mathcal{U}$ and $\mathcal{V}$ string nets in such a way that their topological order is trivialized, by taking modules over $K_{\mathcal{B}}$ as in (10). Physically, there are two ways to view this process. On the one hand, we can bring together the (2+1)D $\mathcal{B}$ boundary of the bottom half of our system and the $\overline{\mathcal{B}}$ boundary from the top half, and annihilate the resulting topological order by condensing the anyons in $K_{\mathcal{B}}$. On the other hand, we can also leave these boundaries spatially separated, and insert a slab of the Walker-Wang bulk ground state with edges labeled by objects in $\mathcal{B}$ to connect the $\mathcal{U}$ and $\mathcal{V}$ defects. Both of these are valid physical interpretations of the mathematical process of taking modules over $K_{\mathcal{B}}$. The $\mathcal{A} - \mathcal{C}$ bimodule tensor category $\mathcal{U} \boxdot_{\mathcal{B}} \mathcal{V}$ describes the remaining surface topological order near the $\mathcal{B}$-slab.

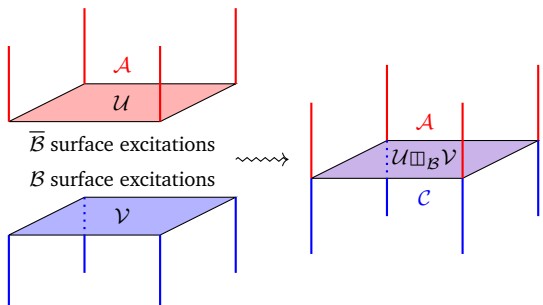

Figure 6: Taking $\mathcal{K}_{\mathcal{B}}$-modules in $\mathcal{U} \boxtimes \mathcal{V}$ glues the Witt-equivalences ${}_{\mathcal{A}}\mathcal{U}_{\mathcal{B}}$ and ${}_{\mathcal{B}}\mathcal{V}_{\mathcal{C}}$ to obtain the Witt-equivalence ${}_{\mathcal{A}}(\mathcal{U} \boxdot_{\mathcal{B}} \mathcal{V})_{\mathcal{C}}$.

In the case that $\mathcal{U}$ and $\mathcal{V}$ are Witt equivalences, this trivializes both surface topological orders, confining all the surface anyons, as seen by the fact that $\mathcal{U} \boxdot_{\mathcal{B}} \mathcal{V}$ is again a Witt equivalence between $\mathcal{A}$ and $\mathcal{C}$. Indeed, by [42, Thm. 3.20],

$$\begin{aligned}
Z(\mathcal{U} \boxdot_{\mathcal{B}} \mathcal{V}) &\cong Z((\mathcal{U} \boxtimes \mathcal{V})_{K_{\mathcal{B}}}) \\
&\cong Z(\mathcal{U} \boxtimes \mathcal{V})^{\text{loc}}_{K_{\mathcal{B}}} \\
&\cong (\overline{\mathcal{A}} \boxtimes \mathcal{B} \boxtimes \overline{\mathcal{B}} \boxtimes \mathcal{C})^{\text{loc}}_{K_{\mathcal{B}}} \\
&\cong \overline{\mathcal{A}} \boxtimes \mathcal{C}.
\end{aligned}$$

Mathematically, the point is that Witt equivalences are invertible 1-morphisms in UBFC, and the composition of two invertible morphisms will be invertible.

---

[19]Given a connected separable algebra $A$ in a fusion category $\mathcal{X}$ which lifts to a commutative (and thus condensable) algebra in $Z(\mathcal{X})$, $\mathcal{X}_A$ is again a fusion category [48]. However, the image of $K_{\mathcal{B}}$ in $\mathcal{U} \boxtimes \mathcal{V}$ is usually not connected. Precisely, $K_{\mathcal{B}}$ will only be sent to a connected algebra in $\mathcal{U} \boxtimes \mathcal{V}$ when no anyon in the $\mathcal{B}$ bulk can condense on both domain walls.

This also gives a simple picture of the case we are interested in: composing a (2+1)D topological order, which we describe as a boundary between the Walker-Wang model associated to the UMTC $\mathcal{A}$ and the vacuum, with a Witt equivalence defect between the Walker-Wang models associated to the UTMCs $\mathcal{A}$ and $\mathcal{B}$. We begin with a bulk $\mathcal{A}$ region, with surface topological order described by an $\mathcal{A}$-enriched fusion category $\mathcal{X}$. We introduce an invertible 1-morphism $\mathcal{W} \in \mathsf{UBFC}(\mathcal{B} \to \mathcal{A})$, corresponding to a Witt-equivalence $\mathcal{B} \simeq \mathcal{A}$, into the bulk, and push this defect to the boundary. The composite boundary is described by the $\mathcal{B}$-enriched multifusion category $\mathcal{W} \boxdot_{\mathcal{A}} \mathcal{X}$. Just as in the case of composing two Witt equivalences, we have

$$
\begin{aligned}
\mathcal{B} \boxtimes Z^{\mathcal{B}}(\mathcal{W} \boxdot_{\mathcal{A}} \mathcal{X}) &\cong Z(\mathcal{W} \boxdot_{\mathcal{A}} \mathcal{X}) \\
&\cong (Z(\mathcal{W}) \boxtimes Z(\mathcal{X}))^{\mathrm{loc}}_{K_{\mathcal{A}}} \\
&\cong (\mathcal{B} \boxtimes \overline{\mathcal{A}} \boxtimes \mathcal{A} \boxtimes Z^{\mathcal{A}}(\mathcal{X}))^{\mathrm{loc}}_{K_{\mathcal{A}}} \\
&\cong \mathcal{B} \boxtimes Z^{\mathcal{A}}(\mathcal{X}),
\end{aligned}
$$

so $Z^{\mathcal{B}}(\mathcal{W} \boxdot_{\mathcal{A}} \mathcal{X}) \cong Z^{\mathcal{A}}(\mathcal{X})$. In other words, the stacking operation depicted above indeed results in a model with a different, Witt equivalent bulk theory $\mathcal{B}$, but the same boundary topological order. Evidently, both bulk enrichments are equally valid from the point of view of the boundary theory.

## 2.4 Topological domain walls and enriched bimodule categories

We begin by sketching the mathematical data and physical implications of a topological domain wall separating two regions with (2+1)D topological order, with anyons in the bulks described by the UMTCs $\mathcal{C}$ and $\mathcal{D}$ respectively. Such domain walls have been studied in [59, 79, 86, 97, 105].

One way to describe a domain wall is to characterize the point-like excitations localized on the wall. These correspond to simple objects of a UmFC $\mathcal{W}$, similar to how anyons correspond to simple objects in a UMTC. The category $\mathcal{W}$ is equipped with additional structure, describing how wall excitations interact with bulk excitations. Mathematically, these take the form of two monoidal functors $F : \mathcal{C} \to \mathcal{W}$ and $G : \overline{\mathcal{D}} \to \mathcal{W}$, dictating which wall excitation is obtained from fusing an anyon from the $\mathcal{C}$ and $\mathcal{D}$ bulk regions, respectively, onto the domain wall. These functors lift to give braided monoidal functors $F, G : \mathcal{C}, \overline{\mathcal{D}} \to Z(\mathcal{W})$, because when fusing a bulk excitation $c$ with a wall excitation $w$, we can bring $c$ to the wall on either side of $w$, relating the two products $F(c)w$ and $wF(c)$ by a half-braiding on $F(c)$. This half-braiding is exactly the data needed to construct the Drinfeld center $Z(\mathcal{W})$ [59, § 3]. Particles from $\mathcal{C}$ are transparent to particles from $\overline{\mathcal{D}}$ inside $Z(\mathcal{W})$, because they approach the domain wall from opposite sides, meaning that the two action functors assemble into a single braided tensor functor $\mathcal{C} \boxtimes \overline{\mathcal{D}} \to Z(\mathcal{W})$. The resulting mathematical structure is a unitary multifusion category $\mathcal{W}$ equipped with actions of $\mathcal{C}$ and $\mathcal{D}$ that are central, in the sense that the images of anyons in $\mathcal{C}$ and $\mathcal{D}$ can braid past wall excitations in $\mathcal{W}$. Equipped with these actions, $\mathcal{W}$ is called a $\mathcal{C} - \mathcal{D}$ *bimodule multifusion category* [67, 90], and written ${}_{\mathcal{C}}\mathcal{W}_{\mathcal{D}}$. In other words, the data that describes point-like defects on $\mathcal{W}$, together with exchange processes consistent with the bulk braiding, is a bimodule UmFC.

In the case that the domain wall hosts a single superselection sector of topologically trivial states, $\mathcal{W}$ will be a UFC; this is often assumed in the literature. However, as we will see, two parallel domain walls, viewed as a single domain wall between the outer bulk regions, can decompose into multiple superselection sectors, even if each individual domain wall was indecomposable. Consequently, there are wall excitations in each superselection sector of ground states, as well as point defects between different sectors of the domain wall. The resulting structure is an indecomposable UmFC category, with one simple summand of the tensor unit associated to each superselection sector.

In fact, not all bimodule UmFCs arise as the categories of wall excitations on a topological domain wall. The bimodule UmFCs $\mathcal{W}$ that appear as categories of wall excitations are precisely those that induce Witt equivalences on the UMTCs $\mathcal{C}$ and $\mathcal{D}$ of anyons [85, § 5] [59, § 4], meaning that the braided tensor functor $\mathcal{C} \boxtimes \overline{\mathcal{D}} \to Z(\mathcal{W})$ is a unitary braided equivalence. Thus, an alternative but equivalent description of topological domain walls is to specify a Witt equivalence between $\mathcal{C}$ and $\mathcal{D}$ [42].

Domain walls between anomaly-free topological orders have been thoroughly studied from the perspective of boundaries between Levin-Wen models in [79]. There, it was shown that a topological boundary between the string-net models associated to the fusion categories $\mathcal{X}$ and $\mathcal{Y}$ comes from the data of an indecomposable $\mathcal{X}-\mathcal{Y}$ bimodule category, i.e. a finitely semisimple category $\mathcal{M}$ equipped with a tensor functor $\mathcal{X} \boxtimes \mathcal{Y}^{\mathrm{mp}} \to \mathrm{End}(\mathcal{M})$. It was also shown that pointlike excitations localized on such a boundary form a fusion category, namely the category $\mathrm{End}_{\mathcal{X}-\mathcal{Y}}(\mathcal{M})$ where objects are endofunctors of $\mathcal{M}$ (i.e. of functors from $\mathcal{M}$ to itself) which commute with the $\mathcal{X}$ and $\mathcal{Y}$ actions (which is extra data on such functors), and morphisms are natural transformations. Specifically, this means that pointlike excitations on the domain wall correspond to functors in $\mathrm{End}(\mathcal{M})$. However, these functors must satisfy certain conditions to ensure that they can be coupled consistently to the bulk. The $\mathcal{X}$ and $\mathcal{Y}$ actions are part of the data describing this coupling; the condition of being in $\mathrm{End}_{\mathcal{X}-\mathcal{Y}}(\mathcal{M}) \subseteq \mathrm{End}(\mathcal{M})$ allows an excitation to be exchanged with other domain wall excitations in a manner consistent with the bulk braiding. The tensor product on $\mathrm{End}_{\mathcal{X}-\mathcal{Y}}(\mathcal{M})$, dictating the fusion rules of excitations on the boundary, is just functor composition.

A special case of the construction of [79] recovers the fact that localized excitations in the $\mathcal{X}$ bulk region are described by the UMTC $Z(\mathcal{X})$, since $\mathrm{End}_{\mathcal{X}-\mathcal{X}}(\mathcal{X}) \cong Z(\mathcal{X})$ (where the $\mathcal{X}-\mathcal{X}$ bimodule structure on $\mathcal{X}$ is just $\otimes$ in $\mathcal{X}$). Here, $\mathcal{X}$ is the identity $\mathcal{X}-\mathcal{X}$ bimodule, which describes the trivial domain wall, i.e. no domain wall, in the (2+1)D string-net bulk determined by $\mathcal{X}$. Since $\mathrm{End}_{\mathcal{X}-\mathcal{Y}}(\mathcal{M})$ is canonically a Witt equivalence between $\mathrm{End}_{\mathcal{X}-\mathcal{X}}(\mathcal{X})$ and $\mathrm{End}_{\mathcal{Y}-\mathcal{Y}}(\mathcal{Y})$, and as we have seen $\mathrm{End}_{\mathcal{X}-\mathcal{X}}(\mathcal{X}) \cong Z(\mathcal{X})$ and $\mathrm{End}_{\mathcal{Y}-\mathcal{Y}}(\mathcal{Y}) \cong Z(\mathcal{Y})$, it follows that $\mathrm{End}_{\mathcal{X}-\mathcal{Y}}(\mathcal{M})$ is a Witt equivalence $Z(\mathcal{X}) \to Z(\mathcal{Y})$. As argued in [85, § 5] [59, § 4], a domain wall between (2+1)D topological orders is topological precisely when the category of wall excitations is a Witt equivalence.

We will now show that the picture in the $\mathcal{A}$-enriched case is analogous. Topological domain walls between (2+1)D TOs $\mathcal{X}, \mathcal{Y} \in \mathrm{UFC}^{\mathcal{A}}$ correspond to $\mathcal{A}$-enriched bimodule categories, i.e. an $\mathcal{X}-\mathcal{Y}$ bimodule category $\mathcal{M}$ such that the two actions

$$\mathcal{A} \to Z(\mathcal{X}) \to \mathcal{X} \to \mathrm{End}(\mathcal{M}),$$
$$\overline{\mathcal{A}} \to \overline{Z(\mathcal{Y})} = Z(\mathcal{Y}^{\mathrm{mp}}) \to \mathcal{Y}^{\mathrm{mp}} \to \mathrm{End}(\mathcal{M}),$$

agree on the underlying fusion category of $\mathcal{A}$ (which is the data of a particular natural isomorphism); see [14] or [74, §2.3] for the precise definition.[20] Concretely, given the data of an $\mathcal{A}$-enriched bimodule category, it is straightforward to combine the lattice model in § 2.2 and the lattice model for a domain wall in [79] to produce a lattice model for the (1+1)D boundary separating the topological orders $Z^A(\mathcal{X})$ and $Z^A(\mathcal{Y})$ on the surface of the 3D bulk constructed from $\mathcal{A}$; this shows that the resulting domain wall is topological. In the resulting lattice model, the first two arrows above collectively indicate how an edge label $a \in \mathrm{Irr}(\mathcal{A})$ in the Walker-Wang bulk can be identified with an edge label $x^a \in \mathrm{Irr}(\mathcal{X})$ of the string net constructed from $\mathcal{X}$. The last arrow tells us that a point defect arises when a string labeled by $x^a \in \mathrm{Irr}(\mathcal{X})$ terminates on the domain wall $\mathcal{M}$. The condition that the $\mathcal{A}$-actions on $\mathcal{M}$ agree simply ensures that the association between bulk and boundary string labels is consistent on both sides of the domain wall, such that the resulting defect resides entirely in the (2+1)D $\mathcal{X}$-boundary, and does not extend into the (3+1)D $\mathcal{A}$-bulk.

---

[20]When a (multi)fusion category $\mathcal{Y}$ is $\mathcal{A}$-enriched, then $\mathcal{Y}^{\mathrm{mp}}$ is $\overline{\mathcal{A}}$-enriched and $\overline{Z^{\mathcal{A}}(\mathcal{Y})} \cong Z^{\overline{\mathcal{A}}}(\mathcal{Y}^{\mathrm{mp}})$.

To understand why this data describes topological domain walls, recall that $Z(\mathcal{X}) \cong Z^{\mathcal{A}}(\mathcal{X}) \boxtimes \mathcal{A}$, and the string-net model associated to $\mathcal{X}$ describes a system with decoupled layers of $Z^{\mathcal{A}}(\mathcal{X})$ and $\mathcal{A}$ topological order (see Fig. 5). As discussed in [79], a topological domain wall between string net models for $\mathcal{X}$ and $\mathcal{Y}$ is described by an $\mathcal{X}-\mathcal{Y}$ bimodule $\mathcal{M}$. By Lemma 2.12 below, the condition that $\mathcal{M}$ is $\mathcal{A}$-enriched is equivalent to requiring that for any anyon $a \in \text{Irr}(\mathcal{A})$, $a$ and $\bar{a}$ will annihilate when fused on the domain wall, even when they are brought in from opposite sides. In other words, the domain wall factors as a domain wall connecting the $Z^{\mathcal{A}}(\mathcal{X})$ and $Z^{\mathcal{A}}(\mathcal{Y})$ layers and a domain wall connecting the two $\mathcal{A}$-layers, and the boundary in the $\mathcal{A}$-layer is the trivial $\mathcal{A}-\mathcal{A}$ domain wall, i.e. no domain wall. Thus, the choice of an $\mathcal{A}$-enriched $\mathcal{X}-\mathcal{Y}$ bimodule $\mathcal{M}$ is just the choice of a domain wall between $Z^{\mathcal{A}}(\mathcal{X})$ and $Z^{\mathcal{A}}(\mathcal{Y})$ topological orders.

To turn the domain wall between $Z(\mathcal{X})$ and $Z(\mathcal{Y})$ described above into a domain wall between $Z^{\mathcal{A}}(\mathcal{X})$ and $Z^{\mathcal{A}}(\mathcal{Y})$, we simply attach the $\mathcal{A}$-Walker Wang bulk to the $\mathcal{A}$-layer on both sides of the domain wall, using the prescription of § 2.2. Since the domain wall is trivial in this layer, this can be done without adding any bulk codimension 1 defects. Moreover, any domain wall that is extended into the bulk via a Witt equivalence can be folded into the surface; from our discussion in § 2.3, it follows that such domain walls are topologically equivalent to the ones that we describe here.

We have just explained that topological domain walls admit two equivalent characterizations. The first involves a Witt equivalence (which may be multifusion) between the two bulk topological orders, describing point-like defects localized to the domain wall, and the second involves fixing an $\mathcal{A}$-enriched bimodule between the $\mathcal{A}$-enriched multifusion categories which determine commuting projector models for the bulk topological orders, which is the data required to construct a commuting projector Hamiltonian for the domain wall.

By identifying domain walls between $Z^{\mathcal{A}}(\mathcal{X})$ and $Z^{\mathcal{A}}(\mathcal{Y})$ with a subset of domain walls between $Z(\mathcal{X})$ and $Z(\mathcal{Y})$, we obtain both of these pictures from our construction. We have already described how the data of an $\mathcal{A}$-enriched bimodule category determines a topological domain wall between (2+1)D bulk topological orders with anomaly described by $\mathcal{A}$, and a corresponding commuting projector Hamiltonian. Moreover, we can use the methods of [79] to show that point-like defects on the domain wall are described by the UmFC $\text{End}^{\mathcal{A}}_{\mathcal{X}-\mathcal{Y}}(\mathcal{M})$.

In the remainder of this section, we explore the mathematical relationship between these two descriptions. Specifically, we give two fundamental constructions which relate the Witt equivalence describing wall excitations to the enriched bimodule categories that define our lattice model. Specifically, Construction 2.8 turns the data of an $\mathcal{A}$-enriched bimodule category into the Witt equivalence of wall-excitations. Construction 2.11 shows that one can also go the other way: given the desired category of wall excitations, by including the action functors which describe bringing bulk excitations to the domain wall, one can produce the data necessary for defining a commuting projector lattice model of a (1+1)D domain wall that hosts those wall excitations and actions. Taken together, these constructions show that the perspective of understanding (2+1)D topological order in terms of enriched fusion categories subsumes the perspective of looking solely at UMTCs, because all possible UMTCs of bulk excitations and Witt equivalences of wall excitations arise from enriched fusion categories and enriched bimodules.

In the unenriched case, i.e. string-net models, Witt equivalences $\mathcal{W}$ between $Z(\mathcal{X})$ and $Z(\mathcal{Y})$ correspond to Lagrangian algebras in $Z(\mathcal{X}) \boxtimes \overline{Z(\mathcal{Y})} \cong Z(\mathcal{X} \boxtimes \mathcal{Y}^{\text{mp}})$ (as we recall in § 3.1 below), which in turn correspond to indecomposable $\mathcal{X} \boxtimes \mathcal{Y}^{\text{mp}}$ module categories [41, Def. 3.3] (see Remark 2.10 below). Construction 2.11 generalizes this to the enriched setting, taking in a Witt equivalence between enriched centers and producing an indecomposable enriched bimodule category which would be mapped to that Witt equivalence under Construction 2.8.

**Construction 2.8.** Just as we take the enriched centers $Z^{\mathcal{A}}(\mathcal{X})$ and $Z^{\mathcal{A}}(\mathcal{Y})$ to obtain excitations

in the (2+1)D bulk regions, wall excitations on the domain wall whose commuting projector model is obtained from the $\mathcal{A}$-enriched $\mathcal{X} - \mathcal{Y}$ bimodule category $\mathcal{M}$ are described by the multifusion category

$$\mathrm{End}_{\mathcal{X}-\mathcal{Y}}^{\mathcal{A}}(\mathcal{M}) := \mathrm{End}_{\mathsf{UBFC}}(\mathcal{M}) = \mathrm{End}_{\mathcal{X} \boxdot_{\mathcal{A}} \mathcal{Y}^{\mathrm{mp}}}(\mathcal{M}),$$

where $\mathcal{X} \boxdot_{\mathcal{A}} \mathcal{Y}^{\mathrm{mp}} = (\mathcal{X} \boxtimes \mathcal{Y}^{\mathrm{mp}})_{K_{\mathcal{A}}}$ by (10). Since

$$Z(\mathrm{End}_{\mathcal{X}-\mathcal{Y}}(\mathcal{M})) \cong Z(\mathcal{X} \boxtimes \mathcal{Y}^{\mathrm{mp}}) \cong Z(\mathcal{X}) \boxtimes \overline{Z(\mathcal{Y})} \cong Z^{\mathcal{A}}(\mathcal{X}) \boxtimes \mathcal{A} \boxtimes \overline{\mathcal{A}} \boxtimes \overline{Z^{\mathcal{A}}(\mathcal{Y})},$$

by [42, Thm. 3.20], we have

$$
\begin{aligned}
Z(\mathrm{End}_{\mathcal{X}-\mathcal{Y}}^{\mathcal{A}}(\mathcal{M})) &\cong Z(\mathcal{X} \boxdot_{\mathcal{A}} \mathcal{Y}^{\mathrm{mp}}) \\
&= Z\big((\mathcal{X} \boxtimes \mathcal{Y}^{\mathrm{mp}})_{K_{\mathcal{A}}}\big) \\
&\cong Z(\mathcal{X} \boxtimes \mathcal{Y}^{\mathrm{mp}})_{K_{\mathcal{A}}}^{\mathrm{loc}} \\
&\cong Z^{\mathcal{A}}(\mathcal{X}) \boxtimes \overline{Z^{\mathcal{A}}(\mathcal{Y})}.
\end{aligned}
$$

Thus the bimodule tensor category $\mathrm{End}_{\mathcal{X}-\mathcal{Y}}^{\mathcal{A}}(\mathcal{M})$, which is the category of excitations on the (1+1)D domain wall whose commuting projector model is described by $\mathcal{M}$, is indeed a Witt equivalence between the enriched centers $Z^{\mathcal{A}}(\mathcal{X})$ and $Z^{\mathcal{A}}(\mathcal{Y})$, verifying that the domain wall is topological [59, § 4].

**Example 2.9.** For the trivial $\mathcal{A}$-enriched $\mathcal{X} - \mathcal{X}$ bimodule ${}_{\mathcal{X}}\mathcal{X}_{\mathcal{X}}$, manifestly we have $\mathrm{End}_{\mathcal{X}-\mathcal{X}}^{\mathcal{A}}(\mathcal{X}) = Z^{\mathcal{A}}(\mathcal{X})$, as forgetting the enrichment, $\mathrm{End}_{\mathcal{X}-\mathcal{X}}(\mathcal{X}) = Z(\mathcal{X}) \cong Z^{\mathcal{A}}(\mathcal{X}) \boxtimes \mathcal{A}$. We provide further discussion in § 3.2 below.

**Remark 2.10.** Before presenting Construction 2.11, we recall the correspondence between indecomposable $\mathcal{X}$-module categories $\mathcal{M}$, describing gapped boundaries from $Z(\mathcal{X})$ to the vacuum, and Lagrangian algebras in $Z(\mathcal{X})$ from [41, Def. 3.3]. Given an $\mathcal{X}$-module category, observe that $\mathrm{End}_{\mathcal{X}}(\mathcal{M})$ is a fusion category Morita equivalent to $\mathcal{X}$, and thus $Z(\mathrm{End}_{\mathcal{X}}(\mathcal{M})) \cong Z(\mathcal{X})$. This means there is a Lagrangian algebra $A \in Z(\mathcal{X})$ such that $\mathrm{End}_{\mathcal{X}}(\mathcal{M}) \cong Z(\mathcal{X})_A$. Indeed, $A = \bigoplus_{m \in \mathrm{Irr}(\mathcal{M})} \underline{\mathrm{End}}(m) \in \mathcal{X}$, which lifts to a Lagrangian algebra in $Z(\mathcal{X})$.

Conversely, given a Lagrangian algebra $A \in Z(\mathcal{X})$, the forgetful image of $A$ in $\mathcal{X}$ is a direct sum of Morita equivalent indecomposable algebra objects $A = \bigoplus A_i$. Then $\mathcal{M} = \mathrm{Mod}_{\mathcal{X}}(A_i)$ is an indecomposable $\mathcal{X}$-module category, and choosing a different $A_j$ gives an equivalent $\mathcal{M}$ by Morita equivalence of $A_i$ and $A_j$.

This correspondence is one-to-one between Lagrangian algebras in $Z(\mathcal{X})$ up to isomorphism and $\mathcal{X}$-module categories (i.e. gapped boundaries to the vacuum) up to equivalence. In § 3.1, we will show how this implies that topological boundaries between (2+1)D topological orders all arise from anyon condensation, up to the folding trick.

**Construction 2.11.** Given a Witt-equivalence $\mathcal{W}$ between $Z^{\mathcal{A}}(\mathcal{X})$ and $Z^{\mathcal{A}}(\mathcal{Y})$, i.e., a fusion category $\mathcal{W}$ such that $Z(\mathcal{W}) \cong Z^{\mathcal{A}}(\mathcal{X}) \boxtimes \overline{Z^{\mathcal{A}}(\mathcal{Y})}$, we can promote it to a canonical $\mathcal{A}$-enriched $\mathcal{X} - \mathcal{Y}$ bimodule category $\mathcal{M}$ which recovers $\mathcal{W}$ as $\mathrm{End}_{\mathcal{X}-\mathcal{Y}}^{\mathcal{A}}(\mathcal{M})$.

Indeed, observe that as braided fusion categories,

$$Z(\mathcal{W} \boxtimes \mathcal{A}) \cong Z(\mathcal{W}) \boxtimes Z(\mathcal{A}) \cong Z^{\mathcal{A}}(\mathcal{X}) \boxtimes \overline{Z^{\mathcal{A}}(\mathcal{Y})} \boxtimes \mathcal{A} \boxtimes \overline{\mathcal{A}} \cong Z(\mathcal{X}) \boxtimes \overline{Z(\mathcal{Y})} \cong Z(\mathcal{X} \boxtimes \mathcal{Y}^{\mathrm{mp}}).$$

Let $L$ in $Z^{\mathcal{A}}(\mathcal{X}) \boxtimes \overline{Z^{\mathcal{A}}(\mathcal{Y})}$ be the Lagrangian algebra corresponding to $\mathcal{W}$, and let $K_{\mathcal{A}} \in \mathcal{A} \boxtimes \overline{\mathcal{A}}$ be the canonical Lagrangian from (9). Then $L \boxtimes K_{\mathcal{A}}$ gives a Lagrangian algebra in $Z(\mathcal{X} \boxtimes \mathcal{Y}^{\mathrm{mp}})$, corresponding to a $\mathcal{X} \boxtimes \mathcal{Y}^{\mathrm{mp}}$-module category $\mathcal{M}$ under the correspondence [41, Def. 3.3] (see

Remark 2.10). Unfolding, the $\mathcal{X} - \mathcal{Y}$ bimodule category $\mathcal{M}$ has an $\mathcal{A}$-enrichment by Lemma 2.12 below. Also by construction, $\mathrm{End}^{\mathcal{A}}_{\mathcal{X}-\mathcal{Y}}(\mathcal{M}) \cong (Z^{\mathcal{A}}(\mathcal{X}) \boxtimes \overline{Z^{\mathcal{A}}(\mathcal{Y})})_L$. Since both $\mathrm{End}^{\mathcal{A}}_{\mathcal{X}-\mathcal{Y}}(\mathcal{M})$ and $\mathcal{W}$ correspond to the Lagrangian algebra $L$, they are equivalent as $Z^{\mathcal{A}}(\mathcal{X}) - Z^{\mathcal{A}}(\mathcal{Y})$ bimodule tensor categories.

Physically, we should think of this construction as follows. The Lagrangian algebra $L \boxtimes K_{\mathcal{A}}$ specifies a domain wall between $\mathcal{X}$ and $\mathcal{Y}$ that factors into a domain wall between $Z^{\mathcal{A}}(X)$ and $Z^{\mathcal{A}}(Y)$ in one layer, and a trivial domain wall in the $\overline{\mathcal{A}}$ layer. This follows from the fact that $L$ is transparent to anyons in $\mathcal{A}$, and that $K_{\mathcal{A}}$ is the Lagrangian algebra in $\mathcal{A} \boxtimes \overline{\mathcal{A}}$ corresponding to the trivial domain wall; see Remark 3.5. Lemma 2.12 does the mathematical work of verifying that an $\mathcal{A}$-enriched bimodule is the same thing as one which is transparent to anyons from $\mathcal{A}$, and is therefore the correct data to describe the resulting domain wall.

**Lemma 2.12.** *Suppose $\mathcal{X}, \mathcal{Y}$ are $\mathcal{A}$-enriched fusion categories and $_{\mathcal{X}}\mathcal{M}_{\mathcal{Y}}$ is an $\mathcal{X} - \mathcal{Y}$ bimodule category. Let $L$ be the Lagrangian algebra in*

$$Z(\mathcal{X} \boxtimes \mathcal{Y}^{\mathrm{mp}}) \cong Z^{\mathcal{A}}(\mathcal{X}) \boxtimes \overline{Z^{\mathcal{A}}(\mathcal{Y})} \boxtimes \mathcal{A} \boxtimes \overline{\mathcal{A}},$$

*corresponding to $\mathcal{M}$. The following are equivalent:*

*(1) There is a compatible $\mathcal{A}$-enrichment on $\mathcal{M}$*

*(2) There is an algebra homomorphism $K_{\mathcal{A}} \to L$, where $K_{\mathcal{A}}$ is in the copy of $\mathcal{A} \boxtimes \overline{\mathcal{A}} \subset Z(\mathcal{X} \boxtimes \mathcal{Y}^{\mathrm{mp}})$.*

*(3) We can factorize $L = L' \boxtimes K_{\mathcal{A}}$ for some Lagrangian algebra $L' \in Z^{\mathcal{A}}(\mathcal{X}) \boxtimes \overline{Z^{\mathcal{A}}(\mathcal{Y})}$.*

*Proof.* By definition, $\mathcal{A}$-enrichments on $_{\mathcal{X}}\mathcal{M}_{\mathcal{Y}}$ correspond to monoidal natural isomorphisms

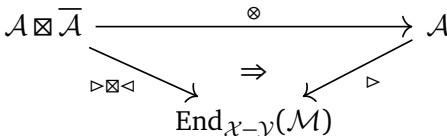

where $\triangleright \boxtimes \triangleleft : \mathcal{A} \boxtimes \overline{\mathcal{A}} \to \mathrm{End}_{\mathcal{X}-\mathcal{Y}}(\mathcal{M})$ is the free module functor for $L \cap \mathcal{A} \boxtimes \overline{\mathcal{A}}$, and $\otimes : \mathcal{A} \boxtimes \overline{\mathcal{A}} \to \mathcal{A}$ is the free module functor for $K_{\mathcal{A}}$.[21] Since $K_{\mathcal{A}}$ is Lagrangian, such a factorization will exist if and only if $K_{\mathcal{A}}$ is isomorphic as an algebra to $L \cap \mathcal{A} \boxtimes \overline{\mathcal{A}}$.

Clearly (3) implies (2). To show that (2) implies $L = L' \boxtimes K_{\mathcal{A}}$ for some Lagrangian algebra $L' \in Z^{\mathcal{A}}(\mathcal{X}) \boxtimes \overline{Z^{\mathcal{A}}(\mathcal{Y})}$, we simply observe that $L$ is a condensable algebra with $1 \boxtimes K_{\mathcal{A}}$ as a subalgebra, and therefore, possible choices of $L$ are in bijection with Lagrangian algebras in

$$(Z(\mathcal{X}) \boxtimes \overline{Z(\mathcal{Y})})^{\mathrm{loc}}_{K_{\mathcal{A}}} \cong Z^{\mathcal{A}}(\mathcal{X}) \boxtimes \overline{Z^{\mathcal{A}}(\mathcal{Y})}.$$

Local $K_{\mathcal{A}}$ modules are all of the form $c \boxtimes K_{\mathcal{A}}$ for $c \in Z^{\mathcal{A}}(\mathcal{X}) \boxtimes \overline{Z^{\mathcal{A}}(\mathcal{Y})}$, so the Lagrangian algebra $L' \in Z^{\mathcal{A}}(\mathcal{X}) \boxtimes \overline{Z^{\mathcal{A}}(\mathcal{Y})}$ corresponds to the algebra $L' \boxtimes K_{\mathcal{A}} \supseteq 1 \boxtimes K_{\mathcal{A}}$, as claimed. $\square$

In the preceding constructions, we saw that an $\mathcal{A}$-enriched bimodule category $_{\mathcal{X}}\mathcal{M}_{\mathcal{Y}}$ between $\mathcal{A}$-enriched fusion categories $\mathcal{X}, \mathcal{Y} \in \mathsf{UFC}^{\mathcal{A}}$ determines a Witt equivalence $\mathrm{End}^{\mathcal{A}}_{\mathcal{X}-\mathcal{Y}}(\mathcal{M}) : Z^{\mathcal{A}}(\mathcal{X}) \to Z^{\mathcal{A}}(\mathcal{Y})$, and that all Witt equivalences between enriched centers arise in this way. By Example 2.3, any UMTC $\mathcal{A}$ arises as an enriched center, so all Witt

---

[21]For an algebra $A$ in $\mathcal{C}$, the free module functor $\mathcal{C} \to \mathcal{C}_A$ is given by $c \mapsto cA_A$, where the module structure on $cA_A$ comes from the multiplication $AA \to A$. By [20], any surjective (dominant) tensor functor between fusion categories $\mathcal{S} \to \mathcal{T}$ is equivalent to the free module functor associated with some commutative algebra $A \in Z(\mathcal{S})$, including the functors $\triangleright \boxtimes \triangleleft$ and $\otimes$ considered here in the lemma. The interpretation of the free module functor in the context of anyon condensation is reviewed in detail in Example 3.2.

equivalences between UMTCs can be obtained from Construction 2.8. In other words, the perspective of Remark 2.4 that (2+1)D topological orders should be described by enriched fusion categories, with domain walls described by enriched bimodule categories, is in harmony with the expectations that bulk excitations are described by UMTCs, wall excitations are described by Witt equivalences between them, and that categories of excitations completely determine the topological order of the bulks and domain walls.

We remark that from the usual viewpoint, if a topological order is described by a UMTC $\mathcal{C}$, then the topological domain walls from $\mathcal{C}$ to itself are described by the fusion 2-category $\mathsf{Mod}(\mathcal{C})$. Indeed, given such a domain wall $\mathcal{M}$, we take the semi-simple category of so-called twist defects, i.e. point defects between the trivial domain wall and $\mathcal{M}$. Bringing in excitations from the bulk makes this into a $\mathcal{C} - \mathcal{C}$ bimodule category, but moving excitations around the bottom of a twist defect allows us to equip this category with a $\overline{\mathcal{C}}$-enriched structure. The fusion 2-category of $\overline{\mathcal{C}}$-enriched $\mathcal{C} - \mathcal{C}$ bimodules is equivalent to $\mathsf{Mod}(\mathcal{C})$ [43]. By [71] this yields an equivalence between the fusion 2-category of topological domain walls from $\mathcal{C}$ to itself and $\mathsf{Mod}(\mathcal{C})$.

On the other hand, using UFC$^{\mathcal{A}}$ as a 3-category of topological orders leads us to expect that topological domain walls from $Z^{\mathcal{A}}(\mathcal{X})$ topological order to itself are described by UFC$^{\mathcal{A}}(\mathcal{X} \to \mathcal{X})$, which is the fusion 2-category $\mathsf{Bim}^{\mathcal{A}}(\mathcal{X})$. To demonstrate the consistency of our framework with the usual point of view, we have the following mathematical result.

**Proposition 2.13.** *There is an equivalence of fusion 2-categories* $\mathsf{Bim}^{\mathcal{A}}(\mathcal{X}) \cong \mathsf{Mod}(Z^{\mathcal{A}}(\mathcal{X}))$.

*Proof.* Observe that $\mathsf{Bim}^{\mathcal{A}}(\mathcal{X}) = \mathsf{End}_{\mathsf{UBFC}}(_{\mathcal{A}}\mathcal{X}_{\mathsf{Hilb}})$. Now consider $\mathcal{X}^{\mathrm{mp}}$ as a $\overline{Z^{\mathcal{A}}(\mathcal{X})} - \mathcal{A}$ Witt-equivalence, i.e., invertible 1-morphism in UBFC. Composing these two 1-morphisms, we get $\mathcal{X}^{\mathrm{mp}} \boxempty_{\mathcal{A}} \mathcal{X}$ as a 1-morphsim in $\mathsf{UBFC}(\overline{Z^{\mathcal{A}}(\mathcal{X})} \to \mathsf{Hilb})$. Since $\mathcal{X}^{\mathrm{mp}} \boxempty_{\mathcal{A}} \mathcal{X}$ is Morita equivalent to $Z^{\mathcal{A}}(\mathcal{X})$ via the $\mathcal{A}$-enriched bimodule $\mathcal{X}$, we see

$$\mathsf{End}_{\mathsf{UBFC}}(_{\overline{Z^{\mathcal{A}}(\mathcal{X})}}Z^{\mathcal{A}}(\mathcal{X})_{\mathsf{Hilb}}) \cong \mathsf{End}_{\mathsf{UBFC}}(_{\mathcal{A}}\mathcal{X}_{\mathsf{Hilb}}).$$

The left hand side is exactly $\mathsf{Mod}(Z^{\mathcal{A}}(\mathcal{X}))$ by [43]. A graphical representation of this map in a 2D projection of the 4D graphical calculus of UBFC[22] is as follows:

$$\mathsf{Bim}^{\mathcal{A}}(\mathcal{X}) \ni \; \begin{array}{c} \mathcal{X} \\[2pt] \boxed{\mathcal{A}} \; \bullet \, \mathcal{M} \\[2pt] \mathcal{X} \end{array} \; \mapsto \; \begin{array}{c} Z^{\mathcal{A}}(\mathcal{X}) \\[2pt] \mathcal{X} \\[2pt] \boxed{\mathcal{X}^{\mathrm{mp}} \; \mathcal{A} \; } \bullet \, \mathcal{M} \\[2pt] \overline{Z^{\mathcal{A}}(\mathcal{X})} \quad \mathcal{X} \\[2pt] Z^{\mathcal{A}}(\mathcal{X}) \end{array} \; \in \mathsf{Mod}(Z^{\mathcal{A}}(\mathcal{X})). \qquad \square$$

## 3 Decomposing composite domain walls

Having outlined our perspective on topological boundaries between (2+1)D topological orders, we now turn to the question of central interest: namely, what can happen when we compose two or more domain walls by stacking them? This question is mathematically important, since any domain wall can be obtained by such a composition. Moreover, in the context of generalized symmetries (see [32,107] for reviews), there has recently been considerable interest in the fusion rules of non-invertible symmetry defects (see, for example, [8,11,31,54,82] and references therein), which are closely related to the composition of topological domain walls studied here. The mathematical structure underpinning our analysis of composition also

---

[22]Hom 2-categories in UBFC are 2-categories themselves, and have a well-defined 2D graphical calculus.

has connections to duality transformations of 1-dimensional lattice models, which have also recently enjoyed renewed interest [47, 96, 102].

In § 3.1, we review how domain walls are related to condensable and Lagrangian algebras in UMTCs, as well as how this relation can be used to describe any domain wall as a composition of elementary domain wall types. In § 3.2, we introduce the mathematical operations corresponding to composing domain walls, i.e. treating parallel domain walls as a single domain wall. Finally, in § 3.3, we explain the mathematics of decomposing parallel domain walls into irreducible summands, including the calculation of the GSD which occurs given a composite of domain walls.

## 3.1 Elementary topological domain walls

To set the stage for the remainder of this work, we now review the results of [44], who describe how every (indecomposable) topological domain wall can be viewed as a composite of several *elementary domain walls* (see Fig. 8). There are two classes of elementary domain walls: invertible domain walls (Example 3.1), which implement a braided tensor equivalence between the UMTCs of excitations on each side, and domain walls where an algebra is condensed on one side (Example 3.2).

**Example 3.1.** Suppose $\mathcal{X}$ is an $\mathcal{A}$-enriched fusion category. Invertible defects between regions with $Z^{\mathcal{A}}(\mathcal{X})$ bulk topological order correspond to invertible enriched $\mathcal{X} - \mathcal{X}$ bimodule categories (i.e. Morita equivalences), which correspond to braided autoequivalences $\Phi \in \text{Aut}^{\text{br}}_{\otimes}(Z^{\mathcal{A}}(\mathcal{X}))$ by [50, 74]. Anyons which cross the boundary are permuted by the symmetry $\Phi$, i.e. an anyon $c \in Z^{\mathcal{A}}(\mathcal{X})$ crosses the domain wall to become $\Phi(c)$. Invertible defects have been discussed extensively in the literature [2, 7, 13, 17, 22, 23, 30, 33, 95, 126]. Locally, such invertible defect lines are simply a benign relabelling of anyon types on one side of the defect relative to the other. However, they can have striking physical consequences, such as altering the ground state degeneracy, when the defect lines terminate.

Another important class of domain walls comes from condensable algebras, a.k.a. unitarily separable étale algebras $A$ in a UMTC $\mathcal{C}$. We include a review of condensable algebras in Appendix § B below. As the name suggests, a condensable algebra $A$ is a collection of anyons which can be condensed, leading to a distinct phase with a different topological order [24, 27, 29, 85]. When the anyons before condensation are described by the UMTC $\mathcal{C}$, the condensed phase is described by the UMTC $\mathcal{C}^{\text{loc}}_A$, which we will describe presently (see Appendix § B for details). By Constructions 2.8 and 2.11, the discussion of condensation in [85] applies equally well, regardless of whether $\mathcal{C}$ is a Drinfeld center or an enriched center.

**Example 3.2.** The article [85] (see also [25]) explains how anyon condensation can be carried out on one side of a domain wall, giving a topological boundary between the uncondensed phase with topological order $\mathcal{C}$ and the condensed phase with topological order $\mathcal{C}^{\text{loc}}_A$, the category of local right $A$-modules in $\mathcal{C}$. Excitations on the boundary are described by $\mathcal{C}_A$, which is the fusion category obtained from $\mathcal{C}$ by condensing $A$, but not demanding locality, which physically corresponds to requiring that the excitation braid trivially with $A$.

The data necessary to perform anyon condensation is a condensable algebra $A$, which is a collection of anyons (described mathematically as a direct sum) in $\mathcal{C}$ together with a collection of fusion channels satisfying consistency conditions [85]. These conditions guarantee that $A$-lines can be treated as equivalent to vacuum lines. Similarly, the data of an excitation on the domain wall created by condensation is a direct sum of anyons $M \in \mathcal{C}$ together with a choice of how the condensate $A$ can be absorbed by $M$, which is mathematically the choice of an $A$-action

morphism in $\mathcal{C}(MA \to M)$.[23] This is the mathematical data of an $A$-module, i.e. an object $M_A$ in $\mathcal{C}_A$. Here, the letter $M$ refers to the direct sum of anyons of the uncondensed phase, and the subscript $\cdot_A$ reminds us of the $A$-action. We emphasize that, although we usually denote simple objects by lowercase letters, we will usually denote simple objects in $\mathcal{C}_A$ by uppercase letters, because the underlying object in $\mathcal{C}$ may not be simple. Physically, this action allows $M$ to absorb excitations from the condensate $A$ through fusion as if they were the vacuum.

The UMTC $\mathcal{C}_A^{\mathrm{loc}}$ which describes bulk excitations in the condensed phase is a subcategory of $\mathcal{C}_A$, consisting of those particles which braid trivially with the condensate $A$. Thus, we typically denote anyons in the condensed phase with the same notation as wall excitations, although we add the superscript $M_A^\circ$ to point out that an object lives in $\mathcal{C}_A^{\mathrm{loc}}$.

Bringing an anyon $M_A^\circ \in \mathrm{Irr}(\mathcal{C}_A^{\mathrm{loc}})$ out of the condensate corresponds to applying the forgetful functor $\mathcal{C}_A^{\mathrm{loc}} \to \mathcal{C}_A \to \mathcal{C} : M_A^\circ \mapsto M_A \mapsto M$; this functor forgets the data of the $A$-action on $M_A$, leaving only the underlying direct sum of anyons $M$. The resulting bulk excitation in the uncondensed region can in general be viewed as a direct sum of different anyon types in $\mathcal{C}$. We provide further explanation of $\mathcal{C}_A$ and $\mathcal{C}_A^{\mathrm{loc}}$, including their physical interpretation, in Appendix B.

Now suppose that $\mathcal{C} = Z(\mathcal{X})$, where $\mathcal{X} \in \mathsf{UFC}$. As we show in Appendix B, the condensed phase $\mathcal{C}_A^{\mathrm{loc}}$ is then the center of the fusion category $\mathcal{X}_A$ of right $A$-modules,[24] $Z(\mathcal{X}_A) \cong Z(\mathcal{X})_A^{\mathrm{loc}}$. The fusion category $\mathcal{X}_A$ can be obtained from $\mathcal{X}$ in the following way. Objects of $\mathcal{X}_A$ are objects $Y$ of $\mathcal{X}$ equipped with a module action $YA \to Y$ satisfying associativity and unitality conditions, which we depict in the diagrammatic calculus as a trivalent vertex. The tensor product on $\mathcal{X}_A$ is the relative tensor product $(Y_A, Z_A) \mapsto Y_A \otimes_A Z_A$ over $A$, which is the image of the following projection.

$$\text{} \in \mathcal{X}(YZ \to YZ).$$

The $Y$ and $Z$ strings in the diagram above represent worldlines of excitations $Y$ and $Z$ confined to the domain wall at the boundary of the region where $A$ is condensed. (We use capital letters for $Y$, and $Z$ because they in general they correspond to direct sums of bulk anyons.) The unlabelled black string is the worldline of the condensable algebra $A \in Z(\mathcal{X})$. The orange and green trivalent vertices describe processes where the anyons $A$ are brought to this gapped boundary, and fused with a $Y$ or $Z$ boundary excitation.

There are several ways to interpret the category $\mathcal{X}_A$. First, as we have just alluded to, objects in $\mathcal{X}_A$ correspond to excitations on the gapped boundary between $Z(\mathcal{X}_A)$ and $Z(\mathcal{X})_A^{\mathrm{loc}}$. Second, because $Z(\mathcal{X}_A) \cong Z(\mathcal{X})_A^{\mathrm{loc}}$, $\mathcal{X}_A$ fixes the data for a commuting projector model for the condensed region. Finally, $\mathcal{X}_A$ can be viewed as a $\mathcal{X} - \mathcal{X}_A$ bimodule category, describing the states of the resulting string-net near the domain wall. The action of $\mathcal{X}_A$ on $\mathcal{X}_A$ is just the tensor product $\otimes_A$, while the action of $\mathcal{X}$ is the free module functor $\mathcal{X} \to \mathcal{X}_A : x \mapsto xA_A$.

To see how this construction generalizes to the enriched case, let $\mathcal{X}$ be an $\mathcal{A}$-enriched fusion category. If $A \in \mathcal{C} := Z^{\mathcal{A}}(\mathcal{X})$ is a condensable algebra, then $\mathcal{X}_A$ is again an $\mathcal{A}$-enriched fusion category,[25] describing the topological order to the right of the domain wall. Since the free module functor $\mathcal{X} \to \mathcal{X}_A$ is monoidal (i.e. it respects the tensor product), we can also view $\mathcal{X}_A$ as an $\mathcal{A}$-enriched $\mathcal{X} - \mathcal{X}_A$ bimodule. Construction 2.8 then shows how to obtain the

---

[23]One can view $M$ as another collection of anyons and consistent fusion channels where the fusion channels have one input leg from $M$ and one leg from $A$, and the output leg is again in $M$. The consistency conditions are entirely similar to those of a condensable algebra.

[24]Here we suppress some of the data associated with $\mathcal{X}_A$; for an explicit discussion of the full data see e.g. [56,57]

[25]By [42, Thm. 3.20], $Z(\mathcal{X}_A) \cong Z(\mathcal{X})_A^{\mathrm{loc}}$. Since $Z(\mathcal{X}) \cong Z^{\mathcal{A}}(\mathcal{X}) \boxtimes \mathcal{A}$ and $A \in Z^{\mathcal{A}}(\mathcal{X})$, we have $Z(\mathcal{X})_A^{\mathrm{loc}} \cong Z^{\mathcal{A}}(\mathcal{X})_A^{\mathrm{loc}} \boxtimes \mathcal{A}$. Thus, $\mathcal{X}_A$ is $\mathcal{A}$-enriched, with $Z^{\mathcal{A}}(\mathcal{X}_A) \cong Z^{\mathcal{A}}(\mathcal{X})_A^{\mathrm{loc}}$.

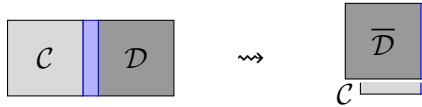

Figure 7: By folding, a topological boundary between $\mathcal{C}$ and $\mathcal{D}$ is equivalent to a topological boundary between $\mathcal{C} \boxtimes \overline{\mathcal{D}}$ and the vacuum. Here, gray indicates regions of the phases $\mathcal{C}$ and $\mathcal{D}$ (or $\overline{\mathcal{D}}$), and light blue indicates that the line defect can be thickened to a region with an intermediate topological order.

Witt equivalence $\mathrm{End}^{\mathcal{A}}_{\mathcal{X}-\mathcal{X}_A}(\mathcal{X}_A) \cong Z^{\mathcal{A}}(\mathcal{X})_A$ describing point-like excitations on the domain wall from this enriched bimodule.

We sometimes wish to condense the algebra $A$ to the left of the domain wall, instead of to the right. In this case, we use the similarly defined $\mathcal{A}$-enriched fusion category $_A\mathcal{X}$ of left $A$-modules in $\mathcal{X}$ for the condensed bulk region, and the domain wall is given by the $\mathcal{A}$-enriched $_A\mathcal{X} - \mathcal{X}$ bimodule category $_A\mathcal{X}$. We do so because the fusion category $\mathcal{X}$ most obviously acts on the opposite side from where $A$ acts. However, the fusion categories $\mathcal{X}_A$ and $_A\mathcal{X}$ are canonically monoidally equivalent: given a right $A$-module $M_A$, we obtain a left action of $A$ on $M$ by composing the right action with the half-braiding on $A$. Similarly, $Z^{\mathcal{A}}(_A\mathcal{X}) \cong {}^{\mathrm{loc}}_A Z^{\mathcal{A}}(\mathcal{X}) \cong Z^{\mathcal{A}}(\mathcal{X})^{\mathrm{loc}}_A$, with the canonical braided monoidal equivalence $Z^{\mathcal{A}}(\mathcal{X})_A \to {}_A Z^{\mathcal{A}}(\mathcal{X})$ preserving the subcategories of local modules. Consequently, we will sometimes tacitly identify $_A\mathcal{X}$ and $\mathcal{X}_A$, e.g. in (38).

**Example 3.3.** Condensing a condensable algebra on one side of a domain wall as in Example 3.2 results in a topological boundary to vacuum precisely when the algebra is Lagrangian [85, Rem. 5.4] [97]. If $A$ is a condensable algebra in a UMTC $\mathcal{C}$, the Drinfeld center of $\mathcal{C}_A$ satisfies

$$Z(\mathcal{C}_A) \cong \mathcal{C} \boxtimes \overline{\mathcal{C}^{\mathrm{loc}}_A}, \tag{11}$$

by [42, Cor. 3.30]. (Recall that $\overline{\mathcal{D}}$ is the same unitary fusion category as $\mathcal{D}$, but equipped with the reverse braiding.) Hence $A$ is a Lagrangian algebra if and only if $\mathcal{C}^{\mathrm{loc}}_A \cong \mathrm{Hilb}$. Eq. (11) implies that this can occur if and only if $\mathcal{C}$ is the Drinfeld center of some unitary fusion category.

A priori, Example 3.3 is a special case of Example 3.2. On the other hand, by the so-called folding trick (see Fig. 7), any gapped boundaries between the topological orders $\mathcal{C}$ and $\mathcal{D}$ are equivalent to gapped boundaries between $\mathcal{C} \boxtimes \overline{\mathcal{D}}$ and the vacuum. Consequently, topological domain walls between $\mathcal{C}$ and $\mathcal{D}$ correspond to Lagrangian algebras in $\mathcal{C} \boxtimes \overline{\mathcal{D}}$, which correspond to fusion categories $\mathcal{X}$ with a choice of equivalence $Z(\mathcal{X}) \cong \mathcal{C} \boxtimes \overline{\mathcal{D}}$ by (11).

**Remark 3.4.** In the $\mathcal{A}$-enriched setting, when performing the folding trick in UBFC, we have an $\mathcal{A}$-region between the two sheets, which corresponds to taking a relative Deligne product over $\mathcal{A}$. That is, gapped $\mathcal{A}$-enriched domain walls between $\mathcal{X}$ and $\mathcal{Y}$ correspond to gapped boundaries for the ordinary Hilb-enriched multifusion category $\mathcal{X} \boxdot_{\mathcal{A}} \mathcal{Y}^{\mathrm{mp}}$.

By [44, Thm. 3.6], Lagrangian algebras $L = L(A, B, \Phi) \in \mathcal{C} \boxtimes \overline{\mathcal{D}}$ are determined by:

- condensable algebras $A \in \mathcal{C}$ and $B \in \overline{\mathcal{D}}$, where

$$A \boxtimes 1 := L \cap \mathcal{C} \boxtimes 1_{\overline{\mathcal{D}}} \quad \text{and} \quad 1 \boxtimes B := L \cap 1_{\mathcal{C}} \boxtimes \overline{\mathcal{D}},$$

and

- a unitary braided equivalence $\Phi : \mathcal{C}^{\mathrm{loc}}_A \to \mathcal{D}^{\mathrm{loc}}_B$.

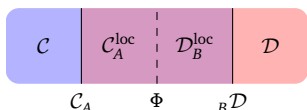

Figure 8: Every topological boundary between topological orders $\mathcal{C}, \mathcal{D}$ can be obtained by juxtaposing condensation boundaries and invertible boundaries cf. [44, §3].

To understand this, consider first when we have an equivalence $\Phi : \mathcal{C} \to \mathcal{D}$. In this case, there is a canonical Lagrangian algebra

$$L \cong \bigoplus_{X \in \mathrm{Irr}(\mathcal{C})} X \boxtimes \overline{\Phi(X)}. \tag{12}$$

In general, $\mathcal{C}$ and $\mathcal{D}$ are not equivalent, but if a Lagrangian algebra exists in $\mathcal{C} \boxtimes \overline{\mathcal{D}}$, we can find algebras $A \in \mathcal{C}$ and $B \in \mathcal{D}$ that can be condensed such that there is an equivalence $\mathcal{C}_A^{\mathrm{loc}} \to \mathcal{D}_B^{\mathrm{loc}}$. We then get a corresponding Lagrangian algebra in

$$(\mathcal{C} \boxtimes \overline{\mathcal{D}})_{A \boxtimes B}^{\mathrm{loc}} \cong \mathcal{C}_A^{\mathrm{loc}} \boxtimes \overline{\mathcal{D}_B^{\mathrm{loc}}},$$

which is necessarily of the form (12).

To illustrate the meaning of $L(A, B, \Phi)$, we describe several special cases. In the case where $A = 1_{\mathcal{C}}$ and $B = 1_{\mathcal{D}}$, the Lagrangian algebra must be $L(1, 1, \Phi)$ for some equivalence $\Phi \in \mathsf{Fun}_{\otimes}^{\mathrm{br}}(\mathcal{C} \to \mathcal{D})$, and we obtain an invertible domain wall, as in Example 3.1. This wall simply applies the relabeling $\Phi$ to anyons crossing the wall, and when $\mathcal{C} = \mathcal{D}$, can be thought of as applying a symmetry action [7]. The inverse domain wall corresponds to $L(1_{\mathcal{D}}, 1_{\mathcal{C}}, \Phi^{-1})$.

The condensation boundaries of Example 3.2 correspond to Lagrangian algebras of the form $L(A, 1_{\mathcal{C}_A^{\mathrm{loc}}}, \mathrm{id}_{\mathcal{C}_A^{\mathrm{loc}}})$ or $L(1_{\mathcal{C}_A^{\mathrm{loc}}}, A, \mathrm{id}_{\mathcal{C}_A^{\mathrm{loc}}})$, depending on which side of the wall $A$ is condensed on. These walls are thoroughly analyzed in [85].

**Remark 3.5.** We note that the trivial $\mathcal{C} - \mathcal{C}$ topological boundary corresponds to the canonical Lagrangian algebra of $\mathcal{C}$ given by $K_{\mathcal{C}} = L(1, 1, \mathrm{id}_{\mathcal{C}}) \in \mathcal{C} \boxtimes \overline{\mathcal{C}}$ from (9). Physically, condensing $c \boxtimes \bar{c}$ means that a $c$ particle from the left $\mathcal{C}$-bulk and a $\bar{c}$ particle from the right $\mathcal{C}$-bulk annihilate on the boundary, or equivalently, that a $c$ particle can pass freely through the wall. Thus, the domain wall is completely transparent to all anyons in $\mathcal{C}$, and must be trivial.

More generally, the Lagrangian algebra $L(A, A, \mathrm{id}_{\mathcal{C}_A^{\mathrm{loc}}})$ contains $(A \boxtimes A) \in \mathcal{C} \boxtimes \overline{\mathcal{C}}$ as a subalgebra, and hence corresponds to a Lagrangian algebra in $\mathcal{C}_A^{\mathrm{loc}} \boxtimes \overline{\mathcal{C}_A^{\mathrm{loc}}}$; this Lagrangian algebra is precisely $K_{\mathcal{C}_A^{\mathrm{loc}}}$.

**Remark 3.6.** An isomorphism $\psi : A \to A'$ between condensable algebras induces a braided tensor equivalence $\widetilde{\psi} : \mathcal{C}_{A'}^{\mathrm{loc}} \to \mathcal{C}_A^{\mathrm{loc}}$. Thus, the Lagrangian algebra $L(A, B, \Phi)$ depends on the isomorphism classes of $A$ and $B$, as well as on the choice of $\Phi$ up to natural isomorphism and composition with elements of $\mathrm{Aut}(A)$ and $\mathrm{Aut}(B)$.

The physical content of [44, Thm. 3.6] is that by *unfolding* the classification, *every* topological domain wall between $(2+1)$D topological orders can be obtained by juxtaposing walls of the two types in Examples 3.1 and 3.2, as in Figure 8.[26] For this reason, we refer to the domain walls described in Examples 3.1 and 3.2 as elementary domain walls.

A dictionary between the mathematical formalism and the physical interpretation for domain walls, including additional details which we will discuss below, appeared in Figure 4.

---

[26]The article [85] gives another method to decompose topological domain walls into two condensations, but in the reverse order of Figure 8.

## 3.2 Composing domain walls

In this section, we review the mathematical operation on bimodule categories for fusion categories which corresponds to composition of domain walls between anomaly-free topological orders. We then argue that the concepts generalize in a straight forward way to the $\mathcal{A}$-enriched case.

**Notation 3.7.** Given UFCs $\mathcal{X}, \mathcal{Y}, \mathcal{Z}$, an $\mathcal{X} - \mathcal{Y}$ bimodule category $\mathcal{M}$, and a $\mathcal{Y} - \mathcal{Z}$ bimodule category $\mathcal{N}$, we can treat parallel domain walls defined by $\mathcal{M}$ and $\mathcal{N}$, as in (13) below, as a single domain wall between the bulk regions determined by $\mathcal{X}$ and $\mathcal{Z}$. As described in [79, § 6], the $\mathcal{X} - \mathcal{Z}$ bimodule category which determines the composite string-net Hilbert space associated with the composite domain wall is the relative Deligne product $\mathcal{M} \boxtimes_{\mathcal{Y}} \mathcal{N}$ [9, 50]. This bimodule category is defined as

$$\boxed{\begin{array}{c|c|c} \mathcal{X} & \mathcal{Y} & \mathcal{Z} \\ \hline & & \end{array}}_{\mathcal{M} \quad \mathcal{N}} \cong {}_{\mathcal{X}}\mathcal{M} \boxtimes_{\mathcal{Y}} \mathcal{N}_{\mathcal{Z}} \cong (\mathcal{M} \boxtimes \mathcal{N})_{S_{\mathcal{Y}}} . \tag{13}$$

The algebra $S_{\mathcal{Y}} \in \mathcal{Y}^{\mathrm{mp}} \boxtimes \mathcal{Y}$ is given by

$$S_{\mathcal{Y}} \cong \bigoplus_{y \in \mathrm{Irr}(\mathcal{Y})} \overline{y} \boxtimes y ,$$

with the multiplication

$$\bigoplus_{x,y,z} \sum_{\gamma \in B^z_{x,y}} \overline{\gamma}^\dagger \boxtimes \gamma : S_{\mathcal{Y}} \otimes S_{\mathcal{Y}} \to S_{\mathcal{Y}} ,$$

where $x, y$ run over $\mathrm{Irr}(\mathcal{Y})$, and $B^z_{x,y}$ is an orthonormal basis of $\mathcal{Y}(xy \to z)$, i.e. a set of fusion channels $xy \to z$. One can check that this multiplication is basis independent. In fact, $\boxtimes_\bullet$ is the definition of 1-composition for 1-morphisms in UFC. Note also that ${}_{\mathcal{X}}\mathcal{M} \boxtimes_{\mathcal{Y}} \mathcal{N}_{\mathcal{Z}}$ is defined to be the category of $S_{\mathcal{Y}}$-modules in $\mathcal{M} \boxtimes \mathcal{N}$, which is itself a module category for the tensor category $\overline{\mathcal{Y}} \boxtimes \mathcal{Y}$ where $S_{\mathcal{Y}}$ lives. This notion is discussed in Remark B.1.

In the diagram in equation (13), as in many diagrams throughout the remainder of this paper, the labels $\mathcal{X}, \mathcal{Y}, \mathcal{Z}$ of spatial regions denote ($\mathcal{A}$-enriched) UFCs which determine a commuting projector model, rather than the corresponding UMTCs of local excitations. Such diagrams naturally live in the graphical calculus of UFC. However, they can also be interpreted physically as representing 2-dimensional spatial regions labelled with the categories necessary to define string-net models, using the constructions of [79] to describe the corresponding gapped domain walls. The corresponding UMTCs can be obtained by taking the (enriched) center. In the string-net picture, there is an equivalent definition of the relative Deligne product as a ladder category [9]. In physical terms, $\mathcal{M} \boxtimes_{\mathcal{Y}} \mathcal{N}$ describes the space of local ground states at the boundary, modulo the action of string operators in the $\mathcal{Y}$ bulk which have one endpoint on each domain wall, as in the following sketch.

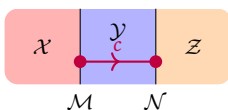

While such a string operator can create nontrivial wall excitations, since these excitations were obtained from a pair of antiparticles, they must fuse to the vacuum and so cannot be detected from far away.

**Warning 3.8.** As with $\boxtimes$ (see Warning 2.6), the operation which we have denoted by $\boxtimes_{\mathcal{Y}}$ is also conventionally denoted by $\boxtimes_{\mathcal{Y}}$, despite the apparently different definition. We have again introduced the notation $\boxtimes_{\mathcal{Y}}$ as a disambiguation.

**Remark 3.9.** One can check that, as an algebra in $\mathcal{Y}^{\mathrm{mp}} \boxtimes \mathcal{Y}$, the image of the canonical Lagrangian algebra (9) $K_{Z(\mathcal{Y})} \in \overline{Z(\mathcal{Y})} \boxtimes Z(\mathcal{Y}) \cong Z(\mathcal{Y}^{\mathrm{mp}} \boxtimes \mathcal{Y})$ is Morita equivalent to $S_{\mathcal{Y}}$, i.e. $(\mathcal{Y}^{\mathrm{mp}} \boxtimes \mathcal{Y})_{S_{\mathcal{Y}}}$ and $(\mathcal{Y}^{\mathrm{mp}} \boxtimes \mathcal{Y})_{K_{Z(\mathcal{Y})}}$ are equivalent as $\mathcal{Y}^{\mathrm{mp}} \boxtimes \mathcal{Y}$ module categories. In other words, ⊡ and ⊠ have equivalent definitions; the difference lies in interpretation, and in the fact that in the (3+1)D case where ⊡ is used, the additional structure of a braiding on the categories describing bulks and a tensor product on the categories describing the boundaries produces a tensor product on $\mathcal{U} \boxdot_{\mathcal{A}} \mathcal{V}$.

**Notation 3.10.** Just as we can compose bimodule categories for parallel domain walls to obtain a bimodule category which dictates the data for the composite boundary in a string net model, we can also compose Witt equivalences to obtain the bimodule (multi)tensor category of wall excitations on the composite domain wall. In the situation of (13), the category of excitations on the composite domain wall is $\mathrm{End}_{\mathcal{X}-\mathcal{Z}}(\mathcal{M} \boxdot_{\mathcal{Y}} \mathcal{N})$. For ordinary (i.e. unenriched) bimodule categories ${}_{\mathcal{Z}}\mathcal{M}_{\mathcal{Y}}$ and ${}_{\mathcal{Y}}\mathcal{N}_{\mathcal{Z}}$, by [90, Thm. 3.1.7 and 3.1.8], we have

$$\mathrm{End}_{\mathcal{X}-\mathcal{Z}}(\mathcal{M} \underset{\mathcal{Y}}{\boxdot} \mathcal{N}) \cong \mathrm{End}_{\mathcal{X}-\mathcal{Y}}(\mathcal{M}) \underset{Z(\mathcal{Y})}{\boxminus} \mathrm{End}_{\mathcal{Y}-\mathcal{Z}}(\mathcal{N}), \tag{14}$$

as $Z(\mathcal{X}) - Z(\mathcal{Z})$ bimodule multifusion categories, where ⊟ is defined identically to ⊡ (see (10)). The product $\underset{Z(\mathcal{Y})}{\boxminus}$ reflects the fact that when the two domain walls are composed, a pair of wall excitations associated with bringing $a$ to $\mathcal{M}$ and $\overline{a}$ to $\mathcal{N}$ is a trivial excitation, since the two anyons can be locally annihilated in the bulk.

**Warning 3.11.** Although the operations $\boxminus_{\mathcal{C}}$ and $\boxdot_{\mathcal{C}}$ have the same mathematical definition, and are both conventionally denoted by $\boxtimes_{\mathcal{C}}$, we use a different symbol to emphasize the different physical interpretations. Namely, the operation ⊡ describes how the data which defines the ground state Hilbert space of a commuting projector model for (2+1)D defects between (3+1)D Walker-Wang models behaves under stacking, while ⊟ describes the category of wall excitations on a composite (1+1)D domain wall between (2+1)D bulks.

We now describe how the above story generalizes to the case of nontrivial anomaly. Mathematically, the picture in (13) can be thought of as a picture in the diagrammatic calculus of $\mathrm{UFC}^{\mathcal{A}}$, where $\mathcal{X}$, $\mathcal{Y}$, $\mathcal{Z}$, $\mathcal{M}$, and $\mathcal{N}$ are now $\mathcal{A}$ enriched. In this case, the figure depicts a 2D region on the boundary of a (3+1)D Walker-Wang bulk, which we have not drawn. The $\mathcal{X} - \mathcal{Z}$ bimodule category $\mathcal{M} \boxdot_{\mathcal{Y}} \mathcal{N}$ will be $\mathcal{A}$-enriched by Lemma 2.12, Remark 3.9, and the fact that, since $Z(\mathcal{Y}) \cong \mathcal{A} \boxtimes Z^{\mathcal{A}}(\mathcal{Y})$, we have $K_{Z(\mathcal{Y})} \cong K_{\mathcal{A}} \boxtimes K_{Z^{\mathcal{A}}(\mathcal{Y})}$ as well.

In the $\mathcal{A}$-enriched setting, it is claimed in [93, Thm. 4.15] that (14) implies

$$\mathrm{End}^{\mathcal{A}}_{\mathcal{X}-\mathcal{Z}}(\mathcal{M} \underset{\mathcal{Y}}{\boxdot} \mathcal{N}) \cong \mathrm{End}^{\mathcal{A}}_{\mathcal{X}-\mathcal{Y}}(\mathcal{M}) \underset{Z^{\mathcal{A}}(\mathcal{Y})}{\boxminus} \mathrm{End}^{\mathcal{A}}_{\mathcal{Y}-\mathcal{Z}}(\mathcal{N}), \tag{15}$$

as $Z^{\mathcal{A}}(\mathcal{X}) - Z^{\mathcal{A}}(\mathcal{Z})$ bimodule multifusion categories.[27] This shows that ⊟ is still the correct operation to compose categories of excitations.

This mathematical argument has the following physical interpretation. As discussed in § 2.4, by Lemma 2.12, in the $\mathcal{A}$-enriched setting, each domain wall factors into a domain wall between the two enriched centers, and a trivial domain wall between $\mathcal{A}$ and itself; the latter is compatible with attaching the system to the boundary of a Walker-Wang model, leaving only a topological domain wall between the enriched centers. From this perspective, composing the domain walls corresponding to $\mathcal{A}$-enriched bimodule categories $\mathcal{M}$ and $\mathcal{N}$ amounts to composing domain walls $Z^{\mathcal{A}}(\mathcal{X}) - Z^{\mathcal{A}}(\mathcal{Y}) - Z^{\mathcal{A}}(\mathcal{Z})$ in one layer, and trivial domain walls in the $\mathcal{A}$ layer.

---

[27]The symmetry enriched version has also recently appeared in [120].

In fact, we can turn this physical picture into a proof of (15). By (the correspondence outlined in) Lemma 2.12, for an arbitrary indecomposable $\mathcal{A}$-enriched $\mathcal{X} - \mathcal{Y}$ bimodule $\mathcal{M}$, $\mathrm{End}_{\mathcal{X}-\mathcal{Y}}(\mathcal{M})$ is a Witt equivalence of the form $(Z(\mathcal{X}) \boxtimes Z(\mathcal{Y}))_{L \boxtimes K_{\mathcal{A}}}$, where $L$ is a Lagrangian algebra in $Z^{\mathcal{A}}(\mathcal{X}) \boxtimes \overline{Z^{\mathcal{A}}(\mathcal{Y})}$. Consequently, the point defects/wall excitations which live on this domain wall factor according to $\mathrm{End}_{\mathcal{X}-\mathcal{Y}}(\mathcal{M}) \cong \mathrm{End}_{\mathcal{X}-\mathcal{Y}}^{\mathcal{A}}(\mathcal{M}) \boxtimes \mathcal{A}$, where $\mathcal{A}$ is the identity $\mathcal{A} - \mathcal{A}$ bimodule.

This insight allows us to obtain (15) from (14), by exploiting the following sequence of equivalences of Witt equivalences $Z(\mathcal{X}) \to Z(\mathcal{Z})$.

$$
\begin{aligned}
&\mathrm{End}^{\mathcal{A}}(\mathcal{M} \underset{\mathcal{Y}}{\square} \mathcal{N}) \boxtimes \mathcal{A} \\
&\cong \mathrm{End}(\mathcal{M} \underset{\mathcal{Y}}{\square} \mathcal{N}) \\
&\cong \mathrm{End}(\mathcal{M}) \underset{Z(\mathcal{Y})}{\boxminus} \mathrm{End}(\mathcal{N}) \\
&\cong (\mathrm{End}^{\mathcal{A}}(\mathcal{M}) \boxtimes \mathcal{A}) \underset{Z^{\mathcal{A}}(\mathcal{Y}) \boxtimes \mathcal{A}}{\boxminus} (\mathrm{End}^{\mathcal{A}}(\mathcal{N}) \boxtimes \mathcal{A}) \\
&\cong (\mathrm{End}^{\mathcal{A}}(\mathcal{M}) \underset{Z^{\mathcal{A}}(\mathcal{Y})}{\boxminus} \mathrm{End}^{\mathcal{A}}(\mathcal{N})) \boxtimes \mathcal{A}.
\end{aligned}
$$

The two sides of (15) are now just the $(Z^{\mathcal{A}}(\mathcal{X}) \boxtimes 1) - (Z^{\mathcal{A}}(\mathcal{Z}) \boxtimes 1)$ bimodule tensor subcategories generated (as bimodule categories) by the tensor unit – in other words, the set of domain wall excitations corresponding to the identity particle in the $\mathcal{A}$ layer. These are precisely the domain wall excitations that are not confined by attaching the Walker-Wang bulk.

We now turn to the key issue which makes understanding the composition of parallel domain walls so difficult. The composition $\mathcal{M} \square_{\mathcal{Y}} \mathcal{N}$ of two indecomposable $\mathcal{A}$-enriched bimodule categories need not be indecomposable, and so the Witt equivalence $\mathrm{End}_{\mathcal{X}-\mathcal{Z}}^{\mathcal{A}}(\mathcal{M} \square_{\mathcal{Y}} \mathcal{N})$ of wall excitations on the composite wall is in general a *multifusion* category. In other words, it does not describe a single topological domain wall, but rather a direct sum of multiple distinct domain wall types. Physically, this means that stacking two domain walls $\mathcal{M}$ and $\mathcal{N}$ in general can decompose into a direct sum of distinct indecomposable superselection sectors. Each superselection sector corresponds to a type of gapped boundary that is invariant under the action of all local operators.

It is instructive to consider the following simple example involving the fusion of boundaries between $\mathbb{Z}/2$-toric code; complete fusion rules for such boundaries appear in [10], [94, Table 1].

**Example 3.12.** Consider a vertical strip of $\mathbb{Z}/2$ toric code, sandwiched between two smooth gapped boundaries to the vacuum, where the $m$-particle becomes condensed. Mathematically, we realize the toric code bulk $\mathcal{D}(\mathbb{Z}/2) := Z(\mathrm{Hilb}[\mathbb{Z}/2])$ from the fusion category $\mathcal{Y} = \mathrm{Hilb}[\mathbb{Z}/2]$, and the vacuum from the trivial fusion category $\mathcal{X} = \mathcal{Z} = \mathrm{Hilb}$. The smooth domain walls are obtained from the bimodule categories $\mathcal{M} = \mathcal{N} = \mathrm{Hilb}[\mathbb{Z}/2]$, with the action of $\mathrm{Hilb}[\mathbb{Z}/2]$ on $\mathrm{Hilb}[\mathbb{Z}/2]$ given by the tensor product.

As for the corresponding Witt equivalence between $\mathcal{D}(\mathbb{Z}/2)$ and $Z(\mathrm{Hilb}) \cong \mathrm{Hilb}$, the action $D(\mathbb{Z}/2) \to \mathrm{End}_{\mathrm{Hilb}[\mathbb{Z}/2]}(\mathrm{Hilb}[\mathbb{Z}/2]) \cong \mathrm{Hilb}[\mathbb{Z}/2]$ is given by the forgetful functor, which forgets the half-braiding. That is, if $\mathrm{Irr}(\mathcal{M}) = \mathrm{Irr}(\mathcal{N}) = \mathbb{Z}/2 = \{1, g\}$, where $g \in \mathbb{Z}/2$ is the nonidentity element, then the anyons 1 and $m$ forget to the tensor unit $1 \in \mathrm{Hilb}[\mathbb{Z}/2]$, i.e. the vacuum, while $e$ and $\epsilon$ forget to the nontrivial object in $\mathrm{Hilb}[\mathbb{Z}/2]$, and become nontrivial wall excitations. This gives the left (right) boundary of our Toric code strip the structure of a $\mathrm{Hilb} - \mathrm{Hilb}[\mathbb{Z}/2]$ ($\mathrm{Hilb}[\mathbb{Z}/2] - \mathrm{Hilb}$) bimodule tensor category.

In this case, the composite bimodule category $\mathcal{M} \square_{\mathcal{Y}} \mathcal{N}$ is just $\mathrm{Hilb}[\mathbb{Z}/2] \cong \mathrm{Hilb} \oplus \mathrm{Hilb}$ as a $\mathrm{Hilb} - \mathrm{Hilb}$ bimodule – i.e. the direct sum of two copies of the trivial domain wall from the trivial topological order to itself. The fact that there are two copies arises because, with

appropriate boundary conditions, the toric code strip has a GSD associated with the number of $m$-lines running between the two boundaries.

Example 3.12 illustrates a general property composing domain walls, described in detail in and below Theorem 3.22: in general, we can project onto an individual summand (or domain wall type) in the (decomposable) composite domain wall using a linear combinations of certain short string operators connecting the two domain walls. The short strings in question do not create wall excitations, because the associated anyons - in this case, the $m$ particles- are condensed at each wall. The resulting projectors are local, as they need not be separated by more than the width of the central $\mathcal{Y}$ strip. To go between different summands, however, requires a non-local operator, such as a string operator which is extended parallel to the domain walls. In our Example 3.12 above, the relevant operator is an $e$-string extending across the $\mathcal{Y}$ strip parallel to $\mathcal{M}$ and $\mathcal{N}$.

In contrast, the Witt equivalence of wall excitations on a composite domain wall, which is the composition of two Witt equivalences in UBFC, is not decomposable, since $\mathrm{Mod}(Z^{\mathcal{A}}(\mathcal{X}) \boxtimes \overline{Z^{\mathcal{A}}(\mathcal{Z})})$ is a connected fusion 2-category [45, Remark 2.1.22]. For example, in Example 3.12, the composition of the two Witt equivalences is $\mathrm{Hilb}[\mathbb{Z}/2] \boxplus_{\mathcal{D}(\mathbb{Z}/2)} \mathrm{Hilb}[\mathbb{Z}/2] \cong M_2(\mathrm{Hilb})$, an indecomposable unitary *multifusion* category (and hence an indecomposable $\mathrm{Hilb} - \mathrm{Hilb}$ bimodule tensor category). The underlying reason for this indecomposability is that the multifusion category $\mathrm{End}^{\mathcal{A}}_{\mathcal{X}-\mathcal{Z}}(\mathcal{M} \boxtimes_{\mathcal{Y}} \mathcal{N})$ of wall excitations on the composite domain wall consists both of localized excitations in the individual summands, and of point defects that connect different summands (i.e. different types of domain walls). We will further explore the details of this in § 4.4, once we have developed the necessary tools.

**Remark (Mathematical) 3.13.** Expanding on Mathematical Remark 2.4, we take a moment to discuss the issue of finding a correct 3-category of (2+1)D topological orders. It is natural that (2+1)D topological orders should form a 3-category, where objects are (2+1)D bulk topological orders, 1-morphisms are codimension 1 defects (i.e. domain walls), 2-morphisms are codimension 2 defects (i.e. point defects), and 3-morphisms are topological local operators. We have proposed $\mathrm{UFC}^{\mathcal{A}}$ as a 3-category of topological orders. However, (2+1)D topological orders are frequently understood in terms of the anyons, which form a UMTC. It is therefore natural to wonder if the 4-category UBFC is somehow related to a 3-category of (2+1)D topological orders.

In § 2.1 and (15), we saw that the categorical data which determine commuting projector models for (2+1)D bulk topological orders and (1+1)D domain walls between them, i.e. data in $\mathrm{UFC}^{\mathcal{A}}$, are mathematically interchangeable with the tensor categories describing bulk and wall excitations, which are their counterparts in UBFC.

A precise formulation of this statement is as follows. Consider the two 'truncated'[28] 1-categories

- $\mathrm{UFC}^{\mathcal{A}}_{\leq 1}$, where objects are $\mathcal{A}$-enriched fusion categories and morphisms are equivalence classes of $\mathcal{A}$-enriched bimodule categories

- $\mathrm{UMTC}_{\leq 1}$, where objects are UMTCs and morphisms are Morita equivalence classes of Witt equivalences.[29]

---

[28]Given an $n$-category $\mathcal{C}$, for $0 \leq k < n$, we can truncate to obtain a $k$-category $\mathcal{C}_{\leq k}$, where 0-morphisms up through $(k-1)$-morphisms are the same as in $\mathcal{C}$, and $k$-morphisms in $\mathcal{C}_{\leq k}$ are $k$-morphisms in $\mathcal{C}$ up to equivalence.

[29]Here, UMTC is the 4-subcategory of UBFC whose objects are UMTCs and whose higher morphisms are all invertible. For mathematical experts, $\mathrm{UMTC} = \Omega(\mathrm{core}(B(\mathrm{UBFC})))$, the loop space of the core of the delooping of UBFC.

The enriched center construction gives a functor $\mathsf{UFC}^{\mathcal{A}}_{\leq 1} \to \mathsf{UMTC}_{\leq 1}$ which is an equivalence onto its image, and by Example 2.3, everything in $\mathsf{UMTC}_{\leq 1}$ appears in the image of $\mathsf{UFC}^{\mathcal{A}}_{\leq 1}$ for some $\mathcal{A}$. Thus, these candidate 1-categories of (2+1)D topological order, where objects are (2+1)D bulk theories and morphisms are domain walls up to invertible point defect, agree.

The only question is whether this agreement can be extended to include point defects and topological local operators, which should respectively be 2-morphisms and 3-morphisms in a 3-category of (2+1)D topological orders. That is, is there an also embedding $\mathsf{UFC}^{\mathcal{A}}_{\leq 2} \to \mathsf{UMTC}_{\leq 2}$ or $\mathsf{UFC}^{\mathcal{A}} \to \mathsf{UMTC}_{\leq 3}$? If that were the case, one could analyze (2+1)D topological order by only considering the categories of excitations (bulk excitations, wall excitations, and super-selection sectors at point defects), and the machinery of enriched fusion categories which we have introduced would be superfluous. However, the equivalence outlined in § 2.1 fails to extend even one category level higher, because the composition of 1-morphisms in $\mathsf{UFC}^{\mathcal{A}}$ can be decomposable, while the composition of the corresponding 1-morphisms in UBFC is indecomposable. This can already be seen in studies of domain wall decomposition in the non-anomalous case [9, 10].

Decomposability of the composition of 1-morphisms in $\mathsf{UFC}^{\mathcal{A}}$ means that there are 2-morphisms, i.e. $\mathcal{A}$-balanced bimodule functors, which project onto individual summands. On the other hand, 2-morphisms in UBFC are bimodule categories between Witt equivalences, and there is no bimodule category which can project onto a corner of an indecomposable multifusion category. Thus, a hypothetical embedding $\mathsf{UFC}^{\mathcal{A}}_{\leq 2} \to \mathsf{UBFC}_{\leq 2}$ has nowhere to map the bimodule functors which project onto summands.

This can be compared to the basic fact that (for $n \geq 2$) the vector space $\mathbb{C}^n$ decomposes as a direct sum $\oplus_n \mathbb{C}$, while the algebra $\mathrm{End}(\mathbb{C}^n) = M_n(\mathbb{C})$ is simple. There are matrices in $M_n(\mathbb{C})$ projecting onto individual components of a vector in $\mathbb{C}^n$, i.e. diagonal matrices with a single nonzero entry, but $M_n(\mathbb{C})$ does not act on any of these 1-dimensional subspaces. We saw something extremely similar in Example 3.12: the space of ground states of the string-net model for the composite domain wall was described by $\mathsf{Hilb} \oplus \mathsf{Hilb}$, with two summands to project onto, but the multifusion category $M_2(\mathsf{Hilb})$ of wall excitations contains point defects between these two summands, and hence does not decompose as a direct sum of the (trivial) categories of wall excitations for each summand.

Thus, as candidate 3-categories of (2+1)D topological order, truncations of UMTC do not agree with [79] as to the possible point defects between domain walls, leading us to adopt UFC, and more generally $\mathsf{UFC}^{\mathcal{A}}$, as 3-categories of (2+1)D topological orders.

## 3.3 Decomposing composite domain walls

Consider the horizontal composition of two indecomposable domain walls, as in (13), which we reproduce here.

$$_{\mathcal{X}}\mathcal{M} \boxtimes_{\mathcal{Y}} \mathcal{N}_{\mathcal{Z}} \cong \boxed{\begin{array}{c|c|c} \mathcal{X} & \mathcal{Y} & \mathcal{Z} \\ & \mathcal{M} \quad \mathcal{N} & \end{array}} \tag{13}$$

As we have seen in Example 3.12, such a composite domain wall can be decomposable, meaning that there are multiple superselection sectors. A superselection sector is characterized by the fact that it remains unchanged when any local operator acts on or near the composite domain wall, and that it cannot be further decomposed into components invariant under such local operators. Each superselection sector is therefore an indecomposable domain wall between $Z^{\mathcal{A}}(\mathcal{X})$ and $Z^{\mathcal{A}}(\mathcal{Z})$ bulk topological orders.

Mathematically, these superselection sectors are the summands of the $\mathcal{A}$-enriched $\mathcal{X} - \mathcal{Z}$ bimodule category $\mathcal{M} \boxtimes_{\mathcal{Y}} \mathcal{N}$. In this section, we provide the mathematical tools needed to

obtain them, and to describe the local operators which project into each superselection sector. We will show how to use these tools in many explicit examples in § 5 below.

Before giving the full mathematical arguments, we present a physical outline of our results. To understand the superselection sectors of composite domain walls, we must first identify the (semisimple) algebra $S$ of topological local operators which act on a composite domain wall and preserve the space of ground states. Because these operators are local, by definition superselection sectors must be fixed under their action. It follows that superselection sectors must be the images of minimal central projections of $S$, i.e. minimal projections in $Z(S)$.[30]

One class of local operator which acts non-trivially on the space of ground states can be obtained by creating a particle-antiparticle pair $c \boxtimes \bar{c}$ in the $Z^{\mathcal{A}}(\mathcal{Y})$ bulk region, and bringing one to each domain wall. If $\bar{c}$ ($c$) is condensed at the left (right) wall, as depicted in (16), then the resulting local operator creates no topological excitations, i.e. the trivial domain wall excitations $1_{\mathcal{M}}$ and $1_{\mathcal{N}}$, and hence can return the system to its ground state.

$$
\begin{array}{c}
\boxed{\mathcal{X} \quad \mathcal{Y} \quad \mathcal{Z}} \\
\mathcal{M} \quad \mathcal{N}
\end{array}
\tag{16}
$$

As we will later show in the proof of Theorem 3.22, modulo operators which act *trivially* on the space of ground states (such as creating a particle-antiparticle pair, moving them onto the same boundary, and then annihilating them), linear combinations of such operators make up all of $S$.

What is the vector space of linear combinations of operators of the form (16)? Suppose the Lagrangian algebras corresponding to the $\mathcal{A}$-enriched bimodule categories $\mathcal{M}$ and $\mathcal{N}$ are $L_1 = L(A, \overline{B_1}, \Phi)$ and $L_2 = L(B_2, \overline{C}, \Psi)$ respectively, so that $B_1$ and $B_2$ are the algebras in $Z^{\mathcal{A}}(\mathcal{Y})$ which condense at each domain wall. Then the data of an operator of the form (16) consists of three choices: an anyon type $c \in \mathrm{Irr}(Z^{\mathcal{A}}(\mathcal{Y}))$, and morphisms $\alpha \in Z^{\mathcal{A}}(\mathcal{Y})(\bar{c} \to B_1) \cong Z^{\mathcal{A}}(\mathcal{Y})(B_1 \to c)$ and $\beta \in Z^{\mathcal{A}}(\mathcal{Y})(c \to B_2)$. which live in the multiplicity spaces of the anyons $\bar{c}, c$ in $B_1, B_2$ respectively. These morphisms determine how $c$ is condensed at each wall. The space of linear combinations of such operators is thus

$$
\bigoplus_{c \in \mathrm{Irr}(Z^{\mathcal{A}}(\mathcal{Y}))} Z^{\mathcal{A}}(\mathcal{Y})(B_1 \to c) \otimes Z^{\mathcal{A}}(\mathcal{Y})(c \to B_2),
\tag{17}
$$

and composing the two tensor factors gives an isomorphism to the hom space[31] $Z^{\mathcal{A}}(\mathcal{Y})(B_1 \to B_2)$, which describes the set of possible short string operators in (16).

Now that we have identified the space of local operators as $Z^{\mathcal{A}}(\mathcal{Y})(B_1 \to B_2)$, we need to be able to multiply two such operators. To do so, we can apply them in parallel. The strings passing through the $Z^{\mathcal{A}}(\mathcal{Y})$ can be topologically deformed, and therefore fused:

$$
\begin{array}{c}
\boxed{\mathcal{X} \quad \mathcal{Y} \quad \mathcal{Z}} \\
\mathcal{M} \quad \mathcal{N}
\end{array}
= \sum
\begin{array}{c}
\boxed{\mathcal{X} \quad \mathcal{Y} \quad \mathcal{Z}} \\
\mathcal{M} \quad \mathcal{N}
\end{array}.
$$

Diagrams of the form

$$
\begin{array}{c}
\boxed{\mathcal{Y} \quad \mathcal{Z}} \\
\mathcal{N}
\end{array},
\tag{18}
$$

---

[30]The Abelian algebra $Z(S)$ is isomorphic to $\mathbb{C}^n$ for some $n$, and the minimal projections are the $e_i \in \mathbb{C}^n$ which are all zeroes except for a 1 in the $i$-th slot.

are local operators which take an $f$-particle in the $Z^{\mathcal{A}}(\mathcal{Y})$ bulk and condense it on the $\mathcal{N}$ wall, so they should be elements of $Z^{\mathcal{A}}(\mathcal{Y})(f \to B_2)$. Resolving them as such, however, is not trivial. For one thing, if anyons $c$ and $d$ appear in the condensate $B_2$, it is still possible that some fusion products $f$ of $c$ and $d$ do not appear in $B_2$, so that the morphism (18) may be 0.

The data required to resolve such vertices comes from the multiplication $m : B_2 B_2 \to B_2$ on $B_2$: if $x : c \to B_2$ and $y : d \to B_2$ are the data used to condense $c$ and $d$ at the wall, then the morphism in (18) is precisely $m \circ (x \otimes y)$. Applying this at both domain walls, we find that $Z^{\mathcal{A}}(\mathcal{Y})(B_1 \to B_2)$ carries a multiplication $\star$ which we now define.

**Definition 3.14.** Suppose $\mathcal{C}$ is a UMTC and $A, B \in \mathcal{C}$ are condensable algebras. The *convolution multiplication* $\star$ on the hom space[31] $\mathcal{C}(A \to B)$ is given in the graphical calculus for $\mathcal{C}$ by

$$\begin{array}{c}\boxed{x}\Big|_A^B \star \boxed{y}\Big|_A^B := \boxed{x}\;\boxed{y}\end{array}.$$

This product gives a way of composing the two maps $x$ and $y$, both of which take objects in $A$ to objects in $B$, into a single map from $A$ to $B$. If $u : 1 \to A$ and $v : 1 \to B$ are the units of each algebra, then $vu^\dagger \in \mathcal{X}(A \to B)$ is the identity for $\star$. The involution[32] is given by

$$\left(\boxed{x}\Big|_A^B\right)^* := \; {}_A\boxed{x^\dagger}{}^B \;.$$

Here, the univalent vertices are the units of the algebras (see Appendix B on condensable algberas). One can view this cap and cup (with the univalent vertices) as standard duality pairings of $A$ and $B$ respectively, which is similar to an $(a, \bar{a}, 1)$ vertex for an anyon $a$ in a UMTC. The hom space $\mathcal{C}(A \to B)$ with this convolution multiplication and involution is a finite dimensional C$^*$ algebra. Moreover, the multiplication is commutative because

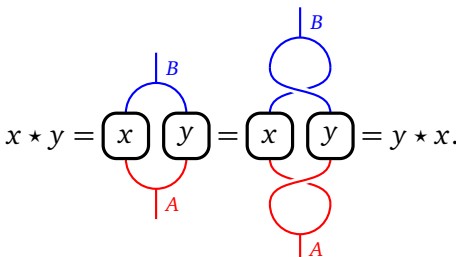

$$x \star y = \boxed{x}\;\boxed{y} = \boxed{x}\;\boxed{y} = y \star x.$$

The commutativity of $(Z^{\mathcal{A}}(\mathcal{Y})(B_1 \to B_2), \star)$ also makes sense in terms of the string operators across the $Z^{\mathcal{A}}(\mathcal{Y})$ bulk going between domain walls. Since the strings can be deformed topologically, the middle parts of two parallel strings can slide past one another, and commutativity of $B_1$ and $B_2$ means that the endpoints on the domain walls can also change places.

Thus, to find the superselection sectors of the composite domain wall $\mathcal{M} \boxtimes_{\mathcal{Y}} \mathcal{N}$, we need only diagonalize the finite dimensional commutative C$^*$ algebra $Z^{\mathcal{A}}(B_1 \to B_2)$.[33] We carry this computation out in each of our examples in § 5.

---

[31]Suppose the simple objects of a UFC $\mathcal{X}$ are $\mathrm{Irr}(\mathcal{X}) = \{x_1, \ldots, x_n\}$. Given objects $a, b \in \mathcal{X}$, we can write $a = \bigoplus_{i=1}^n n_i x_i$ and $b = \bigoplus m_i x_i$. The hom space $\mathcal{X}(a \to b) = \bigoplus_{i=1}^n M_{m_i \times n_i}(\mathbb{C})$. Matrix units $E_{k\ell}$ for $M_{m_i \times n_i}(\mathbb{C})$ are given by the rank one operators $|x_{i,k}\rangle\langle x_{i,\ell}|$, where $x_{i,\ell}$ denotes the $\ell$-th copy of $x_i$ in $a$ and $x_{i,k}$ denotes the $k$-th copy of $x_i$ in $b$.

[32]One can define this $*$ in the case $A, B \in \mathcal{X}$ are Frobenius, i.e., the multiplication $m_A^\dagger$ is an $A - A$ bimodule map, and similarly for $B$.

[33]Since C$^*$-algebras a semisimple, $Z^{\mathcal{A}}(B_1 \to B_2)$ is a finite dimensional Abelian semisimple algebra, i.e. it is isomorphic to $\oplus_n \mathbb{C}^n$ for some $n$. The minimal projections are the $e_i$ as in Footnote 30.

The remainder of this section is devoted to a rigorous justification of the results we have just outlined. The main detail we did not explain above is why $Z^{\mathcal{A}}(B_1 \to B_2)$ is the correct algebra of topological local operators.

### 3.3.1 Dualizability

The calculations in the remainder of this section utilize the graphical calculus for 3-categories from [19], which applies to weak 3-categories by [62]; in particular, we work in the 3-category UFC$^{\mathcal{A}}$, which satisfies important finiteness, semisimplicity, and dualizability conditions as it is a full 3-subcategory of a hom 3-category in UBFC.[34]

There are 2D and 3D diagrams drawn in this section. 3D diagrams represent the top level of 3-morphisms in UFC$^{\mathcal{A}}$, i.e., topological local operators (see Figure 3). These include 1D world-lines of anyons in the bulk regions or point defects on domain walls, 2D surfaces corresponding to world-sheets of domain walls, 3D world-volumes of bulk topological order, and 0D point-like operators. The horizontal 2D slices are spatial, and the third vertical dimension represents time. Reading our diagram from bottom to top represents the time-ordering of applying topological local operators.

More specifically, a bulk 3D space-time region is labelled by an $\mathcal{A}$-enriched fusion category $\mathcal{X}, \mathcal{Y}, \mathcal{Z}$ which determines a (2+1)D topological order with anomaly $\mathcal{A}$. A codimension 1 surface separating a pair of 3D space-time regions is labelled by $\mathcal{A}$-enriched bimodule category $\mathcal{M}, \mathcal{N}$, which specifies a (1+1)-dimensional topological domain wall. A codimension 2 line is labelled by a $\mathcal{A}$-centered bimodule functor, which specifies a point defect. When such a line lives in a 3D bulk region, it depicts an anyon world-line. When it is localized to a domain wall world-sheet, however, it corresponds to the world-line of a point-like excitation on the domain wall, or of a point defect separating two different domain walls. 0D point defects in our 3D space-time diagrams are labelled by natural transformations, which correspond to completely local, topologically trivial, operators.

The 2D diagrams can be thought of as spatial slices of the 3D diagrams at a fixed time. As in previous sections of this paper, each bulk 2-dimensional region can therefore be thought of as a topologically ordered ground state, described by a $\mathcal{A}$-enriched fusion category $\mathcal{X}$. 2D diagrams can also contain domain walls $\mathcal{M}, \mathcal{N}, \ldots$ separating different $\mathcal{A}$-enriched fusion categories, as well as point defects – i.e. with $\mathcal{A}$-centered bimodule functors, which are 2-morphisms in UFC$^{\mathcal{A}}$ one level down. Finally, 2D diagrams which involve arrows connecting point defects, which we have previously used (e.g. (16)) to represent short string operators, should be interpreted as 3-morphisms (or types of 3-morphisms), where the string with the arrow shows the passage of time. In other words, the 2D diagram simultaneously depicts the source and target of a 3-morphism.

In the formulae below, we will associate a 2D diagram with each possible type of point or line defect. Thus, a 2D region labelled by a single $\mathcal{A}$-enriched fusion category $\mathcal{X}$, containing no visible line or point defects, depicts the trivial/identity point defect. This corresponds to the identity endofunctor of $\mathcal{X}$ (viewed as an $\mathcal{A}$-enriched $\mathcal{X} - \mathcal{X}$ bimodule category), namely $1 \in \text{Irr}(Z^{\mathcal{A}}(\mathcal{X}))$, as explained in Example 2.9. Similarly, a 2D diagram with only a single 0D point defect in the bulk represents an $\mathcal{A}$-centered endofunctor of $_{\mathcal{X}}\mathcal{X}_{\mathcal{X}}$, which corresponds to an object of $Z^{\mathcal{A}}(\mathcal{X})$, i.e. an anyon or direct sum of anyons. A 2D diagram with a single 1D defect between 2D regions labelled by an $\mathcal{A}$-enriched $\mathcal{X} - \mathcal{Y}$ bimodule $_{\mathcal{X}}\mathcal{M}_{\mathcal{Y}}$ with no point defect denotes the identity endofunctor of $\mathcal{M}$ in $\text{End}^{\mathcal{A}}_{\mathcal{X}-\mathcal{Y}}(\mathcal{M})$, i.e. the trivial wall excitation. Throughout this section, we will consistently use pink to denote $\mathcal{X}$-labeled regions, and light

---

[34]Here, we assume the 3-category UFC$^{\mathcal{A}}$ has sufficient additional structure to permit the use of the 3D graphical calculus, as in [19, 45]. Physically, the use of this graphical calculus can be justified via the notion of *topological Wick rotation* [92, 93].

blue to denote $\mathcal{Y}$-labeled regions, unless otherwise stated

$$\boxed{\mathcal{X}} = 1_{Z^{\mathcal{A}}(\mathcal{X})}, \qquad \boxed{\substack{\mathcal{M} \\ \mathcal{X} \mid \mathcal{Y}}} = \mathrm{id}_{\mathcal{M}}.$$

Dualizability in UFC$^{\mathcal{A}}$ means we can topologically deform world-sheets of domain walls in space-time. We use dualizability in this section in several important ways.

**Example 3.15.** Dualizability allows us to close a line corresponding to an (irreducible) $\mathcal{A}$-enriched bimodule $_{\mathcal{X}}\mathcal{M}_{\mathcal{Y}}$ into a closed loop. Zooming out, we may view this loop as a point defect, labeled by direct sum of those anyons in $Z^{\mathcal{A}}(\mathcal{X})$ that can dissappear at the domain wall $\mathcal{M}$ with no energy cost. Dualizability also means we can form a 'pair of pants' multiplication for this object, under which it becomes a condensable algebra in $Z^{\mathcal{A}}(\mathcal{X})$ (up to scaling).

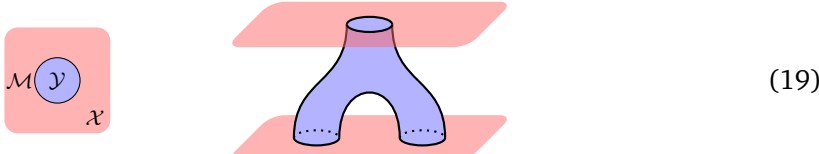

$$(19)$$

**Sub-Example 3.16.** When $\mathcal{A} = \mathsf{Hilb}_{\mathsf{fd}}$ is the trivial enrichment, it is well known [66, 69, 85, 106] that, if a module category $_{\mathcal{X}}\mathcal{M}$ labels a topological boundary to vacuum, and $L \in Z(\mathcal{X})$ is the Lagrangian algebra associated to $_{\mathcal{X}}\mathcal{M}$, i.e. $\mathrm{End}_{\mathcal{X}}(\mathcal{M}) \cong Z(\mathcal{X})_L$, then we have

$$\boxed{\substack{\mathcal{M}\bigcirc \\ \mathcal{X}}} = L.$$

Physically, this means that the point defect can contain any anyon in $L$, since these are condensed in the white region and therefore cost no energy.

**Sub-Example 3.17.** More generally, we can consider the case when $\mathcal{M}$ is a condensation boundary. If the blue region is obtained from the pink one via condensing $A$, i.e. $Z^{\mathcal{A}}(\mathcal{Y}) = Z^{\mathcal{A}}(\mathcal{X})_A^{\mathrm{loc}}$, then the circle bounded by $\mathcal{M}$ is a droplet of condensate. As a defect in the $Z^{\mathcal{A}}(\mathcal{X})$ bulk, it is equivalent to $A$, the the direct sum (with multiplicity) of all the anyons which have condensed.

On the other hand, if the pink region is obtained from the blue one via condensing $B$, i.e. $Z^{\mathcal{A}}(\mathcal{X}) = Z^{\mathcal{A}}(\mathcal{Y})_B^{\mathrm{loc}}$, then the circle bounded by $\mathcal{M}$ is a small region where $B$ is not condensed. Because this region is surrounded by condensate, and does not contain an excitation, it is equivalent to the vacuum of the $Z^{\mathcal{A}}(\mathcal{Y})_B^{\mathrm{loc}}$-bulk; the circle is the same as a pink sheet with nothing in it.

These assertions are justified in Lemma 3.19 and Proposition 3.20 below.

**Example 3.18.** Just as we have the isomorphism $\mathrm{End}_{\mathcal{Z}}(z) \cong \mathrm{Hom}_{\mathcal{Z}}(1_{\mathcal{Z}} \to z\bar{z})$ in a fusion category $\mathcal{Z}$, given an $\mathcal{A}$-enriched bimodule $_{\mathcal{X}}\mathcal{M}_{\mathcal{Y}}$, we have an isomorphism

$$\mathrm{End}^{\mathcal{A}}\left( \boxed{\substack{\mathcal{M} \\ \mathcal{X} \mid \mathcal{Y}}} \right) \cong \mathrm{Hom}_{Z^{\mathcal{A}}(\mathcal{X})}\left( \boxed{1_{Z^{\mathcal{A}}(\mathcal{X})}} \to \boxed{\substack{\mathcal{M}\bigcirc \\ \mathcal{X}}} \right),$$

where the maps both ways are given by

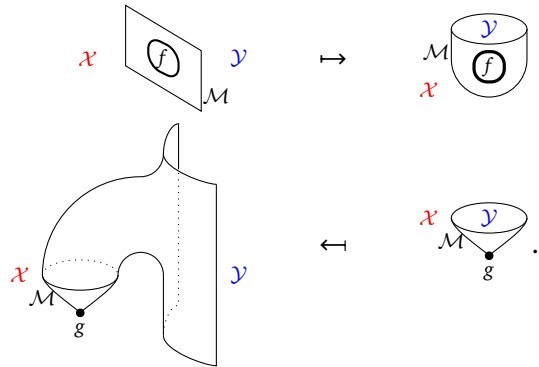

### 3.3.2 Decomposing a domain wall

Suppose $\mathcal{X}, \mathcal{Y} \in \mathsf{UFC}^{\mathcal{A}}$ are $\mathcal{A}$-enriched fusion categories, and $\mathcal{M}$ is an $\mathcal{A}$-enriched $\mathcal{X} - \mathcal{Y}$ bimodule category, which may be decomposable. We now see how the condensable algebra of an $\mathcal{M}$-loop can be used to decompose $\mathcal{M}$ into irreducible $\mathcal{A}$-enriched $\mathcal{X} - \mathcal{Y}$ bimodule summands.

First, we note that the hom 2-category $\mathsf{UFC}^{\mathcal{A}}(\mathcal{X} \to \mathcal{Y})$ of $\mathcal{A}$-enriched $\mathcal{X} - \mathcal{Y}$ bimodule categories is finitely semisimple in the sense of [45, Definition 1.4.2], as it is a hom 2-category in UBFC. By [45, Prop. 1.3.16], indecomposable $\mathcal{A}$-enriched $\mathcal{X} - \mathcal{Y}$ bimodule summands of ${}_{\mathcal{X}}\mathcal{M}_{\mathcal{Y}}$ correspond to minimal projections in $\mathrm{End}^{\mathcal{A}}_{\mathcal{X}-\mathcal{Y}}(\mathrm{id}_{\mathcal{M}})$. Now, using Example 3.18, these minimal projections correspond to copies of the unit $1_{Z^{\mathcal{A}}(\mathcal{X})}$ in the condensable algebra corresponding to the $\mathcal{M}$-loop from Example 3.15. This algebra is the image of $1_{Z^{\mathcal{A}}(\mathcal{Y})}$ under the lax monoidal functor $\overline{Z^{\mathcal{A}}(\mathcal{Y})} \to Z^{\mathcal{A}}(\mathcal{X})$ given by

$$\overline{Z^{\mathcal{A}}(\mathcal{Y})} \to \mathrm{End}^{\mathcal{A}}_{\mathcal{X}-\mathcal{Y}}(\mathcal{M}) \xrightarrow{I} Z^{\mathcal{A}}(\mathcal{X}),$$

where $I$ is the adjoint of the tensor functor $Z^{\mathcal{A}}(\mathcal{X}) \to \mathrm{End}^{\mathcal{A}}_{\mathcal{X}-\mathcal{Y}}(\mathcal{M})$.

In the case of trivial anomaly, we identify the condensable algebra in question in the following lemma. We will then generalize to the case of arbitrary $\mathcal{A}$ in Proposition 3.20.

**Lemma 3.19.** *When $\mathcal{A} = \mathsf{Hilb}$ is the trivial enrichment,*

$$\boxed{\mathcal{M}\,\mathcal{Y}\atop\mathcal{X}} = L \cap Z(\mathcal{X}) \boxtimes 1, \tag{20}$$

*where $L \in Z(\mathcal{X}) \boxtimes \overline{Z(\mathcal{Y})}$ is the Lagrangian algebra corresponding to the $\mathcal{X} \boxtimes \mathcal{Y}^{\mathrm{mp}}$-module category $\mathcal{M}$.*

*Proof.* By the folding trick, we can view $\mathcal{M}$ as a $\mathcal{X} \boxtimes \mathcal{Y}^{\mathrm{mp}}$-Hilb module category, corresponding to a Lagrangian algebra $L \in Z(\mathcal{X}) \boxtimes \overline{Z(\mathcal{Y})}$ by [41, Def. 3.3]. As in Sub-Example 3.16,

$$\mathcal{M}\bigcirc = {\mathcal{Y}^{\mathrm{mp}}\atop{\mathcal{X}}} = L, \tag{21}$$

where the purple color denotes the stacking of the blue and pink sheets. We then obtain the left-hand side of Equation (20) by closing up the $\mathcal{Y}^{\mathrm{mp}}$ sheet to a hemisphere

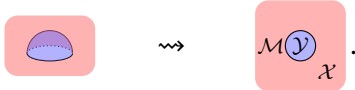

Since the only excitation supported in a contractible region of $Z(\mathcal{Y}^{\mathrm{mp}})$-bulk is the vacuum, this reduces $L$ to the right-hand side of Equation (20). □

**Proposition 3.20.** *For an arbitrary $\mathcal{A}$-enrichment,*

$$\text{ } = A \in Z^{\mathcal{A}}(\mathcal{X}),$$

*where $L(A, \overline{B}, \Phi) \in Z^{\mathcal{A}}(\mathcal{X}) \boxtimes \overline{Z^{\mathcal{A}}(\mathcal{Y})}$ is the Lagrangian algebra corresponding to the Witt-equivalence $\mathrm{End}^{\mathcal{A}}_{\mathcal{X}-\mathcal{Y}}(\mathcal{M})$ from Construction 2.8.*

*Proof.* By Lemma 3.19, ignoring the $\mathcal{A}$-enrichment, i.e. forgetting the $\mathcal{A}$-centered structure and considering $\mathcal{M}$ as an $\mathcal{X} - \mathcal{Y}$ bimodule category, we know that a closed $\mathcal{M}$-loop with external $\mathcal{X} \boxtimes \mathcal{Y}^{\mathrm{mp}}$ shading corresponds to a Lagrangian algebra $L$. But since $\mathcal{X}, \mathcal{Y}$ are $\mathcal{A}$-enriched and $_{\mathcal{X}}\mathcal{M}_{\mathcal{Y}}$ is an $\mathcal{A}$-enriched bimodule, by Lemma 2.12, $L$ contains the canonical Lagrangian $K_{\mathcal{A}}$ as a subalgebra where

$$Z(\mathcal{X} \boxtimes \mathcal{Y}^{\mathrm{mp}}) \cong Z(\mathcal{X}) \boxtimes \overline{Z(\mathcal{Y})} \cong Z^{\mathcal{A}}(\mathcal{X}) \boxtimes \overline{Z^{\mathcal{A}}(\mathcal{Y})} \boxtimes \mathcal{A} \boxtimes \overline{\mathcal{A}}.$$

Now instead of ignoring the $\mathcal{A}$-enrichment at the beginning, we can perform the $\mathcal{A}$-enriched folding trick (see Remark 3.4) for $\mathcal{A}$-enriched fusion categories, which also requires we perform a relative tensor product over $\mathcal{A}$, leading to an anomaly cancellation. This relative tensor product over $\mathcal{A}$ is accomplished by condensing $K_{\mathcal{A}}$. So the enriched Lagrangian algebra we obtain from the $\mathcal{A}$-enriched folding trick is isomorphic to the image of $L$ after condensing $K_{\mathcal{A}}$ in

$$Z^{\mathcal{A}}(\mathcal{X}) \boxtimes \overline{Z^{\mathcal{A}}(\mathcal{Y})} \cong Z(\mathcal{X} \boxtimes \mathcal{Y})^{\mathrm{loc}}_{K_{\mathcal{A}}}.$$

This yields exactly $L(A, B, \Phi)$, by Construction 2.11. The result now follows by gluing in a hemisphere (with attached $\mathcal{A}$-bulk) corresponding to $\mathcal{Y}$, as in Lemma 3.19. □

Combining Example 3.18, [45, Prop. 1.3.16], and Proposition 3.20 above, we get the following result.

**Corollary 3.21.** *Indecomposable $\mathcal{X} - \mathcal{Y}$ summands of an $\mathcal{A}$-enriched $\mathcal{X} - \mathcal{Y}$ bimodule $\mathcal{M}$ correspond to copies of $1_{Z^{\mathcal{A}}(\mathcal{X})}$ in the condensable algebra $A \in Z^{\mathcal{A}}(\mathcal{X})$, where $L(A, \overline{B}, \Phi) \in Z^{\mathcal{A}}(\mathcal{X}) \boxtimes \overline{Z^{\mathcal{A}}(\mathcal{Y})}$ is the Lagrangian algebra corresponding to the Witt-equivalence $\mathrm{End}^{\mathcal{A}}_{\mathcal{X}-\mathcal{Y}}(\mathcal{M})$.*

### 3.3.3 Decomposing a composite domain wall

Now, we are finally ready to mathematically justify our identification of the algebra of local operators which preserve the ground state of the composite domain wall $\mathcal{M} \boxtimes_{\mathcal{Y}} \mathcal{N}$ with $Z^{\mathcal{A}}(\mathcal{Y})(B_1 \rightarrow B_2)$. As just explained in § 3.3.2, this algebra is $\mathrm{End}^{\mathcal{A}}_{\mathcal{X}-\mathcal{Z}}(\mathrm{id}_{\mathcal{M} \boxtimes_{\mathcal{Y}} \mathcal{N}})$, by definition. By applying the results of the previous subsection, we will verify that $\mathrm{End}^{\mathcal{A}}_{\mathcal{X}-\mathcal{Z}}(\mathrm{id}_{\mathcal{M} \boxtimes_{\mathcal{Y}} \mathcal{N}}) \cong (Z^{\mathcal{A}}(\mathcal{Y})(B_1 \rightarrow B_2), \star)$, as promised.

**Theorem 3.22.** *Suppose $_{\mathcal{X}}\mathcal{M}_{\mathcal{Y}}$ and $_{\mathcal{Y}}\mathcal{N}_{\mathcal{Z}}$ are two $\mathcal{A}$-enriched bimodules between the $\mathcal{A}$-enriched fusion categories $\mathcal{X}, \mathcal{Y}, \mathcal{Z}$. Let*

$$L_1 = L(A, \overline{B_1}, \Phi) \in Z^{\mathcal{A}}(\mathcal{X}) \boxtimes \overline{Z^{\mathcal{A}}(\mathcal{Y})},$$
$$L_2 = L(B_2, \overline{C}, \Psi) \in Z^{\mathcal{A}}(\mathcal{Y}) \boxtimes \overline{Z^{\mathcal{A}}(\mathcal{Z})},$$

*be the Lagrangian algebras corresponding to the Witt-equivalences* $\mathrm{End}^{\mathcal{A}}_{\mathcal{X}-\mathcal{Y}}(\mathcal{M})$ *and* $\mathrm{End}^{\mathcal{A}}_{\mathcal{Y}-\mathcal{Z}}(\mathcal{N})$ *respectively from Construction 2.8. Indecomposable $\mathcal{A}$-enriched $\mathcal{X}-\mathcal{Z}$ bimodule summands of* $_{\mathcal{X}}\mathcal{M} \boxdot_{\mathcal{Y}} \mathcal{N}_{\mathcal{Z}}$ *correspond to minimal projections[33] in the Abelian algebra $(Z^{\mathcal{A}}(\mathcal{Y})(B_1 \to B_2), \star)$, as defined in Definition 3.14.*

For the graphical proof, we use the same region shadings from (13).

*Proof.* Since the $\mathcal{A}$-enriched $\mathcal{X}-\mathcal{Z}$ bimodules form a finitely semisimple 2-category, indecomposable summands of $_{\mathcal{X}}\mathcal{M} \boxdot_{\mathcal{Y}} \mathcal{N}_{\mathcal{Z}}$ correspond to minimal projections in $\mathrm{End}^{\mathcal{A}}_{\mathcal{X}-\mathcal{Z}}(\mathrm{id}_{\mathcal{M}\boxdot_{\mathcal{Y}}\mathcal{N}})$ [45, Prop. 1.3.16].

In more physical terms, the $\mathcal{A}$-enriched bimodule (a 1-morphism in $\mathsf{UFC}^{\mathcal{A}}$) $_{\mathcal{X}}\mathcal{M}\boxdot_{\mathcal{Y}}\mathcal{N}_{\mathcal{Z}}$ describes the system depicted in (13), the endofunctor $\mathrm{id}_{_{\mathcal{X}}\mathcal{M}\boxdot_{\mathcal{Y}}\mathcal{N}_{\mathcal{Z}}}$ (a 2-morphism) corresponds to the vacuum states of this system, i.e. when there are no pointlike localized excitations, and 3-morphisms in $\mathrm{id}_{_{\mathcal{X}}\mathcal{M}\boxdot_{\mathcal{Y}}\mathcal{N}_{\mathcal{Z}}}$ are topological local operators that preserve the ground states. Thus, minimal projections in $\mathrm{End}^{\mathcal{A}}_{\mathcal{X}-\mathcal{Z}}(\mathrm{id}_{\mathcal{M}\boxdot_{\mathcal{Y}}\mathcal{N}})$ are topological local operators which project onto a single summand of the composite domain wall.

Therefore, we need only check that $\mathrm{End}^{\mathcal{A}}_{\mathcal{X}-\mathcal{Z}}(\mathrm{id}_{\mathcal{M}\boxdot_{\mathcal{Y}}\mathcal{N}}) \cong (Z^{\mathcal{A}}(\mathcal{Y})(B_1 \to B_2), \star)$. By dualizability, we have

$$\mathrm{End}^{\mathcal{A}}_{\mathcal{X}-\mathcal{Z}}\left( \begin{array}{c} \mathcal{X}\ \mathcal{Y}\ \mathcal{Z} \\ \mathcal{M}\ \ \mathcal{N} \end{array} \right) \cong \mathrm{Hom}_{Z^{\mathcal{A}}(\mathcal{Y})}\left( \begin{array}{c} \mathcal{X}\mathcal{M} \\ \mathcal{Y} \end{array} \to \begin{array}{c} \mathcal{N}\mathcal{Z} \\ \mathcal{Y} \end{array} \right).$$

By Proposition 3.20, we have

$$\begin{array}{c} \mathcal{X}\mathcal{M} \\ \mathcal{Y} \end{array} = \overline{L_1} \cap 1 \boxtimes Z^{\mathcal{A}}(\mathcal{Y}) = B_1,$$

$$\begin{array}{c} \mathcal{N}\mathcal{Z} \\ \mathcal{Y} \end{array} = L_2 \cap Z^{\mathcal{A}}(Y) \boxtimes 1 = B_2,$$

so $\mathrm{End}^{\mathcal{A}}_{\mathcal{X}-\mathcal{Z}}(\mathrm{id}_{\mathcal{M}\boxdot_{\mathcal{Y}}\mathcal{N}}) \cong (Z^{\mathcal{A}}(\mathcal{Y})(B_1 \to B_2), \star)$ as $*$-algebras, and the result follows. $\qquad\square$

We will decompose $\mathcal{M} \boxdot_{\mathcal{Y}} \mathcal{N}$ using Theorem 3.22 for many explicit examples in § 5 below.

**Remark 3.23.** One reason that we investigated the algebra of local operators abstractly, rather than as operators on the Hilbert space of ground states, is that the space of ground states depends on our choice of manifold. We point out one particular case. Observe that, if we place the parallel domain walls in the statement of Theorem 3.22 on a sphere, as in

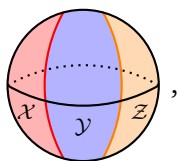

,

then the space of ground states is exactly $Z^{\mathcal{A}}(\mathcal{Y})(B_1 \to B_2)$. Thus, we see that in the right circumstances, there is a factor of topological ground state degeneracy local to the strip between the two walls which is isomorphic to $Z^{\mathcal{A}}(\mathcal{Y})(B_1 \to B_2)$ as a representation of $Z^{\mathcal{A}}(\mathcal{Y})(B_1 \to B_2)$. We expand on this idea in § 4 and § 5, where we will see that, when the strip of $Z^{\mathcal{A}}(\mathcal{Y})$ bulk is closed up to a tube, noncontractible (and thus nonlocal) Wilson loop operators in the $Z^{\mathcal{A}}(\mathcal{Y})$ strip which fail to commute with $Z^{\mathcal{A}}(\mathcal{Y})(B_1 \to B_2)$ act transitively on the superselection sectors of the composite domain wall.

So far, we have identified the algebra of local operators which can be analyzed to determine superselection sectors of the composite domain wall. However, we have not yet explained how to characterize the indecomposable domain wall in each sector, or computed any concrete examples; that will be the work of the following sections. In the next section, we will flesh out our particle mobility perspective on domain walls, and outline how it can be used in conjunction with Theorem 3.22 to characterize the summands of the composite wall. Concrete examples where the minimal projections in $Z^{\mathcal{A}}(\mathcal{Y})(A \to B)$ are computed, along with the Witt equivalences of wall excitations in each summand, will appear in § 5.

# 4 Anyon mobility and tunneling operators

An important property of domain walls between regions with (2+1)D topological order is anyon mobility: which anyons from each bulk are confined to one side of the wall, which can pass through the wall, and the different possibilities for what an anyon can become when crossing the wall. In this section, we will explain how to turn the data of a Witt equivalence of wall excitations into a set of *tunneling channels*, which move an anyon from one side of a domain wall to another. In a (2+1)D bulk region, we have local operators which bring a pair of anyons together to produce a single anyon, which one might well call fusion operators; these all arise as linear combinations of a finite set of operators satisfying an orthogonality condition, which select distinct "fusion channels." We will see below that, in an analogous way, possible tunneling operators are also generated as linear combinations of operators which select "tunneling channels." Indeed we can think of tunneling channels as a special kind of fusion channel, in which two anyons from opposite sides of a domain wall fuse to the vacuum on the wall. We will then apply the results of the previous section to describe sets of tunneling operators through the composition of two domain walls in terms of tunneling operators through the individual domain walls, so that Theorem 3.22 can be used to identify the domain wall present in each superselection sector.

We define tunneling channels as follows.

**Definition 4.1.** Let $_{\mathcal{X}}\mathcal{M}_{\mathcal{Y}}$ be an $\mathcal{A}$-enriched bimodule category, and let $c \in \mathrm{Irr}(Z^{\mathcal{A}}(\mathcal{X}))$ and $d \in \mathrm{Irr}(Z^{\mathcal{A}}(\mathcal{Y}))$ be anyons. A *set of tunneling channels* $\mathcal{T}_{c \to d} = \{T_i\}$ from $c$ to $d$ through the domain wall corresponding to $_{\mathcal{X}}\mathcal{M}_{\mathcal{Y}}$ is a maximal set of orthogonal partial isometries. In other words, a tunneling channel is an operator local to the domain wall and adjacent bulk regions which acts as a partial isometry between the space of states containing a $c$ anyon at a given location in the $Z^{\mathcal{A}}(\mathcal{X})$-bulk, and the space of states containing a $d$ anyon at a given location in the $Z^{\mathcal{A}}(\mathcal{Y})$-bulk, and distinct tunneling channels $T_i$ and $T_j$ satisfy the condition that whenever $j \neq i$, $T_j^{\dagger} T_i = 0$.

We now unpack Definition 4.1. We begin by defining the space of *tunneling operators* as the space of morphisms

$$\mathrm{Hom}_{\mathrm{UFC}^{\mathcal{A}}} \left( \boxed{\substack{\bullet \\ c} \; \mathcal{M}} \longrightarrow \boxed{\mathcal{M} \; \substack{\bullet \\ d}} \right) .$$

Here $c$ and $d$ denote anyon types that are fixed, and we are interested in the space of operators that bring $c$ across the domain wall to give $d$. In fact, this is equivalent to the space of operators which take a $c$ particle from the left bulk, and a $\bar{d}$ particle from the right bulk, annihilating them on the domain wall. However, because $c$ and $\bar{d}$ can each correspond to direct sums of different domain wall excitations, there can be multiple distinct ways in which this

annihilation can occur; in this case there are multiple distinct tunneling channels associated with this annihilation.[35]

A set $\{T_i\}$ of tunneling channels $c \to d$ is a basis for this space of tunneling operators satisfying additional conditions, just as fusion channels are special elements of the space of operators which fuse two anyons. By semisimplicity, any tunneling operator factors as a linear combination of operators which first bring $c$ to the domain wall as a simple wall excitation $m$, and then bring $m$ off the wall as a $d$ particle, as shown below.

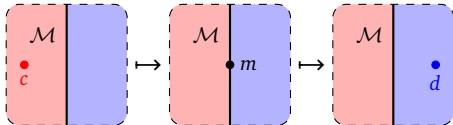

Thus, the space of tunneling operators $c \to d$ is of the form

$$\bigoplus_{m \in \mathrm{Irr}(\mathcal{W})} \mathcal{W}(c \rhd 1_{\mathcal{W}} \to m) \otimes \mathcal{W}(m \to 1_{\mathcal{W}} \lhd d) \cong \bigoplus_{m \in \mathrm{Irr}(\mathcal{W})} M_{n_m^d \times n_m^c}(\mathbb{C}).$$

Here we adopt the notation $\mathcal{W} := \mathrm{End}^{\mathcal{A}}_{\mathcal{X}-\mathcal{Y}}(\mathcal{M})$ for brevity, $c \rhd 1_{\mathcal{W}}$ and $1_{\mathcal{W}} \lhd d$ denote the direct sums of wall excitations obtained by bringing the anyon $c$ or $d$ to the domain wall, and $n_m^c$ and $n_m^d$ are the multiplicities of $m$ as a summand of $c \rhd 1_{\mathcal{W}}$ and $1_{\mathcal{W}} \lhd d$. The maximal set of partial isometries in $\bigoplus_{m \in \mathrm{Irr}(\mathcal{W})} M_{n_m^d \times n_m^c}(\mathbb{C})$ is, up to a unitary change of basis, $\{e_{i,j}^m\}$, where all entries of $e_{i,j}^m$ are 0 except the $i - j$ entry of the $m$ summand. Thus, by specifying that tunneling channels must be partial isometries, Definition 4.1 yields tunneling channels which factor through a single $m \in \mathrm{Irr}(\mathcal{W})$, rather than a direct sum of multiple wall excitation types $m$. Evidently, a set of tunneling channels $c \to d$ is mapped to a set of tunneling channels $d \to c$ by †.

**Remark 4.2.** By dualizability, the space of tunneling operators above is isomorphic to

$$\mathrm{Hom}_{\mathsf{UFC}^{\mathcal{A}}} \left( \begin{array}{c} \underset{c}{\bullet} \mathcal{M} \end{array} \longrightarrow \begin{array}{c} \underset{d}{\bullet} \end{array} \right).$$

This second process is directly analogous to the process which 'injects a droplet' and then selects the $d$ anyon [79, §5].

Tunneling channels to the vacuum have a special interpretation: a particle which has a tunneling channel to the vacuum can condense on the domain wall. Moreover, in the setting of Theorem 3.22, the set of projections in $(Z^{\mathcal{A}}(\mathcal{Y})(B_1 \to B_2), \star)$ onto superselection sectors of the composition of two domain walls is exactly the set of tunneling channels from the vacuum to the vacuum across the composite domain wall!

As we will illustrate below, the set of possible tunneling channels is uniquely fixed (up to a unitary change of basis for each type of simple wall excitation) by the choice of bulk topological orders and the topological domain wall, and does not depend on non-universal details of the boundary conditions. In other words, different choices of $\mathcal{A}$-enriched fusion categories, enriched bimodules, and anomaly $\mathcal{A}$ which produce equivalent UMTCs of bulk exictations and Witt equivalences of wall excitations will also give rise to equivalent sets of tunneling channels. To see this, observe that because a set of tunneling channels through $\mathcal{M}$ is a set of morphisms in the Witt equivalence $\mathrm{End}^{\mathcal{A}}_{Z^{\mathcal{A}}(\mathcal{X})-Z^{\mathcal{A}}(\mathcal{Y})}(\mathcal{M})$, defined in terms of the

---

[35]It is also possible that $c$ or $\overline{d}$ could decompose into wall excitations with non-trivial multiplicities, introducing another source for multiple tunneling channels.

actions of $Z^{\mathcal{A}}(\mathcal{X})$ and $Z^{\mathcal{A}}(\mathcal{Y})$, the possible sets of tunneling channels depend only on the Witt equivalence $\mathrm{End}^{\mathcal{A}}_{Z^{\mathcal{A}}(\mathcal{X})-Z^{\mathcal{A}}(\mathcal{Y})}(\mathcal{M})$.

Next, we turn to the question of how to compose tunneling operators across parallel domain walls. If $_{\mathcal{X}}\mathcal{M}_{\mathcal{Y}}$ and $_{\mathcal{Y}}\mathcal{N}_{\mathcal{Z}}$ are two $\mathcal{A}$-enriched bimodule categories, with $c \in \mathrm{Irr}(Z^{\mathcal{A}}(\mathcal{X}))$, $d \in \mathrm{Irr}(Z^{\mathcal{A}}(\mathcal{Y}))$, and $e \in \mathrm{Irr}(Z^{\mathcal{A}}(\mathcal{Z}))$, tunneling operators $T$ from $c$ to $d$ and $S$ from $d$ to $e$ can be concatenated to obtain tunneling operators $c \to e$:

$$\tag{22}$$

Notice that this operation involves all levels of our 3-category of topological order $\mathrm{UFC}^{\mathcal{A}}$. We will discuss this composition of tunneling operators further in § 4.3.

In § 4.2 we identify *elementary* tunneling channels, i.e. the tunneling channels through elementary domain walls. We will see in § 4.3 that sets of tunneling channels through multiple parallel domain walls can be obtained by composing sets of tunneling channels through the individual domain walls. That is, all tunneling operators through a composite domain wall can be obtained as compositions, as in (22), and there is no redundancy among tunneling operators obtained in this way which is not implied by linearity. In particular, this will allow us to characterize tunneling channels through any indecomposable domain wall, using the decomposition into elementary domain walls shown in Figure 8. Finally, in § 4.4, we will describe the interaction between sets of tunneling channels through the composition of two domain walls and the decomposition of two parallel walls into indecomposable summands obtained in Theorem 3.22. This will allow us to understand each summand of a composite domain wall in terms of the fates of anyons near the domain wall. The examples in § 5 will also contain computations of sets of tunneling channels.

## 4.1 A first example of tunneling channels

We begin by discussing a simple example, in which we compare sets of tunneling channels in two models for the same topological domain wall between Abelian topologically ordered phases. Our example illustrates the following important phenomenon: two gapped boundaries between a pair of phases with the same topological order, but different microscopic realizations, give different, but equivalent, sets of tunneling channels.

**Example 4.3.** For a minimal example, we choose a bulk topological order with anyons described by $\mathcal{C} \cong \mathcal{D}(\mathbb{Z}/4)$, the $\mathbb{Z}/4$-toric code. This is realized by a lattice model for the fusion category $\mathcal{X} \cong \mathsf{Hilb}_{\mathsf{fd}}[\mathbb{Z}/4]$, with $\mathbb{C}^4$ spins on each edge of an oriented square lattice, where links in the lattice are oriented upwards and to the right, while those in the dual lattice are oriented downwards and to the right.

The usual lattice model for $\mathbb{Z}/n$-toric code [77] (see also [5, 103]) on a 2D square lattice with $n$-state spin degrees of freedom on each edge is described by the Hamiltonian:

$$H = -\sum_v A_v - \sum_p B_p\,, \tag{23}$$

where

$$A_v = \sum_{k=1}^n \; Z_n^k \; \underset{Z_n^k}{\overset{(Z_n^k)^\dagger}{\rule{0pt}{0pt}\big|_v}} \; (Z_n^k)^\dagger\,, \qquad B_p = \sum_{j=1}^n X_n^j \boxed{\;p\;} (X_n^j)^\dagger \; . \tag{24}$$

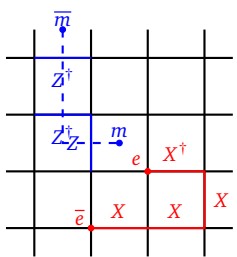

Figure 9: String operators for the particles $e$ (red) and $m$ (blue).

Here $X_n$ and $Z_n$ are $n \times n$ clock matrices which satisfy the relations $X_n Z_n = \omega_n Z_n X_n$, where $\omega_n := e^{2\pi i/n}$ is a primitive $n$-th root of unity. For the case $n = 4$, these have the form:

$$
X_4 = \begin{pmatrix} 0 & 1 & 0 & 0 \\ 0 & 0 & 1 & 0 \\ 0 & 0 & 0 & 1 \\ 1 & 0 & 0 & 0 \end{pmatrix}, \qquad Z_4 = \begin{pmatrix} i & 0 & 0 & 0 \\ 0 & -1 & 0 & 0 \\ 0 & 0 & -i & 0 \\ 0 & 0 & 0 & 1 \end{pmatrix}.
$$

For general $n$, these models have two types of excitations. The first, which we denote $e^k, k = 1, ..., n-1$, are pair-created from the ground state by the string operator $S_e^k(\pi)$. Here $\pi$ is an oriented path on the lattice running from an initial vertex $v_i$ to a final vertex $v_f$. $S_e^k(\pi)$ applies $X^k$ ($X^{-k} = (X^k)^\dagger$) along upward and rightward- (downward and leftward)-oriented edges, creating $e^k$ ($e^{-k}$) at $v_f$ ($v_i$). The resulting state $|\phi(k, v_i, v_f)\rangle$ hosts an $e^k$ particle (anti-particle) at the vertex $v_f$ ($v_i$), with $A_{v_f}|\phi\rangle = \omega_n^k|\phi\rangle, A_{v_i}|\phi\rangle = \omega_n^{n-k}|\phi\rangle$, with $A_v|\phi\rangle = B_p|\phi\rangle = \phi\rangle$ for all $p$, and $v \neq v_i, v_f$.

The second type of excitation, which we denote $m^j$, are pair-created from the ground state by the string operator $S_m^k(\tilde{\pi})$, where $\tilde{\pi}$ is an oriented path in the dual lattice, running from an initial plaquette $p_i$ to a final plaquette $p_f$. The operator $S_m^k(\tilde{\pi})$ applies $Z^j$ ($Z^{-j}$) along downward and rightward- (upward and leftward)-oriented edges, creating $m^k$ ($m^{-k}$) at $p_f$ ($p_i$). The resulting state $|\phi(j, p_i, p_f)\rangle$ obeys $B_{p_f}|\phi\rangle = \omega_n^j|\phi\rangle$, $B_{p_i}|\phi\rangle = \omega_n^{n-j}|\phi\rangle$, and $A_v|\phi\rangle = B_p|\phi\rangle = \phi\rangle$ for all $v$, and $p \neq p_i, p_f$. Sample string operators are shown in Figure 9.

We now consider a lattice with a boundary between $\mathbb{Z}/4$ and $\mathbb{Z}/2$ toric code.



Above, all edges carry $\mathbb{C}^4$ spins, and we change the Hamiltonian to condense $m^2$ on the right hand side, i.e. creating the condensation boundary corresponding to $A = 1 \oplus m^2$. There are multiple different ways to do this. For example, we may redefine the $B_p$ term in the condensed (green) region as

$$
B_q' = \frac{1}{2}\left( 1 + X^2 \begin{array}{c} X^2 \\ \boxed{q} \\ X^2 \end{array} X^2 \right), \tag{25}
$$

on any plaquette $q$ containing at least one green edge, and choose the Hamiltonian to be

$$
H = -\sum_v A_v - \sum_p B_p - \sum_q B_q' - K \sum_\ell C_\ell \,,
$$

with

$$C_\ell = \frac{1}{2}\left(1 + Z_\ell^2\right)$$

on any green edge. Notice that $[C_\ell, B_p] = 0$, so this Hamiltonian is frustration-free; for $K > 0$ its ground state is a simultaneous eigenstate of all $A_v, B_p$ and $C_\ell$ operators with eigenvalue 1. Since $Z_\ell^2$ is the operator which creates a pair of $m^2$ particles on the two plaquettes adjacent to $\ell$, in this ground state, $m^2$ is condensed on any plaquettes with at least one green edge. Moreover, $C_\ell$ has the effect of confining $e$ and $e^3$ particles: since $[Z^2, X] \neq 0$, the corresponding string operators incur a finite energy cost per unit length. (All other quasiparticle string operators commute with $C_\ell$.) Here we consider the limit $K \gg 1$, where the confinement scale is very short, and such $C_\ell$- violating terms do not enter into the low energy physics.

In this case, we can tunnel an $m$ particle into the green region using the usual string operator $S_m^k(\tilde{\pi})$. On the green links, $(1 + Z_\ell^2)$ acts as the identity on states in the low-energy Hilbert space, and this string operator becomes equivalent to the string operator creating $mA \cong m \oplus m^3$. More generally, since $Z - Z^3 = Z(1 - Z^2)$ and $(1 - Z^2)C_\ell = 0$, any operator $T = \alpha(Z - Z^3) + \beta(Z + Z^3)$ in the linear span of $\{Z, Z^3\}$ acts in this way on the low-energy Hilbert space. Thus $T = Z + Z^3$ is the unique choice of string operator on the green links, up to a scalar. In other words, the set of tunneling channels $m \to mA$ contains only one element, $T$.

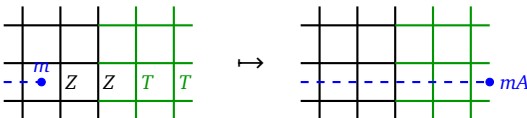

To tunnel from the condensed (green) region to the left hand side, we must bring the particle $mA \cong m \oplus m^3$ across a vertex from a green link to a black one using an operator that commutes with all terms in the Hamiltonian. Since applying $Z^2$ to any green edge commutes with the Hamiltonian, there are two choices for the resulting anyon, corresponding to tunneling operators from $mA \cong m \oplus m^3 \to m$ and from $mA \to m^3$, which differ by an application of $Z^2$ on all links the particles crosses after exiting the green region. The resulting operators are:

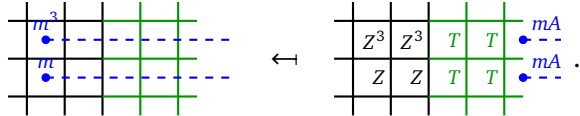

Since the space of tunneling operators in each case is 1-dimensional, each of these choices is unique up to a scalar. In other words, there are two 1-element sets of tunneling channels: one from $mA \to m$, and one from $mA \to m^3$.

One may also describe an antiferromagnetic condensed region, by instead choosing[36]

$$C'_\ell = \frac{1}{2}\left(1 - Z_\ell^2\right).$$

The two choices of $C_\ell$ above are orthogonal projections, with $C_\ell + C'_\ell = 1$. The eigenspaces of each $C_\ell$ can exchanged by applying $X$ or $X^3$ at $\ell$. Since $(1 + Z^2)C_\ell = 0$, the string operator acting on green edges is now $T = Z - Z^3$. Again, we obtain two single-element sets of tunneling channels, one from $m \to mA$, and one from $mA \to m$, and $mA \to m^3$, each unique up to phase.

The analysis of tunneling operators and channels for other anyons is completely analogous. For each anyon $c \in \mathrm{Irr}(\mathcal{D}(\mathbb{Z}/4))$, there are a unique (up to phase) tunneling channels $c \to cA \to c$, and $c \to cA \to cm^2$. Tunneling operators $cA \to c$ and $cA \to cm^2$ again differ by applying the operator $Z^2$ on a string of black edges in the dual lattice.

---

[36]A more general discussion of antiferromagnetic couplings, including the relationship to $Z(\mathsf{Ising})$ topological order, appears in [118].

In string-net models for (2+1)D topological orders [79, 99, 103, 104], *string operators* create pairs of excitations, while *hopping operators* [65] move anyons adiabatically around the system; in Abelian models, these operators coincide. The present discussion illustrates the general fact that in string net models, tunneling operators are closely related to the hopping operators which move an existing excitation from one place to another. Indeed, tunneling channels can be viewed as hopping operators which end on different sides of a domain wall; similarly the set of tunneling channels $c \to c$ through the trivial domain wall contains a single element, the hopping operator for $c$. In the example at hand, the operator $T$ applied at the boundary of the green and black regions is also simply the hopping operator for an $mA$ excitation in the green bulk region; this coincidence is an artifact of working with Abelian anyons.

## 4.2  Elementary tunneling channels

In this section, we will work out sets of elementary tunneling channels, i.e. sets of tunneling channels through elementary domain walls. Tunneling operators through an invertible domain wall are simple to understand, because locally, an invertible domain wall does nothing more than relabel anyons. By Remark 4.2, we see that tunneling operators $c \to d$ through an invertible domain wall $\mathcal{M}$ which applies the braided tensor autoequivalence $\Phi : \mathcal{C} \to \mathcal{D}$ live in a space isomorphic to the hom space (see Footnote 31) $\mathcal{D}(\Phi(c) \to d)$, which is either 0 or 1-dimensional, depending on whether $\Phi(c)$ and $d$ are the same anyon type. In other words, surrounding an anyon by $\mathcal{M}$ to create a 'droplet' amounts to applying the functor $\Phi$, and droplets can be freely attached to and detached from the domain wall. Thus,



and there is a unique choice of tunneling channel $c \to \Phi(c)$ up to a phase.

Next, we will consider the elementary tunneling channels through a domain wall corresponding to a condensation. We begin with domain walls obtained by condensation, as in Example 3.2. Suppose we start with $\mathcal{C}$ topological order and condense the algebra $A \in \mathcal{C}$, so that wall excitations are described by the Witt equivalence $\mathcal{C}_A$ from $\mathcal{C}$ to $\mathcal{C}_A^{\mathrm{loc}}$. On the domain wall, particles can fuse freely with the $A$ condensate by local operators. If $M_A^\circ \in \mathrm{Irr}(\mathcal{C}_A^{\mathrm{loc}})$ is an anyon in the condensed phase,[37] then the elementary tunneling channels can be chosen from the set of morphisms which take $c$ through the wall to become $M_A^\circ$:

$$\left\{ v \in \mathcal{C}_A(cA_A \to M_A^\circ) \,\middle|\, vv^\dagger = \mathrm{id}_{M_A^\circ} \right\}. \tag{26}$$

Here, $\mathcal{C}_A(cA_A \to M_A^\circ)$ is a hom-space[31] in the fusion category $\mathcal{C}_A$. This set spans the space of tunneling operators, but is overcomplete as a basis, because not all the elements are orthogonal. Thus, a set of tunneling channels is just a maximal subset $\{v_i\}$ of the set (26) satisfying $v_j^\dagger v_i = 0$ for $j \neq i$.

Similarly, bringing a particle $M_A^\circ \in \mathrm{Irr}(\mathcal{C}_A^{\mathrm{loc}})$ out of the condensate corresponds to applying the functor $\mathcal{C}_A^{\mathrm{loc}} \to \mathcal{C} : M_A^\circ \mapsto M$ which forgets the $A$ action. The resulting bulk excitation in the uncondensed region can in general be viewed as a direct sum of different anyon types in $\mathcal{C}$; because this direct sum is not a simple object, we denote it with a capital letter. An elementary

---

[37]Here, the anyon $M$ is given the subscript $A$ to remind the reader that $M$ has a right $A$-action. Similarly, the unit of $\mathcal{C}_A^{\mathrm{loc}}$ is denoted by $A_A^\circ$. Even though $M_A$ is an anyon, we denote it by an uppercase letter, to remind ourselves that the underlying collection of anyons $M$ can contain more than one summand.

tunneling channel which brings $M_A^\circ$ across the domain wall to become a particular anyon $c$ in this direct sum corresponds to a choice

$$\left\{ w \in \mathcal{C}(M \to c) \middle| ww^\dagger = \mathrm{id}_c \right\}. \tag{27}$$

Again, a set of tunneling channels is just a maximal orthogonal set of members of (27).

In order to see how the choices of operators $v$ and $w$ in equations (26) and (27) are related explicitly, recall that the free module functor is adjoint to the forgetful functor $\mathcal{C}_A \to \mathcal{C}$, which forgets about $A$-module structures, taking an $A$-module $M_A$ to the object $M \in \mathcal{C}$. (Here, we drop the $\circ$, since the fact that $M_A^\circ$ is a *local* $A$-module is not needed.) This means that we have isomorphisms of vector spaces

$$\mathcal{C}_A(cA_A \to M_A) \cong \mathcal{C}(c \to M) \cong \mathcal{C}(M \to c), \tag{28}$$

where the first isomorphism is from [83, Fig. 4] and the second isomorphism is the antilinear $\dagger$ operation.

Because $c \in \mathrm{Irr}(\mathcal{C})$ and $M_A^\circ \in \mathrm{Irr}(\mathcal{C}_A^{\mathrm{loc}})$ are simple objects, the isomorphisms (28) allows us identify the possible choices of $v$ in (26) with those of $w$ in (27), up to some strictly positive scalar depending on $c, A \in \mathcal{C}$ and $M_A^\circ \in \mathcal{C}_A^{\mathrm{loc}}$. In other words, sets of tunneling channels from $c$ to $M_A^\circ$ or from $M_A^\circ$ to $c$ correspond to orthonormal bases of $\mathcal{C}(c \to M)$ and $\mathcal{C}(M \to c)$ respectively. Notice that these general statements about tunneling channels through an elementary domain wall with condensation match with our observations in Example 4.3.

**Remark 4.4.** We have seen that if $c$ and $d$ are equivalent after fusing with the condensate $A$, then a $c$ particle can enter the condensed region, and return as a $d$ particle. A natural related question is whether, in the case where $c$ splits to $\sum_i (M_i)_A$ in the condensed region, particles from the condensed region can enter the uncondensed region and return as a different anyon type. In particular, if $A \in \mathcal{C}$ is a condensable algebra in a UMTC, and $c \in \mathrm{Irr}(\mathcal{C})$ is an anyon in the uncondensed region, and $M_A$ and $M_A'$ are distinct anyon types which appear as summands of $cA_A$, one might suspect that an $M_A$-particle can be brought across the domain wall as $c$, and then back into the condensate as $M_A'$.

It turns out that this is not the case; in the absence of other excitations, $M_A$ can only tunnel through the uncondensed region and back to become $M_A$. This is because a nonzero operator which took $M_A$ to $M_A'$ could be deformed smoothly into a local operator on the boundary, where wall excitations are given by $\mathcal{C}_A$. But there is no non-zero operator local near the wall that can turn $M_A$ into $M_A'$, since they are distinct simple objects in the Witt equivalence of wall excitations.

However, it is sometimes possible for $M_A$ to enter the uncondensed region as $c$, braid around an anyon $c'$ which is confined by the wall, and return as $M_A'$. We will explore this in detail at the end of § 4.4 below.

## 4.3 Composition of tunneling operators

Having described elementary tunneling operators, we now discuss how one composes tunneling operators through a composite domain wall. As an application, we can characterize tunneling channels through an indecomposable domain wall using its horizontal decomposition from Figure 8. The main result of this section is Proposition 4.5, which says that a set of tunneling channels through the composition of two domain walls can be obtained by composing the members of sets of tunneling channels through each wall.

**Proposition 4.5.** *Suppose* $_\mathcal{X}\mathcal{M}_\mathcal{Y}$ *and* $_\mathcal{Y}\mathcal{N}_\mathcal{Z}$ *are* $\mathcal{A}$-*enriched bimodules. For each* $c \in \mathrm{Irr}(Z^\mathcal{A}(\mathcal{X}))$, $d \in \mathrm{Irr}(Z^\mathcal{A}(\mathcal{Y}))$, *and* $e \in \mathrm{Irr}(Z^\mathcal{A}(\mathcal{Z}))$, *let* $T_{c \to d}$ *be a set of tunneling channels* $c \to d$, *and* $T_{d \to e}$ *be a set of tunneling channels* $d \to e$. *Then the set*

$$\cup_d \left\{ (\mathrm{id}_\mathcal{M} \boxtimes u) \circ (v \boxtimes \mathrm{id}_\mathcal{N}) \middle| u \in T_{d \to e}, v \in T_{c \to d} \right\} \tag{29}$$

*of horizontal composites of tunneling channels from $T_{c\to d}$ and $T_{d\to e}$ is a set of tunneling channels $c \to e$ across $_{\mathcal{X}}\mathcal{M} \boxtimes_{\mathcal{Y}} \mathcal{N}_{\mathcal{Z}}$.*

*Proof.* By 3-dualizability, i.e. by two applications of Remark 4.2, we have an isomorphism from the hom-space

$$\mathrm{Hom}_{\mathrm{UFC}^{\mathcal{A}}}\left( \quad \longrightarrow \quad \right),$$

to the hom-space

$$\mathrm{Hom}\left( \quad \longrightarrow \quad \right). \tag{30}$$

As this latter hom-space (30) is contained in the UMTC $\mathcal{Z}^{\mathcal{A}}(\mathcal{Y})$ which is semisimple, any morphism in this hom-space factors through a simple object/anyon in $Z^{\mathcal{A}}(\mathcal{Y})$. This fact has the following two important consequences:

(1) composites of nonzero morphisms in the hom-spaces

$$\mathrm{Hom}\left( \quad \longrightarrow \quad \right), \tag{31}$$

and

$$\mathrm{Hom}\left( \quad \longrightarrow \quad \right) \tag{32}$$

are non-zero, and

(2) the dimension of the hom-space (30) is the product of the dimensions of the two hom-spaces (31) and (32).

By (1) above, the elements of (29) are non-zero partial isometries, and by (2) above, they are a complete set of tunneling operators $c \to e$ across $_{\mathcal{X}}\mathcal{M} \boxtimes_{\mathcal{Y}} \mathcal{N}_{\mathcal{Z}}$, as claimed. $\qquad\square$

**Example 4.6.** We now use this proposition to characterize tunneling operators through an indecomposable domain wall by considering the domain wall as a concatenation of elementary domain walls from § 3.1.

Let $\mathcal{X}$ and $\mathcal{Y}$ be $\mathcal{A}$-enriched fusion categories, with $\mathcal{M}$ an $\mathcal{A}$-enriched $\mathcal{X}-\mathcal{Y}$ bimodule category corresponding to the Lagrangian algebra $L(A, B, \Phi)$. Then $\mathcal{M}$ is the composition of three boundaries: the condensation boundary which condenses $A$, i.e. the $\mathcal{X}-\mathcal{X}_A$ bimodule $\mathcal{X}_A$, a bimodule category $\mathcal{I}$ which is a Morita equivalence between $\mathcal{X}_A$ and $_B\mathcal{Y}$, and the condensation boundary which condensed $B$, which is the $_B\mathcal{Y}-\mathcal{Y}$ bimodule $_B\mathcal{Y}$. Categories of excitations on the three boundaries are $Z^{\mathcal{A}}(\mathcal{X})_A$, $Z^{\mathcal{A}}(\mathcal{X})_A^{\mathrm{loc}} \cong Z^{\mathcal{A}}(\mathcal{Y})_B^{\mathrm{loc}}$, and $_B Z^{\mathcal{A}}(\mathcal{Y})$, respectively.

Suppose $c \in \mathrm{Irr}(Z^{\mathcal{A}}(\mathcal{X}))$ and $d \in \mathrm{Irr}(Z^{\mathcal{A}}(\mathcal{Y}))$ are two anyon types. Tunneling operators across the three individual boundaries were explained in § 4.2 above. For a local module $M_A^{\circ} \in \mathrm{Irr}(Z^{\mathcal{A}}(\mathcal{X})_A^{\mathrm{loc}})$, tunneling operators across the condensation boundary for $A$ taking $c \to M_A^{\circ}$ are morphisms in $Z^{\mathcal{A}}(\mathcal{X})_A(cA_A \to M_A)$; similarly, for $N_B^{\circ} \in \mathrm{Irr}(Z^{\mathcal{A}}(\mathcal{Y})_B^{\mathrm{loc}})$, tunneling operators across the condensation boundary for $B$ taking $N_B^{\circ} \to d$ are morphisms in $Z^{\mathcal{A}}(\mathcal{Y})_B(N_B \to dB_B)$. Finally, there are unique tunneling channels, and 1-dimensional spaces

of tunneling operators, $M_A^\circ \to \Phi(M_A^\circ)$ across the invertible middle domain wall. Thus, we see that the spaces of tunneling operators $c \to d$ across the indecomposable domain wall $\mathcal{M}$ are

$$\bigoplus_{M_A^\circ \in \mathrm{Irr}(Z^{\mathcal{A}}(\mathcal{X})_A^{\mathrm{loc}})} Z^{\mathcal{A}}(\mathcal{X})_A(cA_A \to M_A) \otimes Z^{\mathcal{A}}(\mathcal{Y})_B(\Phi(M_A) \to dB_B)$$

$$\cong Z^{\mathcal{A}}(\mathcal{Y})_B^{\mathrm{loc}}(\Phi(\ell(cA_A)) \to \ell(dB_B)),$$

where $\ell(\cdot)$ denotes the local part of a module.

Moreover, the tunneling channels $c \to d$ have a straightforward description, as the composite of tunneling channels across all three boundaries. The choice of a tunneling channel $c \to M_A^\circ$ corresponds to choosing an isometry $c \to M$, i.e. including $c$ as a summand of the direct sum $M$. There is a unique channel across the middle invertible boundary, which relabels the anyon $M_A^\circ$ as $\Phi(M_A^\circ)$. Finally, picking a tunneling channel $\Phi(M_A^\circ) \to d$ amounts to choosing a coisometry $\Phi(M_A^\circ) \to d$, i.e. projecting onto a particular summand of $\Phi(M_A^\circ)$ which is of type $d$. (Here, we forget the object $\Phi(M_A^\circ)$ into $\mathcal{D}$ when choosing a coisometry.)

## 4.4 Tunneling operators and the decomposition of parallel domain walls

In addition to Example 4.6, we can also consider the case of a domain wall between two bulk topological orders, where each is obtained by condensing a third topological order. This is the context in which gapped boundaries were discussed in [85, § 3.2]. In other words, given a bulk topological order with UMTC $\mathcal{C}$ and two condensable algebras $A, B \in \mathcal{C}$, one can obtain a Witt equivalence between $\mathcal{C}_A^{\mathrm{loc}}$ and $\mathcal{C}_B^{\mathrm{loc}}$ by composing the Witt equivalences $\mathcal{C}_A^{\mathrm{loc}} \to \mathcal{C} \to \mathcal{C}_B^{\mathrm{loc}}$. In terms of enriched fusion categories, if $\mathcal{C} \cong Z^{\mathcal{A}}(\mathcal{X})$, then one composes the $_A\mathcal{X} - \mathcal{X}$ bimodule $_A\mathcal{X}$ and the $\mathcal{X} - \mathcal{X}_B$ bimodule $\mathcal{X}_B$. Here, we will see that methods similar to those used in proving Theorem 3.22 give a method to explicitly compute the tunneling operators for each summand of a composite domain wall. We will carry out these computations in several examples in § 5 below.

**Remark 4.7.** One reason to look specifically at examples involving the horizontal composition $_A\mathcal{X} \boxtimes_{\mathcal{X}} \mathcal{X}_B$ is that, according to Theorem 3.22, the ground state degeneracy which leads to parallel domain walls decomposing into superselection sectors depends only on the algebras of anyons from the middle bulk which condense on each domain wall. Therefore, this special case already captures all the possible complexity of the interaction between the decomposition of a composite domain wall into superselection sectors and the composition of tunneling operators across the individual domain walls to give tunneling operators across the composite wall. If we want to analyze the composition of an arbitrary pair of indecomposable domain walls, we only need to compose a wall of the form $_A\mathcal{X} \boxtimes_{\mathcal{X}} \mathcal{X}_B$ with other elementary domain walls in a way that does not contribute additional degeneracy.

By Proposition 4.5, if $M_A^\circ \in \mathrm{Irr}(Z^{\mathcal{A}}(\mathcal{X})_A^{\mathrm{loc}})$ and $N_B^\circ \in \mathrm{Irr}(Z^{\mathcal{A}}(\mathcal{X})_B^{\mathrm{loc}})$, then tunneling channels $M_A^\circ \to N_B^\circ$ across the composite domain wall are compositions of tunneling channels $M_A^\circ \to c$ and $c \to N_B^\circ$ for $c \in \mathrm{Irr}(Z^{\mathcal{A}}(\mathcal{X}))$. The space of such compositions is just

$$\bigoplus_{c \in \mathrm{Irr}(Z^{\mathcal{A}}(\mathcal{X}))} Z^{\mathcal{A}}(\mathcal{X})(M \to c) \otimes Z^{\mathcal{A}}(\mathcal{X})(c \to N), \tag{33}$$

or (by composing the two tensor factors) $Z^{\mathcal{A}}(\mathcal{X})(M \to N)$. Note the similarity to (17), which described operators local to the $\mathcal{X}$ region, connecting left and right condensates. Here, the condensates have been replaced with modules over those condensates, because we wish to consider processes that move non-trivial anyons from $_A\mathcal{X}$ to $\mathcal{X}_B$. An immediate consequence is that, if an anyon $c \in \mathrm{Irr}(Z^{\mathcal{A}}(\mathcal{X}))$ splits at one or both domain walls so that $cA_A$ or $cB_B$

has multiple local summands, then there are nonzero tunneling operators taking any local summand of $cA_A$ to any local summand of $cB_B$.

On the other hand, as we saw in Theorem 3.22, composite domain walls of this form are in general decomposable, with superselection sectors corresponding to minimal projections in the algebra $(Z^{\mathcal{A}}(\mathcal{X})(A \to B), \star)$. From Definition 4.1, it is clear that a set of tunneling channels through a direct sum of domain walls $\mathcal{W} = \bigoplus_i \mathcal{W}_i$ is a disjoint union of sets of tunneling channels for each summand $\mathcal{W}_i$. We would therefore like to determine which tunneling channels correspond to each superselection sector.

In order to do so, we observe that the space of tunneling operators $Z^{\mathcal{A}}(\mathcal{X})(M \to N)$ carries an action of the algebra $(Z^{\mathcal{A}}(\mathcal{X})(A \to B), \star)$ (see Definition 3.14), given by

$$
\begin{array}{ccc}
\boxed{f}\begin{smallmatrix}N\\\\M\end{smallmatrix} & \lhd & \boxed{\phi}\begin{smallmatrix}B\\\\A\end{smallmatrix}
\end{array}
:=
\quad
\boxed{f}\,\boxed{\phi}
\tag{34}
$$

Here, $\lhd$ indicates that $\phi$ (an element of the algebra of short string operators) acts on the tunneling operator $f$. Note that (34) makes sense for any $A$- and $B$-modules $M_A$ and $N_B$, not just local modules.

We now give a physical interpretation of the action (34). If $Z^{\mathcal{A}}(\mathcal{X})(M \to N)$ is nonzero, then there is some $c \in \mathrm{Irr}(Z^{\mathcal{A}}(\mathcal{X}))$ such that $M_A$ is a summand of $cA_A$ and $N_B$ is a summand of $cB_B$. Morphisms in $Z^{\mathcal{A}}(\mathcal{X})(M \to N)$ correspond to operators of the form

$$
\tag{35}
$$

If $M_A^\circ \in \mathrm{Irr}(Z^{\mathcal{A}}(_A\mathcal{X})^{\mathrm{loc}})$ and $N_B^\circ \in \mathrm{Irr}(Z^{cA}(\mathcal{X}_B)^{\mathrm{loc}})$, then $Z^{\mathcal{A}}(\mathcal{X})(M \to N)$ is the space of tunneling operators $M_A \to N_B$ across the composite domain wall. Indeed, if $M_A^\circ$ is simple, then $M_A$ is simple (though in general there are multiple choices for $c$), so we can deform (35) to obtain the following

Indeed, as in (17), operators of the form (35) are determined by a choice of anyon $c$ in the middle bulk $Z^{\mathcal{A}}(\mathcal{X})$ topological order, an operator in $Z^{\mathcal{A}}(\mathcal{X})(M, c)$ bringing the $M_A$ excitation out of the left region of condensate as $c$, and an operator in $Z^{\mathcal{A}}(\mathcal{X})(c, N)$ bringing $c$ into the right region of condensate as $N_B^\circ$. Thus, the overall space of such operators is $Z^{\mathcal{A}}(\mathcal{X})(M \to N)$, as it was in (33). Recall that in § 3.3, we saw that the algebra $(Z^{\mathcal{A}}(\mathcal{X})(A \to B), \star)$ was spanned by local operators of the form

$$
,
$$

where $c \in \mathrm{Irr}(Z^{\mathcal{A}}(\mathcal{X}))$ appears as a summand of both $A$ and $B$. The action shown in (34) simply involves applying these two kinds of operators in parallel, as shown below

$$
\tag{36}
$$

Since applying operators in parallel was also how we obtained the multiplication $\star$ on $Z^{\mathcal{A}}(\mathcal{X})(A \to B)$, this is clearly an algebra action.

We now introduce one more computational tool. In § 5, we will consider the case $A = B$, i.e. where a strip where the condensable algebra $A \in Z^{\mathcal{A}}(\mathcal{X})$ is *not* condensed is considered as a domain wall between two regions where $A$ is condensed, which have $Z^{\mathcal{A}}(\mathcal{X})_A^{\mathrm{loc}}$ topological order. We will see that the trivial domain wall between two regions with $Z^{\mathcal{A}}(\mathcal{X})_A^{\mathrm{loc}}$ topological order always appears as a summand, corresponding to the identity morphism $\mathrm{id}_A \in Z^{\mathcal{A}}(\mathcal{X})(A \to A)$, which is a projection (but not the identity) for the convolution product $\star$. If $M_A^\circ, N_A^\circ \in \mathrm{Irr}(Z^{\mathcal{A}}(\mathcal{X})_A^{\mathrm{loc}})$ are anyons, then tunneling operators $M_A^\circ \to N_A^\circ$ through the trivial summand corresponding to the projection $\mathrm{id}_A$ are just the subspace $Z^{\mathcal{A}}(\mathcal{X})_A(M_A \to N_A) \subseteq Z^{\mathcal{A}}(\mathcal{X})(M \to N)$, by the following lemma.

**Lemma 4.8.** *Suppose $M_A, N_A \in Z^{\mathcal{A}}(\mathcal{X})_A$. Then $f \in Z^{\mathcal{A}}(\mathcal{X})(M \to N)$ is in $Z^{\mathcal{A}}(\mathcal{X})_A(M_A \to N_A)$ if and only if*

$$f = \boxed{f} = \boxed{f}\,A = f \triangleleft \mathrm{id}_A\,.$$

Note that the condition that $f = f \triangleleft \mathrm{id}_A$ is not trivial, since $\mathrm{id}_A$ is the identity for the composition $\circ$, and not the convolution multiplication $\star$. We leave the routine proof of the lemma as an exercise.

We conclude by exploring the relationship between anyon types in the middle $Z^{\mathcal{A}}(\mathcal{X})$ bulk region and summands of the composite domain wall $_A\mathcal{X} \boxdot_{\mathcal{X}} \mathcal{X}_B$. There is an obvious monoidal[38] functor $Z^{\mathcal{A}}(\mathcal{X}_A) \boxtimes Z^{\mathcal{A}}(\mathcal{X}) \boxtimes Z^{\mathcal{A}}(\mathcal{X}_B) \to \mathrm{End}_{_A\mathcal{X}-\mathcal{X}_B}^{\mathcal{A}}(_A\mathcal{X} \boxdot_{\mathcal{X}} \mathcal{X}_B)$, given by

$$M_A \boxtimes c \boxtimes N_B \mapsto \begin{array}{|c|c|c|} \hline M_A & c & N_B \\ \bullet & \bullet & \bullet \\ _A\mathcal{X} & \mathcal{X} & \mathcal{X}_B \\ \hline \end{array}\,. \tag{37}$$

In Lemma 4.9 below, we will show that this functor is dominant, i.e. the image generates the whole multifusion category $\mathrm{End}_{_A\mathcal{X}-\mathcal{X}_B}^{\mathcal{A}}(_A\mathcal{X} \boxdot_{\mathcal{X}} \mathcal{X}_B)$.

Before proving Lemma 4.9, we explain its physical consequences and applications. We can think of the multifusion category $\mathrm{End}_{_A\mathcal{X}-\mathcal{X}_B}^{\mathcal{A}}(_A\mathcal{X} \boxdot_{\mathcal{X}} \mathcal{X}_B)$ as a matrix of categories of bimodule functors: if

$$_A\mathcal{X} \boxdot_{\mathcal{X}} \mathcal{X}_B \cong \bigoplus_i \mathcal{W}_i\,,$$

then

$$\mathrm{End}_{_A\mathcal{X}-\mathcal{X}_B}^{\mathcal{A}}(_A\mathcal{X} \boxdot_{\mathcal{X}} \mathcal{X}_B) \cong \bigoplus_{i,j} \mathrm{Hom}_{_A\mathcal{X}-\mathcal{X}_B}^{\mathcal{A}}(\mathcal{W}_i \to \mathcal{W}_j)\,.$$

The $i$-th diagonal entry is $\mathrm{End}_{_A\mathcal{X}-\mathcal{X}_B}^{\mathcal{A}}(\mathcal{W}_i)$, the category of wall excitations in the $i$-th superselection sector, while the $i, j$ entry is $\mathrm{Hom}_{_A\mathcal{X}-\mathcal{X}_B}^{\mathcal{A}}(\mathcal{W}_i \to \mathcal{W}_j)$, the category of point defects between the domain walls in the $i$-th and $j$-th sectors. The fact that $\mathrm{End}_{_A\mathcal{X}-\mathcal{X}_B}^{\mathcal{A}}(_A\mathcal{X} \boxdot_{\mathcal{X}} \mathcal{X}_B)$ is a Witt equivalence, and therefore indecomposable as a multifusion category, means that all of these categories of point defects are nonzero; such point defects always exist.

On the other hand, since the superselection sectors that $_A\mathcal{X} \boxdot_{\mathcal{X}} \mathcal{X}_B$ decomposes into are $_A\mathcal{X}-\mathcal{X}_B$ bimodule summands, acting by an anyon from the left and right $Z^{\mathcal{A}}(_A\mathcal{X})$ and $Z^{\mathcal{A}}(\mathcal{X}_B)$ bulk regions cannot change the superselection sector; the image of these categories in

---

[38]This functor does not lift to a braided monoidal functor to $Z(\mathrm{End}_{_A\mathcal{X}-\mathcal{X}_B}^{\mathcal{A}}(_A\mathcal{X} \boxdot_{\mathcal{X}} \mathcal{X}_B))$, because the right action on the left domain wall is a braided monoidal functor $\overline{Z^{\mathcal{A}}(\mathcal{X})} \to \mathrm{End}_{_A\mathcal{X}-\mathcal{X}_B}^{\mathcal{A}}$, rather than being from $Z^{\mathcal{A}}(\mathcal{X})$.

$\text{End}^{\mathcal{A}}_{A\mathcal{X}-\mathcal{X}_B}({}_A\mathcal{X} \boxdot_{\mathcal{X}} \mathcal{X}_B)$ lies on the diagonal. Consequently, all of the off-diagonal entries must come from the image of $Z^{\mathcal{A}}(\mathcal{X})$, the UMTC of excitations in the middle bulk region. In other words, all point defects between the domain walls in distinct superselection sectors are of the form

$$ \text{(figure)} \tag{38} $$

Here, $w_i$ and $w_j$ refer to minimal central projections in $(Z^{\mathcal{A}}(\mathcal{X})(A \to B), \star)$; when $i \neq j$, $c \in \text{Irr}(Z^{\mathcal{A}}(\mathcal{X}))$ is an anyon type in the middle bulk region which becomes confined (in at least one fusion channel) by both condensates $A$ and $B$. The red line labelled by $w_j$ braids under the $c$-string because $w_j$ is applied second, i.e. on states where the $c$ anyon already exists.

To see that these are the only point defects that can separate distinct superselection sectors, observe that if either free module $cA_A$ or $cB_B$ happens to be local, then the point defect $c$ appearing in (38) is deconfined in either the left or the right bulk region, and is therefore a direct sum of anyons in either the left $Z^{\mathcal{A}}({}_A\mathcal{X})$ or right $Z^{\mathcal{A}}(\mathcal{X}_B)$ bulks regions. Such point defects cannot change the domain wall type, and hence only occur when $i = j$. This reflects the fact that only anyon types which are confined by both condensates can braid non-trivially with the short string operators which comprise the algebra algebra $Z^{\mathcal{A}}(\mathcal{Y})(A \to B)$ of operators that preserve the ground state.

Another interpretation of (38) is that, after applying a minimal projection $w_i \in Z^{\mathcal{A}}(\mathcal{X})(A \to B)$ to select a particular superselction sector, an *extended* string operator associated with a confined anyon type $c$ (which is not a local operator) can take the system into other superselection sectors. In other words, the operator:

$$ \text{(figure)} \tag{39} $$

can be nonzero for $i \neq j$. If we place our system on a sphere, as in Remark 3.23, then the extended $c$ string operator can become a closed loop operator. The same can be done on a torus, or any other topology where the middle $Z^{\mathcal{A}}(\mathcal{X})$ strip is closed up to a tube.

We now turn to the mathematical justification for the above description of the relationship between different superselection sectors.

**Lemma 4.9.** *The functor* (37) *is dominant.*

*Proof.* By (15), the canonical functor

$$ \text{End}^{\mathcal{A}}_{A\mathcal{X}-\mathcal{X}}({}_A\mathcal{X}) \underset{Z^{\mathcal{A}}(\mathcal{X})}{\boxdot} \text{End}^{\mathcal{A}}_{\mathcal{X}-\mathcal{X}_B}(\mathcal{X}_B) \to \text{End}^{\mathcal{A}}_{A\mathcal{X}-\mathcal{X}_B}({}_A\mathcal{X} \boxdot_{\mathcal{X}} \mathcal{X}_B), $$

is an equivalence. Hence, we only need to check that the map $Z^{\mathcal{A}}({}_A\mathcal{X}) \boxtimes Z^{\mathcal{A}}(\mathcal{X}) \boxtimes Z^{\mathcal{A}}(\mathcal{X}_B) \to \text{End}^{\mathcal{A}}_{A\mathcal{X}-\mathcal{X}}({}_A\mathcal{X}) \boxdot_{Z^{\mathcal{A}}(\mathcal{X})} \text{End}^{\mathcal{A}}_{\mathcal{X}-\mathcal{X}_B}(\mathcal{X}_B)$ is dominant.

Since the bimodule category ${}_A\mathcal{X}$ is indecomposable, $\text{End}^{\mathcal{A}}_{A\mathcal{X}-\mathcal{X}}({}_A\mathcal{X})$ is a fusion category (rather than multifusion). This means that the forgetful functor $Z(\text{End}^{\mathcal{A}}_{A\mathcal{X}-\mathcal{X}}({}_A\mathcal{X})) \to \text{End}^{\mathcal{A}}_{A\mathcal{X}-\mathcal{X}}({}_A\mathcal{X})$ is a dominant tensor functor. Since $\text{End}^{\mathcal{A}}_{A\mathcal{X}-\mathcal{X}}({}_A\mathcal{X})$ is a Witt equivalence $Z^{\mathcal{A}}({}_A\mathcal{X}) \to Z^{\mathcal{A}}(\mathcal{X})$, the action functor $Z^{\mathcal{A}}({}_A\mathcal{X}) \boxtimes \overline{Z^{\mathcal{A}}(\mathcal{X})} \to Z(\text{End}^{\mathcal{A}}_{A\mathcal{X}-\mathcal{X}}({}_A\mathcal{X}))$ is an equivalence. Composing, we have a dominant tensor functor

$$ Z^{\mathcal{A}}({}_A\mathcal{X}) \boxtimes \overline{Z^{\mathcal{A}}(\mathcal{X})} \to Z(\text{End}^{\mathcal{A}}_{A\mathcal{X}-\mathcal{X}}({}_A\mathcal{X})) \to \text{End}^{\mathcal{A}}_{A\mathcal{X}-\mathcal{X}}({}_A\mathcal{X}). $$

Similarly, the action gives a dominant tensor functor

$$Z^{\mathcal{A}}(\mathcal{X}) \boxtimes \overline{Z^{\mathcal{A}}(\mathcal{X}_B)} \to \mathrm{End}^{\mathcal{A}}_{\mathcal{X}-\mathcal{X}_B}(\mathcal{X}_B).$$

Overall, so far, we have a dominant tensor functor

$$Z^{\mathcal{A}}(_A\mathcal{X}) \boxtimes \overline{Z^{\mathcal{A}}(\mathcal{X})} \boxtimes Z^{\mathcal{A}}(\mathcal{X}) \boxtimes \overline{Z^{\mathcal{A}}(\mathcal{X}_B)} \to \mathrm{End}^{\mathcal{A}}_{_A\mathcal{X}-\mathcal{X}}(_A\mathcal{X}) \boxtimes \mathrm{End}^{\mathcal{A}}_{\mathcal{X}-\mathcal{X}_B}(\mathcal{X}_B).$$

Note that, as tensor categories (i.e. forgetting the braiding), $Z^{\mathcal{A}}(\mathcal{X}) \cong \overline{Z^{\mathcal{A}}(\mathcal{X})}$. Composing with the canonical dominant tensor functor

$$\mathrm{End}^{\mathcal{A}}_{_A\mathcal{X}-\mathcal{X}}(_A\mathcal{X}) \boxtimes \mathrm{End}^{\mathcal{A}}_{\mathcal{X}-\mathcal{X}_B}(\mathcal{X}_B) \to \mathrm{End}^{\mathcal{A}}_{_A\mathcal{X}-\mathcal{X}}(_A\mathcal{X}) \boxdot_{Z^{\mathcal{A}}(\mathcal{X})} \mathrm{End}^{\mathcal{A}}_{\mathcal{X}-\mathcal{X}_B}(\mathcal{X}_B),$$

we obtain a dominant tensor functor which is $Z^{\mathcal{A}}(\mathcal{X})$-balanced, and hence factors through

$$(Z^{\mathcal{A}}(_A\mathcal{X}) \boxtimes \overline{Z^{\mathcal{A}}(\mathcal{X})}) \boxdot_{Z^{\mathcal{A}}(\mathcal{X})} (Z^{\mathcal{A}}(\mathcal{X}) \boxtimes \overline{Z^{\mathcal{A}}(\mathcal{X}_B)}) \cong Z^{\mathcal{A}}(_A\mathcal{X}) \boxtimes Z^{\mathcal{A}}(\mathcal{X}) \boxtimes Z^{\mathcal{A}}(\mathcal{X}_B),$$

completing the proof. □

As another application, we are now in a position to justify our the assertion in Remark 4.4: that, if anyons $M_A^\circ$ and $N_A^\circ$ in $Z^{\mathcal{A}}(_A\mathcal{X}) \cong Z^{\mathcal{A}}(\mathcal{X})_A^{\mathrm{loc}}$ are both summands of the free module $cA_A$ for $c \in \mathrm{Irr}(Z^{\mathcal{A}}(\mathcal{X}))$, i.e. they are both obtained as summands when a $c$ anyon is brought to the domain wall, then there is a nonzero local operator

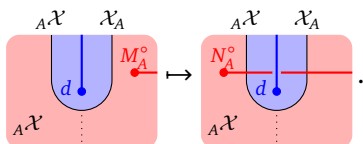

(40)

After some topological deformation, the above picture becomes

Here, the dotted line is the identity/trivial domain wall, which is a superselection sector of the composite domain wall given by the $\mathcal{A}$-enriched $_A\mathcal{X} - {_A\mathcal{X}}$ bimodule $_A\mathcal{X} \boxdot_{\mathcal{X}} {_A\mathcal{X}}$. As we will explain in detail in § 5 below, this identity bimodule is a summand of the composite domain wall, corresponding under Theorem 3.22 to the convolution projection $\mathrm{id}_A \in Z^{\mathcal{A}}(\mathcal{X})(A \to A)$. The blue strip corresponds to another summand of this same composite domain wall. The local operators depicted above are then just (adjoints of) tunneling operators $N_A^\circ \to M_A^\circ$ through this composite domain wall; such tunneling operators always exist by Proposition 4.5. A priori we might worry that some summands of the composite wall could fail to appear in the above diagram. However, because the pair of parallel domain walls are joined by a cup, Lemma 4.9 tells us that all summands will appear, for *some* choice of $d \in \mathrm{Irr}(Z^{\mathcal{A}}(\mathcal{X}))$. On the other hand, when $d \cong 1$, the cup is just the inclusion of the trivial domain wall into the composite, and nonzero tunneling operators through that wall only take $M_A^\circ$ to itself.

# 5  $\mathcal{XYX}$ examples

In this section, we will use the tools developed in § 3 and § 4 to work out the decomposition of composite domain walls in several explicit examples. We will investigate composite domain walls of the form

$$\mathcal{W} := {}_A\mathcal{Y} \boxtimes_\mathcal{Y} \mathcal{Y}_A \cong {}_A\mathcal{Y}_A,$$

where $\mathcal{Y}$ is an $\mathcal{A}$-enriched fusion category and $A \in Z^{\mathcal{A}}(\mathcal{Y})$ is a condensable algebra. For brevity, we denote $\mathcal{C} := Z^{\mathcal{A}}(\mathcal{Y})$, $\mathcal{X} := \mathcal{Y}_A$, and $\mathcal{W} \cong {}_A\mathcal{Y} \boxtimes_\mathcal{Y} \mathcal{Y}_A \cong {}_A\mathcal{Y}_A$ is the $\mathcal{Y}_A - \mathcal{Y}_A$ bimodule category describing the composite domain wall. Thus, $Z^{\mathcal{A}}(\mathcal{X}) \cong Z^{\mathcal{A}}(\mathcal{Y}_A) \cong \mathcal{C}_A^{\text{loc}}$, and the $\mathcal{C}_A^{\text{loc}} - \mathcal{C}_A^{\text{loc}}$ bimodule multifusion category of excitations on the composite domain wall is $\text{End}(\mathcal{W}) \cong {}_A\mathcal{C} \boxminus_\mathcal{C} \mathcal{C}_A \cong {}_A\mathcal{C}_A$, the category of $A-A$ bimodules in $\mathcal{C}$.

We take a moment to discuss our choice to focus on such examples, which we call $\mathcal{XYX}$ *examples*. In the previous section, we have just seen that $\mathcal{XYX}$ examples arise naturally when analyzing operators of the form (40), which are natural to consider whenever a domain wall based on anyon condensation appears. Beyond that, $\mathcal{XYX}$ examples are parameterized by a minimal amount of data: we only need a choice of anomaly representative $\mathcal{A}$, an $\mathcal{A}$-enriched fusion category $\mathcal{Y}$ to determine the uncondensed bulk topological order, and a choice of condensable algebra $A \in Z^{\mathcal{A}}(\mathcal{Y})$. Then, the $\mathcal{A}$-enriched fusion and bimodule categories labelling the regions where $A$ is condensed and the walls bounding those regions are all just $\mathcal{Y}_A$. As explained in Remark 4.7, examples of the form ${}_A\mathcal{X} \boxtimes_\mathcal{X} \mathcal{X}_B$ are already able to illustrate all the interactions between the decomposition of composite domain walls and tunneling operators. In studying $\mathcal{XYX}$-examples, we only impose the further restriction that $B = A$. This class of examples therefore simplifies the computational task ahead, and as we will see, still allows for a variety of interesting behavior.

We now show that, for $\mathcal{XYX}$ examples, the trivial domain wall from $Z^{\mathcal{A}}(\mathcal{X})$ topological order to itself always appears as a summand of the composite domain wall $\mathcal{W}$. This observation will prove useful in the analyses of the examples that follow. To show this, we recall that under the correspondence set out in Theorem 3.22, (indecomposable) summands of $\mathcal{W}$ correspond to (minimal) projections in the commutative algebra $\mathcal{C}(A \to A)$ with the convolution product. However, $\mathcal{C}(A \to A)$ also carries another product: the composition $\circ$ of morphisms in the category $\mathcal{C}$, with identity $\text{id}_A$.

$$\boxed{x} \circ \boxed{y} := \begin{array}{c} \boxed{x} \\ \boxed{y} \end{array} \qquad \boxed{\text{id}_A} = {\Large|}_A \ .$$

That is, if $A \cong \oplus_i x_i$, where each $x_i \in \text{Irr}(\mathcal{C})$ and $p_i : A \to x_i$ is the projection onto the $i$-th summand, then $\text{id}_A = \sum_i p_i^\dagger p_i$; interpreted as short string operators as in (16), $\text{id}_A$ is a sum of operators which bring each channel in $A$ across the strip. Because $A \in Z^{\mathcal{A}}(\mathcal{X})$ is separable, the morphism $\text{id}_A$ is a projection for $\star$:

$$\boxed{\text{id}_A} \star \boxed{\text{id}_A} = \bigcirc = {\Large|}_A = \boxed{\text{id}_A} \ .$$

Since $A$ is connected, it follows directly from $\text{End}_{\mathcal{C}_A}(A_A) = \mathbb{C} \, \text{id}_A$ that $\text{id}_A$ is a minimal projection in $(\mathcal{C}(A \to A), \star)$ (visibly, for any projection $p \leq \text{id}_A$, $p = p \star \text{id}_A \in \text{End}_{\mathcal{C}_A}(A_A)$).

Therefore, its image is a superselection sector denoted $\mathcal{W}_1$ of the composite domain wall $\mathcal{W}$. In order to characterize this sector, we compute the action of $\mathrm{id}_A$ on the spaces of tunneling operators; as described in § 4.4, the space of tunneling operators $M_A^\circ \to N_A^\circ$ through $\mathcal{W}_1$ is just the image of $\mathrm{id}_A$ acting on $\mathcal{C}(M \to N)$ with the action (34). But by Lemma 4.8, the image of $\mathrm{id}_A$ is just $\mathcal{C}_A(M_A \to N_A) \subseteq \mathcal{C}(M \to N)$, showing that $\mathrm{End}_{\mathcal{Y}-\mathcal{Y}}^{\mathcal{A}}(\mathcal{W}_1)$ is just the identity Witt autoequivalence of $\mathcal{C}_A^{\mathrm{loc}}$. Thus, as claimed, for any $\mathcal{X}\mathcal{Y}\mathcal{X}$ composite domain wall, the trivial domain wall appears as a superselection sector, since it is the image $\mathcal{W}_1$ of a minimal projection. We will therefore refer to $\mathcal{W}_1$ as the identity superselection sector.

Furthermore, we see that the inclusions $\mathcal{C}_A(M_A \to N_A) \subseteq \mathcal{C}(M \to N)$ have two natural interpretations in our setting. The first is that $M$ and $N$ are classical mixtures of anyons, which result from bringing a domain wall excitation (in $\mathcal{C}_A$) into the $\mathcal{C}$ bulk. $\mathcal{C}(M \to N)$ is thus the space of topological local operators taking one such mixture to another, and $\mathcal{C}_A(M_A \to N_A)$ is the subspace of these operators which is stable under fusion with the condensate $A$.[39] The second is that $\mathcal{C}(M \to N)$ is the space of tunneling operators $M_A^\circ \to N_A^\circ$ through the composite domain wall, and $\mathcal{C}_A(M_A \to N_A)$ is the subspace of tunneling operators through the identity superselection sector.

Both interpretations will play a role in the examples below. In particular, suppose $M_A = N_A = cA_A$ is a free module, where $c \in \mathrm{Irr}(\mathcal{C})$ is an anyon. Then under the first interpretation, $\mathcal{C}_A(cA_A, cA_A) \subseteq \mathcal{C}(cA, cA)$ contains the morphisms which split the anyon $c$ into indecomposable wall excitations at the domain wall. Under the second interpretation, $\mathcal{C}(cA, cA)$ consists of local operators bringing a $c$ anyon from the left domain wall to the right one, and $\mathcal{C}_A(cA_A, cA_A)$ is again the subspace of those operators supported in the superselection sector $\mathcal{W}_1$ associated to $\mathrm{id}_A$.

**Remark 5.1.** There is an alternative way to see that the identity domain wall must appear as a summand in our $\mathcal{X}\mathcal{Y}\mathcal{X}$-examples, which does not involve the results of § 3 and 4. In our chosen examples, $\mathcal{X} \cong \mathcal{Y}_A$, and the bimodule category labelleing the composite domain wall is $_A\mathcal{Y} \boxtimes_{\mathcal{Y}} \mathcal{Y}_A \cong {_A}\mathcal{Y}_A$, the category of $A-A$-bimodules in $\mathcal{Y}$. As described in Appendix B, the tensor product on $\mathcal{Y}_A$ comes from a monoidal embedding into $_A\mathcal{Y}_A$, and both the left and right actions of $\mathcal{Y}_A$ on $_A\mathcal{Y}_A$ will also come from this embedding. Therefore, the trivial domain wall, corresponding to the identity $\mathcal{Y}_A - \mathcal{Y}_A$-bimodule category $\mathcal{Y}_A$, appears as a summand of $_A\mathcal{Y}_A$.

In the following examples, we will show that the other superselection sectors can exhibit a wide variety of behaviors: they may also be equivalent to the trivial domain wall, as in § 5.1, they can be different invertible domain walls, as in § 5.2 and § 5.3, or they can be noninvertible domain walls, where additional anyons become condensed or confined, as in § 5.4.

For each of the examples below, we use all of the tools developed in the preceding sections to analyze the composite domain wall, according to the following steps.

(1) Specify a UMTC $\mathcal{A}$ (fixing the anomaly), an $\mathcal{A}$-enriched fusion category $\mathcal{Y}$ (§ 2.1), and a condensable algebra $A \in Z^{\mathcal{A}}(\mathcal{Y})$ (§ B). The condensed phase is given by the enriched fusion category $\mathcal{X} = \mathcal{Y}_A$, which has bulk excitations $Z^{\mathcal{A}}(\mathcal{X}) \cong Z^{\mathcal{A}}(\mathcal{Y}_A) \cong Z^{\mathcal{A}}(\mathcal{Y})_A^{\mathrm{loc}}$. In all cases, we also provide references to literature describing a specific lattice model for the domain wall in question.

(2) Compute the convolution algebra $(Z^{\mathcal{A}}(\mathcal{Y})(A \to A), \star)$ of short string operators on the middle strip, where $\star$ is the convolution multiplication from Definition 3.14. Identify the minimal projections in this algebra, which are local operators that project onto superselection sectors of the composite domain wall, by Theorem 3.22.

---

[39]As described in Footnote 31, operators in $\mathcal{C}(M \to N)$ are not very interesting; in general, $\mathcal{C}(M \to N)$ is just a multiplicity space. However, the choice of subspace $\mathcal{C}_A(M_A \to N_A)$ is part of the data specifying the wall excitations $M_A$ and $N_A$.



(3) Identify the summands of any anyons $c \in Z^{\mathcal{A}}(\mathcal{Y})$ from the uncondensed phase which split on the domain wall to the condensed phase, by computing the decomposition of free modules $cA_A$. Identify the spaces of tunneling channels for each pair of anyons through the composite domain wall, or at least give dimensions.

(4) Using sets of tunneling channels (Definition 4.1) for the original domain walls, the decomposition of splitting anyons, and the action (34) of $(Z^{\mathcal{A}}(\mathcal{Y})(A \to A), \star)$ on spaces of tunneling operators, we work out sets of tunneling channels for the indecomposable domain wall in each superselection sector. This allows us to identify the indecomposable domain wall associated to each minimal central projection $w_i \in Z^{\mathcal{A}}(\mathcal{Y})(A \to A)$, either as an indecomposable $\mathcal{X} - \mathcal{X}$ $\mathcal{A}$-enriched bimodule category $\mathcal{W}_i$, the associated Witt equivalence $\mathrm{End}^{\mathcal{A}}_{\mathcal{X}-\mathcal{X}}(\mathcal{W}_i)$, or the Lagrangian algebra in $Z^{\mathcal{A}}(\mathcal{X}) \boxtimes \overline{Z^{\mathcal{A}}(\mathcal{X})}$ which is condensed at the wall.

(5) Based on (38), we briefly describe how different superselection sectors are related by extended string operators which are confined to the uncondensed region, as well as how anyons from the middle bulk become point defects between superselection sectors.

## 5.1 Toric code

(1) In this example, we choose

$$
\begin{aligned}
\mathcal{A} &= \mathsf{Hilb}, \\
\mathcal{X} &= \mathsf{Hilb}[\mathbb{Z}/2], \\
\mathcal{Y} &= \mathsf{Hilb}[\mathbb{Z}/4], \\
{}_{\mathcal{X}}\mathcal{M}_{\mathcal{Y}} &= \mathsf{Hilb}[\mathbb{Z}/2].
\end{aligned}
$$

Setting $A = \mathbb{C}^{\mathbb{Z}/2} \in Z(\mathcal{Y})$ to be the algebra obtained by condensing the boson $m^2$, we obtain $Z(\mathcal{X}) = Z(\mathcal{Y})^{\mathrm{loc}}_A = \mathcal{D}(\mathbb{Z}/2)$, the toric code.

We will use the same Hamiltonian as described above in (23)-(25), but a different configuration of green and black edges, as shown.

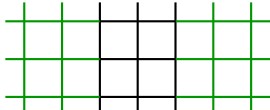

(2) We now analyze the wall $\mathcal{W} := \mathcal{M}$ using Theorem 3.22. As an object in $\mathcal{C} := Z(\mathcal{Y}) \cong \mathcal{D}(\mathbb{Z}/4)$, we have $A \cong 1 \oplus m^2$. Therefore, $\mathcal{C}(A \to A)$ is 2-dimensional, generated by projections $\pi_1 : A \to 1 \to A$ and $\pi_{m^2} : A \to m^2 \to A$ for the usual *categorical composition* in $\mathrm{End}_{Z(\mathcal{Y})}(A)$ (as opposed to the convolution multiplication $\star$ for $\mathrm{End}_{Z(\mathcal{Y})}(A)$). The convolution multiplication $\star$ has identity $2\pi_1$, and $\pi_{m^2} \star \pi_{m^2} = \frac{1}{2}\pi_1$, so that the generators $2\pi_1$ and $2\pi_{m^2}$ form the group $\mathbb{Z}/2$, i.e. $(\mathcal{C}(A \to A), \star) \cong \mathbb{C}[\mathbb{Z}/2]$. (The factors of 2 and $\frac{1}{2}$ appear because $|\mathbb{Z}/2| = 2$, and are necessary to make the condensate $A$ unitarily separable.) The minimal convolution projections in $\mathcal{C}(A \to A)$ are thus

$$
\begin{aligned}
w_1 &:= \mathrm{id}_A = \pi_1 + \pi_{m^2}, \\
w_{-1} &:= \pi_1 - \pi_{m^2}.
\end{aligned}
$$

(3) In this case, the objects of $\mathrm{Irr}(\mathcal{C}_A)$ are all of the form $x \oplus m^2 x$ for $x \in \mathrm{Irr}(\mathcal{C})$, so they partition the objects of $\mathrm{Irr}(\mathcal{C})$. Hence, for $M_A \neq N_A \in \mathrm{Irr}(\mathcal{C}^{\mathrm{loc}}_A)$, we have $\mathcal{C}(M \to N) \cong 0$

meaning that there are no tunneling operators between distinct simple objects. On the other hand, for $M_A \in \mathrm{Irr}(\mathcal{C}_A^{\mathrm{loc}})$, there are 2 tunneling channels $M_A \to M_A$, one for each projection $w_{\pm 1}$.

(4) Since $w_1 = \mathrm{id}_A$, the corresponding summand $\mathcal{W}_1$ of the composite domain wall is equivalent to the trivial domain wall (i.e. no domain wall) from $Z^{\mathcal{A}}(\mathcal{X})$ topological order to itself. The other summand $\mathcal{W}_{-1}$ is also an invertible domain wall, which applies a $\mathbb{Z}_2$ symmetry $\Phi \in \mathrm{Aut}_{\otimes}^{\mathrm{br}}(\mathcal{D}(\mathbb{Z}/2))$ which does not permute anyon types, but under which the $m$ anyon is charged. For a single domain wall, however, the resulting phase cannot be detected physically; hence $\mathcal{W}_{-1}$ must be equivalent to $\mathcal{W}_1$, in the sense that both produce the same Witt autoequivalence of $D(\mathbb{Z}/2)$.

(5) We can understand why there are two distinct summands $\mathcal{W}_1$ and $\mathcal{W}_{-1}$ by examining the ground-state degeneracy obtained from putting our system on a sphere with $A$ uncondensed at the equator and condensed near the poles, as in Remark 3.23. In other words, we impose periodic boundary conditions orthogonal to the domain walls, and then cap off the two bulk regions where $A$ is condensed. In this case, the algebra $\mathcal{C}(A \to A)$ becomes the 2-dimensional space of ground states.

To understand the difference between $\mathcal{W}_1$ and $\mathcal{W}_{-1}$, consider the closed string operator $L_e$ of type $e$ which runs around a non-contractible loop in the middle unshaded region.

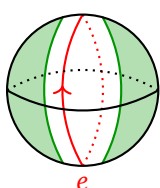

$L_a$ for $a = e^2, m, m^2$, or $m^3$ act trivially on the space of ground states, since these strings can slide into the green region and be contracted. However, $e$ is confined in the green regions; hence $L_e$ need not have a trivial action. Indeed, because $[L_e, \pi_{m^2}] = -1$, we know that $L_e w_1 = w_{-1} L_e$, so $L_e$ exchanges the two summands of $\mathcal{W}$. In summary, the two summands of $\mathcal{W}$ can be distinguished by the number of $e$-lines modulo 2 running around a non-contractible loop in the $\mathcal{C}$-bulk region.

Thus, we see that in the geometry depicted above, the image of the projectors $w_1$ and $w_{-1}$ differs by an extended $e$-string operator in the strip of the original uncondensed phase. Although the two summands exhibit the same behavior in terms of particle mobility, the composite tunneling operators $T_m : mA_A \to m \to mA_A$ and $T_{m^3} : mA_A \to m^3 \to mA_A$ will pick up different phases on each summand, which we denote by $\lambda_{m,1}$, $\lambda_{m^3,1}$, $\lambda_{m,-1}$, and $\lambda_{m^3,-1}$ respectively. The phases themselves are not well defined, since they can be modified by adding strictly local operators in the vicinity of the domain wall. However, we can define a phase that is independent of these details, by first tunneling an $m$ across using $T_m$ (or $T_{m^3}$), then applying the non-local $L_e$ operator, and finally tunneling the particle back using $T_m^{-1}$ (or $T_{m^3}^{-1}$). The net phase acquired in this process is $\lambda_{m,-1}/\lambda_{m,1}$ (or $\lambda_{m^3,-1}/\lambda_{m^3,-1}$), which is $\pm i$. The overall sign is also not well defined, since it can be modified by adding a contractible $L_{e^2}$ string encircling the domain wall. However, the ratios

$$(\lambda_{m,-1})^2/(\lambda_{m,1})^2 = (\lambda_{m^3,-1})^2/(\lambda_{m^3,-1})^2 = -1$$

are well-defined, in the sense that they are independent of any local or contractible operators. This is the sense in which the two boundaries represent different $\mathbb{Z}_2$-symmetry actions on the $m$ particle, which carries a $1/2$ $\mathbb{Z}_2$ charge under one of the symmetry actions.

The process of computing $\lambda_{m,-1}/\lambda_{m,1}$ described above cannot tell us whether the system was initially in the image of $\mathcal{W}_1$, or that of $\mathcal{W}_{-1}$. On the spherical geometry, this is true in general: no physical process can reliably tell us which of the two domain wall types we have, since there is no way to measure the presence of the $L_e$ string that is robust to adding local operators near the domain wall; the physically well-defined information is simply that the two domain wall types are different. This is reflected mathematically by the fact that $\mathcal{W}_1$ and $\mathcal{W}_{-1}$ are equivalent as $\mathcal{X} - \mathcal{X}$ bimodule categories, even though they are different summands of $\mathcal{W}$. On the other hand, on the torus with a single white region around the meridian, the presence of an extended $e$ string in the white region can be detected by a process that transports an $m$ particle around the torus.

**Remark 5.2.** In a given microscopic realization, we can see that we obtain the summand with an odd number of extended $e$-strings parallel to the domain walls in either the image of $w_1$ or of $w_{-1}$, but not both. Recall that when we defined the lattice model, we had to make choice of operator $C_\ell$ to condense $m^2$ in the green region. If we choose $C_\ell = \frac{1}{2}(1+Z)^2$ in both regions, then $w_1$ selects states with an even number of extended $e$-strings, and $w_{-1}$ selects states with an odd number. If we instead choose $C_\ell = \frac{1}{2}(1+Z)^2$ in the left bulk region and $C_l = \frac{1}{2}(1-Z)^2$ in the right bulk region, then the situation is reversed.

Finally, we consider adding an $e$-$e^3$ pair to the uncondensed $\mathcal{D}(\mathbb{Z}/4)$ region. Each of these excitations corresponds to a *nontrivial* point defect separating the two distinct types of domain wall identified above. As above, because there is no physical process that can detect a closed $L_e$ string, a priori the resulting domain wall can either have $\mathcal{W}_1$-type domain wall below the $e$ particle, and a $\mathcal{W}_{-1}$ wall above it, or be in a configuration where the roles of $\mathcal{W}_1$ and $\mathcal{W}_{-1}$ are reversed. Note also that there is no universal meaning to which particle we call $e$ and which we call $e^3$, since this can be altered by bringing a pair of $e^2$ particles from one of the toric code regions, and binding one to each defect.

In the presence of such defects, however, we can see the projective $\mathbb{Z}_2$ symmetry action more explicitly. Tunneling an $m := mA_A$ particle from the condensed $\mathcal{D}(\mathbb{Z}/2)$ toric code in a closed path encircling an $e$ or $e^3$ defect gives a phase of $\phi = \pm i$. The overall sign is not universal, in the sense that it can be modified by binding an $e^2$ particle to the defect. However, the fact that $\phi^2 = -1$ is universal, and cannot be modified by either local operators, or binding anyons from the Toric code bulk to the defects. Thus, we see that the $m$ particle carries a $1/2$ charge under one of the $\mathbb{Z}_2$ symmetries associated with this domain wall.

## 5.2 Doubled Ising

(1) In this example, we choose

$$\begin{aligned}
\mathcal{A} &= \mathsf{Hilb}\,, \\
\mathcal{X} &= \mathsf{Hilb}[\mathbb{Z}/2]\,, \\
\mathcal{Y} &= \mathsf{Ising}\,, \\
{}_{\mathcal{X}}\mathcal{M}_{\mathcal{Y}} &= \mathsf{Hilb}[\mathbb{Z}/2]\,.
\end{aligned}$$

Recall that the Ising UMTC[40] has anyons $1$, $\psi$, and $\sigma$, where $\psi$ is a fermion and $\dim(\sigma) = \sqrt{2}$. Like any modular tensor category, its double $\mathcal{C} := Z(\mathcal{Y}) \cong \mathsf{Ising} \boxtimes \overline{\mathsf{Ising}}$

---

[40]The Ising UMTC (which is distinct from $SU(2)_2$) is the semisimple part of the Temperley-Lieb-Jones category $\mathcal{TLJ}(ie^{-\frac{2\pi i}{16}})$ with braiding (4), pivotal structure given by the identity for all simples, and dagger structure from Footnote 12. The formulas for $S, T$ are given in [124, p. 19]; compare with Footnotes 13 and 14.

consists of two copies with opposite chirality. The particle $\psi \boxtimes \overline{\psi}$ is a boson, and after condensing this boson, i.e. condensing the algebra

$$1 \boxtimes \overline{1} \oplus \psi \boxtimes \overline{\psi} \in \mathsf{Ising} \boxtimes \overline{\mathsf{Ising}} \cong Z(\mathsf{Ising}),$$

we obtain $\mathcal{C}_A^{\mathsf{loc}} \cong Z(\mathcal{X}) = \mathcal{D}(\mathbb{Z}/2)$ [24, 28], the toric code topological order.

A concrete description of the string-net [103] lattice model of $Z(\mathsf{Ising}) \cong \mathsf{Ising} \boxtimes \overline{\mathsf{Ising}}$, including the modifications needed to condense $\psi \boxtimes \overline{\psi}$, appears in [28, 64]. Many mathematical details of the resulting boundary between doubled Ising and $\mathbb{Z}/2$-toric code resulting from the condensation of $A$ appear in [34], and another lattice model of the boundary with detailed analysis appears in [127].

(2) As in § 5.1, the algebra $\mathcal{C}(A \to A)$ is 2-dimensional, and generated by composition projections $\pi_1$ and $\pi_\psi$ corresponding to the simple objects $1 \boxtimes \overline{1}$ and $\psi \boxtimes \overline{\psi}$ respectively. Again, $2\pi_1$ is the identity for $\star$, and $\pi_\psi * \pi_\psi = \frac{1}{2}\pi_1$. The minimal convolution projections are thus

$$w_1 := \mathrm{id}_A = \pi_1 + \pi_\psi,$$
$$w_{-1} := \pi_1 - \pi_\psi.$$

**Remark 5.3.** One might wonder why, despite the fact that the UMTC $\mathcal{C} = Z(\mathsf{Ising})$ and algebra $A$ in this example are different from the UMTC $\mathcal{C}' = \mathcal{D}(\mathbb{Z}/4)$ and algebra $A' \cong 1 \oplus m^2$ from the previous example, the algebras $(\mathcal{C}(A \to A), \star)$ and $(\mathcal{C}'(A' \to A'), \star)$ are isomorphic. This happens because the subcategories generated by each algebra are equivalent; both are equivalent to $\mathsf{Hilb}[\mathbb{Z}/2]$ with the trivial/symmetric braiding. On the level of objects, the equivalence is given by $1 \mapsto 1$ and $m^2 \mapsto \psi \boxtimes \overline{\psi}$, and it indeed maps $A'$ to $A$. However, because the overall UMTCs $\mathcal{C}'$ and $\mathcal{C}$ are different, the remaining details of this example will differ significantly from those in the previous one.

(3) After condensing $\psi \boxtimes \overline{\psi}$, the boson $\sigma \boxtimes \overline{\sigma}$ of $Z(\mathsf{Ising})$ splits into the direct sum $e \oplus m$ in the toric code, while $1 \boxtimes \overline{\psi}$ and $\psi \boxtimes 1$ become $em \cong \epsilon$. Thus there are tunneling operators $T_{e \to m}$ and $T_{m \to e}$, as well as tunneling operators $T_{a \to a}$ for $a = 1, e, m, \epsilon$. There are no tunneling operators between any other pairs of Toric code anyons, since fusion with $A$ at the domain wall is equivalent to fusion with the vacuum. More precisely, the free modules $cA_A$ for $c \in \mathrm{Irr}(\mathcal{C})$ partition the simple objects of $\mathcal{C}$, i.e. each $c \in \mathrm{Irr}(\mathcal{C})$ appears in exactly one (isomorphism class of) $dA_A$ where $d \in \mathrm{Irr}(\mathcal{C})$.

For the free modules $A_A$ and $\epsilon \cong (\psi \boxtimes 1)A_A$, we have $\mathcal{C}(A \to A) \cong \mathbb{C} \oplus \mathbb{C} \cong \mathcal{C}((\psi \boxtimes 1)A \to (\psi \boxtimes 1)A)$, so there are 2 tunneling channels $1 \to 1$ and 2 channels $\epsilon \to \epsilon$, corresponding to the two choices of string operator (1 and $\psi \boxtimes \overline{\psi}$, or $(\psi \boxtimes 1)$ and $(1 \boxtimes \overline{\psi})$, respectively) which can transport these between the left and right domain walls. As in the previous example, this leads to one tunneling channel for each particle in each of the two summands $\mathcal{W}_{\pm 1}$. Again, this is because the toric code particles $1$ and $\epsilon$ are free modules, and hence the spaces of tunneling operators $1 \to 1$ and $\epsilon \to \epsilon$ are isomorphic to $\mathcal{C}(A \to A)$ as representations of the algebra $(\mathcal{C}(A \to A), \star)$ of short string operators.

On the other hand, both $e$ and $m$ are local $A$-modules (i.e. anyons in the condensed region) with the same underlying *simple* object (anyon) in the uncondensed topological order, namely $\sigma \boxtimes \overline{\sigma} \in \mathrm{Irr}(\mathcal{C})$. Thus each of the 4 spaces of tunneling operators involving $e$ and $m$ is 1-dimensional, given by $\mathcal{C}(\sigma \boxtimes \overline{\sigma} \to \sigma \boxtimes \overline{\sigma}) \cong \mathbb{C}$. In other words, there are unique (up to phase) tunneling channels $e \to e$, $e \to m$, $m \to e$, and $m \to m$. In a given summand of $\mathcal{W}$, only one of the two possible channels $e \to e$, $e \to m$ is non-vanishing (and similarly for $m$).

(4) To see how $\mathcal{W}_1$ and $\mathcal{W}_{-1}$ differ, we must analyze the spaces of tunneling operators through each wall. The analysis for 1 and $\epsilon$ is analogous to that in § 5.1; as promised, we find that $\mathcal{W}_1$ acts as the trivial superselection sector for these anyons, while $\mathcal{W}_{-1}$ induces an extra phase factor when a $\psi$ particle crosses.

To analyze the tunneling channels for $e$ and $m$, we must compute the action of $(\mathcal{C}(A \to A), \star)$ on the spaces of tunneling operators $e \to e$, $m \to m$, $m \to e$, and $e \to m$. As noted above, $e$ and $m$ particles are both transported through the $Z(\mathsf{Ising})$ strip connecting the left and right Toric code regions using the hopping operator for the anyon $\sigma \boxtimes \overline{\sigma}$. Hence the four tunneling spaces are all isomorphic to $\mathcal{C}(\sigma \boxtimes \overline{\sigma} \to \sigma \boxtimes \overline{\sigma}) \cong \mathbb{C}$; however, they carry different actions of $(\mathcal{C}(A \to A), \star)$.

We now show how to use this action to obtain a description of the superselection sectors, using the mathematical machinery of § 4.4. We begin by investigating how, at either domain wall, the anyon $\sigma \boxtimes \overline{\sigma} \in \mathrm{Irr}(D(\mathsf{Ising}))$ splits as the mixture $e \oplus m$ of toric code anyons. If a $\sigma \boxtimes \overline{\sigma}$ anyon is near a domain wall to the $A$-condensate, we can apply an operator bringing a $\psi \boxtimes \overline{\psi}$ anyon out of the wall and fusing it with the $\sigma \boxtimes \overline{\sigma}$. Since two copies $\psi \boxtimes \overline{\psi}$ fuse to 1, performing this operation twice returns us to the original state. Thus, our $\sigma \boxtimes \overline{\sigma}$ anyon has a 2-dimensional space of configurations. The local operators which can act on a $\sigma \boxtimes \overline{\sigma}$ anyon near the $A$ condensate are the space

$$\mathcal{C}((\sigma \boxtimes \overline{\sigma})A \to (\sigma \boxtimes \overline{\sigma})A) \cong M_2(\mathbb{C}), \tag{41}$$

However, the operators which are stable under fusion with the $A$ condensate are the subspace

$$\mathcal{D}(\mathbb{Z}/2)(e \oplus m \to e \oplus m) \cong \mathcal{C}_A((\sigma \boxtimes \overline{\sigma})A_A \to (\sigma \boxtimes \overline{\sigma})A_A) \cong \mathbb{C} \oplus \mathbb{C}, \tag{42}$$

spanned by projectors which pick out the toric code excitations $e \oplus m$ into which $\sigma \boxtimes \overline{\sigma}$ splits at the wall.

In order to write down operators in $\mathcal{C}((\sigma \boxtimes \overline{\sigma})A \to (\sigma \boxtimes \overline{\sigma})A)$ as $2 \times 2$ matrices, we must choose a particular isomorphism (41), analogous to a choice of orthonormal basis for the 2-dimensional configuration space. We choose the basis $\{|0\rangle, |1\rangle\}$, where $|j\rangle$ is a state where the number of $\psi \boxtimes \overline{\psi}$ lines from the condensate fused into the $\sigma \boxtimes \overline{\sigma}$ anyon modulo 2 is $j$. Mathematically, we can describe this choice as follows. Since $\mathrm{End}_{\mathcal{C}}(A)$ is generated by the orthogonal projections $\pi_1$ and $\pi_\psi$ onto the $1 \boxtimes \overline{1}$ and $\psi \boxtimes \overline{\psi}$ anyons, and there are unique fusion channels $(\sigma \boxtimes \overline{\sigma})(1 \boxtimes \overline{1}) \cong \sigma \boxtimes \overline{\sigma}$ and $(\sigma \boxtimes \overline{\sigma})(\psi \boxtimes \overline{\psi}) \cong \sigma \boxtimes \overline{\sigma}$, $\mathrm{End}_{\mathcal{C}}((\sigma \boxtimes \overline{\sigma}A)$ contains $e_1 := \mathrm{id}_{\sigma \boxtimes \overline{\sigma}} \otimes \pi_1$ and $e_\psi := \mathrm{id}_{\sigma \boxtimes \overline{\sigma}} \otimes \pi_\psi$ as orthogonal rank 1 projections. Then $e_1$ is the projection onto $|0\rangle$ and $e_\psi$ is the projection onto $|1\rangle$, so the isomorphism (41) becomes

$$e_1 = \begin{bmatrix} 1 & 0 \\ 0 & 0 \end{bmatrix}, \qquad e_\psi = \begin{bmatrix} 0 & 0 \\ 0 & 1 \end{bmatrix}.$$

The operator (unique up to phase) which brings a $\psi \boxtimes \overline{\psi}$ line out of the condensate and fuses it with $\sigma \boxtimes \overline{\sigma}$ exchanges $|0\rangle$ and $|1\rangle$, so it is just the Pauli matrix

$$X = \begin{bmatrix} 0 & 1 \\ 1 & 0 \end{bmatrix}.$$

Now, we can determine matrices for the subalgebra (42) of operators stable under fusion with the $A$ condensate. To do so, we will first compute the $(\mathcal{C}(A \to A), \star)$ action on (41), and then apply Lemma 4.8. We know that $\mathrm{id}_A = \pi_1 + \pi_\psi$, that $2\pi_1$ is the identity for the convolution $\star$, and that $\pi_\psi \star \pi_\psi = \frac{1}{2}\pi_1$. Thus, the matrices satisfying the condition

from Lemma 4.8 are exactly those which are preserved under the action of $2\pi_\psi$, which is conjugation by Pauli $X$, i.e.

$$M \mapsto \begin{bmatrix} 0 & 1 \\ 1 & 0 \end{bmatrix} M \begin{bmatrix} 0 & 1 \\ 1 & 0 \end{bmatrix}. \tag{43}$$

If we think of $\mathrm{End}_{\mathcal{C}}((\sigma \boxtimes \overline{\sigma})A \to (\sigma \boxtimes \overline{\sigma})A)$ as a space of tunneling operators through the composite domain wall, we can interpret this fact as follows. Operators which take a $\sigma \boxtimes \overline{\sigma}$ anyon from the left domain wall to the right domain wall map the 2-dimensional space of configurations near the left wall to the 2-dimensional space of configurations near the right wall. Thus, for example, $e_\psi$ is the operator $|1_R\rangle\langle 1_L|$, where the subscripts $L$ and $R$ denote which configuration space a state lives in. If $T$ is such a tunneling operator, then $T \lhd 2\pi_\psi$ is obtained by applying the short string operator associated with $\psi$ parallel to $T$; this short string operator can be fused into the $\sigma \boxtimes \overline{\sigma}$ string near each domain wall, which evidently has the effect (43).

The matrices which are invariant under (43) are just those which are diagonalized in the eigenbasis of Pauli $X$, and are hence spanned by

$$\frac{1}{2}\begin{bmatrix} 1 & 1 \\ 1 & 1 \end{bmatrix} \quad \text{and} \quad \frac{1}{2}\begin{bmatrix} 1 & -1 \\ -1 & 1 \end{bmatrix}.$$

Since these matrices are orthogonal Hermitian projections, one must be $\mathrm{id}_e$, the operator that projects onto an $e$-type domain wall excitation, and the other must be $\mathrm{id}_m$. The choice of which is which is arbitrary, because making a different choice amounts to applying the symmetry exchanging the $e$ and $m$ labels in toric code. We choose

$$\mathrm{id}_e := \frac{1}{2}\begin{bmatrix} 1 & 1 \\ 1 & 1 \end{bmatrix} \quad \text{and} \quad \mathrm{id}_m := \frac{1}{2}\begin{bmatrix} 1 & -1 \\ -1 & 1 \end{bmatrix}.$$

Now that we know the projections which select an $e$ or $m$ particle on each domain wall, it is easy to determine the tunneling operators $e \to e$, $e \to m$, $m \to e$, and $m \to m$ across the composite domain wall. Since each of these projections is rank 1, for each $x, y \in \{e, m\}$, the space of tunneling operators $x \to y$ is spanned by the unique (up to phase) partial isometry $T_{x \to y}$ from $\mathrm{id}_x$ to $\mathrm{id}_y$. Of course, a projection is a partial isometry from itself to itself, so the unique (up to scalar) tunneling operators $e \to e$ and $m \to m$ are respectively

$$T_{e \to e} := \frac{1}{2}\begin{bmatrix} 1 & 1 \\ 1 & 1 \end{bmatrix} \quad \text{and} \quad T_{m \to m} := \frac{1}{2}\begin{bmatrix} 1 & -1 \\ -1 & 1 \end{bmatrix},$$

As $2 \times 2$ matrices, these are the same as $\mathrm{id}_e$ and $\mathrm{id}_m$, but we now interpret them as tunneling operators which map between the two configuration spaces of $\sigma \boxtimes \overline{\sigma}$ near each of the two boundaries, rather than local operators on the single configuration space near one of the boundaries. The other two partial isometries, which take $\mathrm{id}_e$ to $\mathrm{id}_m$ and vice-versa, are:

$$T_{e \to m} = \frac{1}{2}\begin{bmatrix} 1 & 1 \\ -1 & -1 \end{bmatrix} \quad \text{and} \quad T_{m \to e} = \frac{1}{2}\begin{bmatrix} 1 & -1 \\ 1 & -1 \end{bmatrix}.$$

These correspond to tunneling channels exchanging $e$ and $m$.

Now that we have concrete matrices for each tunneling channel $T_{x \to y}$, as well as for the action of $(\mathcal{C}(A \to A), \star)$ on these tunneling operators, we can reap the rewards. That is, we can compute $T_{x \to y} \lhd w_i$ for each choice of $(x, y, i)$, and thereby determine which tunneling operators correspond to which summand of the composite domain wall $\mathcal{W}$. The

tunneling operators $T_{e \to e}$ and $T_{m \to m}$ are stable under conjugation by Pauli $X$, and hence are tunneling operators through the summand $\mathcal{W}_1$ determined by the projection $w_1 = \mathrm{id}_A$. Meanwhile, conjugating the tunneling operators $T_{e \to m}$ and $T_{m \to e}$ by $X$ introduces a factor of $-1$, so these are the tunneling operators through the summand $\mathcal{W}_{-1}$ determined by the projection $w_{-1}$.

This shows that $\mathrm{End}_{\mathcal{X} - \mathcal{X}}(\mathcal{W}_1)$ is the trivial Witt autoequivalence of $\mathcal{C}_A^{\mathrm{loc}} \cong \mathcal{D}(\mathbb{Z}/2)$, while $\mathrm{End}_{\mathcal{X} - \mathcal{X}}(\mathcal{W}_{-1})$ is the autoequivalence coming from the symmetry $\Phi : \mathcal{D}(\mathbb{Z}/2) \to \mathcal{D}(\mathbb{Z}/2)$ which exchanges the anyons $e$ and $m$.

(5) To understand the different superselection sectors intuitively, we once again place our system on a sphere as in Remark 3.23. The ground state Hilbert space contains states in which extended $\sigma \boxtimes 1$ or $1 \boxtimes \overline{\sigma}$ string operators encircle the strip where $A$ is not condensed. Since $\sigma \boxtimes 1$ and $1 \boxtimes \overline{\sigma}$ are the only anyons that become confined in the Toric code regions, and they differ by fusion with anyons that are deconfined everywhere, the operator $L_\sigma$ that inserts such strings is the only operator that can act non-trivially on the ground state Hilbert space.



$\sigma \boxtimes 1$

The operator $L_\sigma$ anti-commutes with $\pi_\psi$, since $S_{(\psi \boxtimes \overline{\psi})(\sigma \boxtimes 1)}/S_{1(\sigma \boxtimes 1)} = S_{(\psi \boxtimes \overline{\psi}),(1 \boxtimes \overline{\sigma})}/S_{1,(1 \boxtimes \overline{\sigma})} = -1$. Thus $L_\sigma$ exchanges the two summands $\mathcal{W}_1$ and $\mathcal{W}_{-1}$. Moreover, with this geometry, the image of $w_1$ ($w_{-1}$) is a strip with an odd (even) number of $\sigma \boxtimes 1$ or $1 \boxtimes \overline{\sigma}$ lines encircling the white strip at the center of the sphere.

We can now understand why the projections $w_{\pm 1}$ have the action that they do. First, because the $\mathcal{W}_{-1}$ domain wall contains an odd number of extended $\sigma$ strings, as shown in [28, 64], an $e$ particle that enters this domain wall will exit as an $m$, and vice versa. Moreover, an $\epsilon$ particle crossing such a wall will incur an extra phase of $\pm i$ relative to the $\epsilon$ particle crossing the trivial, $\mathcal{W}_1$ domain wall due to its half-braiding with the extended $\sigma$ line.

Finally, if we allow pairs of $\sigma \boxtimes 1$ or $1 \boxtimes \overline{\sigma}$ particles in our uncondensed strip, each particle corresponds to a defect separating $\mathcal{W}_1$ and $\mathcal{W}_{-1}$ domain wall types. Once again, local operators cannot be used to distinguish the locus of each domain wall type, but they are clearly distinct: if we braid an $e$ particle around such a defect it returns as an $m$, and vice versa. Bringing an $\epsilon$ around such a defect in principle incurs a net phase of $-1$; however in this case the phase is not well-defined, as it can be modified by attaching an $e$ or $m$ anyon from the bulk to the defect.

**Remark 5.4.** As in the previous example, in the spherical geometry discussed above the two summands $\mathcal{W}_{\pm 1}$ cannot be distinguished by any local process, since the choice of which anyon to call $e$ or $m$ in each region is a matter of convention. However, as before, we can see that the two domain wall types are distinct, by first tunneling an $e$ across the domain wall from left to right, then applying $L_\sigma$, and finally tunneling the same anyon back across from right to left. The particle that returns is necessarily an $m$, showing that one of the two domain wall types exchanges $e$ and $m$, while the other does not.

However, if we put the system on the torus, then we find that in $\mathcal{W}_{-1}$, an $e$ particle that traverses the torus in the direction perpendicular to the strip returns as an $m$. This changes the nature of the topologically distinct ground states on the torus (though not their number).

**Remark 5.5.** Depending on the topology of our system, we can relabel the toric code anyons on only one side of the composite domain wall, exchanging $e$ and $m$. However, such a relabeling will only preserve the Witt equivalence labelling one domain wall $\mathcal{C}_A \cong \mathrm{End}_{\mathcal{X}_A - \mathcal{X}}(\mathcal{X}_A) \cong \mathrm{End}_{\mathcal{X} - \mathcal{X}_A}(\mathcal{X}_A)$ up to monoidal bimodule equivalence, so it will also exchange which short string operators are identified with $w_{\pm 1}$. Thus, the Witt equivalences $End_{\mathcal{X}_A - \mathcal{X}_A}(\mathcal{W}_{\pm 1})$ will be preserved.

## 5.3 Chiral example: $\mathcal{TY}_{3,-}$ wall between $SU(3)_1$ and $SU(2)_4$

We now turn to a chiral example, based on Example 2.2.1.

(1) We take

$$
\begin{aligned}
\mathcal{A} &= \overline{SU(3)_1}\,, \\
\mathcal{X} &= SU(3)_1\,, \\
\mathcal{Y} &= \mathcal{TY}_{3,-} = (SU(2)_4)_{1 \oplus g}\,, \\
_\mathcal{X}\mathcal{M}_\mathcal{Y} &= {}_\mathcal{X}\mathcal{Y}_\mathcal{Y}^{\mathrm{mp}}\,.
\end{aligned}
$$

As we saw in Example 2.2.1, $Z^\mathcal{A}(\mathcal{Y}) = SU(2)_4$. It is clear that $Z^\mathcal{A}(\mathcal{X}) = SU(3)_1$, as $SU(3)_1$ is modular. The algebra in $Z^\mathcal{A}(\mathcal{Y}) \cong SU(2)_4$ which condenses at the boundary is $1 \oplus g$. It follows that $\mathcal{M} := \mathrm{End}^\mathcal{A}_{\mathcal{X} - \mathcal{Y}}(\mathcal{Y}) \cong ((SU(2)_4)_{1 \oplus g})^{\mathrm{mp}} \cong (\mathcal{TY}_{3,-})^{\mathrm{mp}}$.

These choices of $\mathcal{Y}$ and of the bimodule $_\mathcal{X}\mathcal{Y}_\mathcal{X}$ have an obvious implementation in terms of the lattice model described in § 2.2: we simply restrict the set of labels of edges in regions labelled by $\mathcal{X}$ to the simple objects of $\mathcal{X} \cong \mathcal{A}$ (as UFCs), rather than allowing the additional simple object of $\mathcal{Y} = \mathcal{TY}_{3,-}$. In other words, while the description of the topological boundary corresponding to $\mathcal{Y}$ in § 2.2 was somewhat involved, $\mathcal{X}$ is the topological boundary for the $\mathcal{A}$-bulk obtained by simply cutting it off at a plane.

(2) The analysis of the algebra $(\mathcal{C}(A \to A), \star) \cong \mathbb{C}[\mathbb{Z}/2]$ is identical to the non-chiral examples 5.1 and 5.2. In summary, if $\pi_1$ and $\pi_g$ are minimal composition projections in $\mathcal{C}(A \to A)$, then the identity for $\star$ is $2\pi_1$, and minimal projections for $\star$ are

$$
\begin{aligned}
w_1 &:= \mathrm{id}_A = \pi_1 + \pi_g\,, \\
w_{-1} &:= \pi_1 - \pi_g\,.
\end{aligned}
$$

The domain wall $\mathcal{W} := \mathcal{M} \boxtimes_\mathcal{Y} \overline{\mathcal{M}}$ therefore decomposes as a direct sum $\mathcal{W} = \mathcal{W}_1 \oplus \mathcal{W}_{-1}$ of two indecomposable domain walls. As before, $\mathcal{W}_1$ is the identity domain wall of $\mathcal{C}_A^{\mathrm{loc}} \cong SU(3)_1$. As we will see, $\mathcal{W}_{-1}$ is the nontrivial invertible boundary which exchanges the anyons $\alpha$ and $\alpha^2$ of $SU(3)_1$.

(3) The analysis is parallel to that of § 5.2. The space of tunneling operators $1 \to 1$ across the composite $\mathcal{W}$ domain wall is, as always, just $(\mathcal{C}(A \to A), \star)$. The only other anyons in $SU(3)_1$, $\alpha$ and $\alpha^2$, arise as summands when the anyon $f_2 \in \mathcal{C}$ splits on the domain wall: $f_2 A_A \cong \alpha \oplus \alpha^2$. As in the previous example, the forgetful functor $SU(3)_1 \cong \mathcal{C}_A \to \mathcal{C} \cong SU(2)_4$ maps both $\alpha$ and $\alpha^2$ to $f_2$. Therefore, the space

$$
\mathcal{C}(f_2 A \to f_2 A) \cong M_2(\mathbb{C})\,, \tag{44}
$$

which can be interpreted as the space of operators bringing an $f_2$ anyon from the left domain wall to the right domain wall, is the direct sum of the four 1-dimensional spaces of tunneling operators $\alpha \to \alpha$, $\alpha \to \alpha^2$, $\alpha^2 \to \alpha$, and $\alpha^2 \to \alpha^2$.

(4) To analyze the tunneling operators and determine which Witt autoequivalence of $SU(3)_1$ describes each superselection sector, we must again work out a concrete isomorphism for (44). One can think of an $f_2$ anyon near either of the domain walls as having a 2-dimensional configuration space, again spanned by two orthogonal states $|0\rangle$ and $|1\rangle$, where the state $|i\rangle$ contains $i$ modulo 2 $g$-lines between the condensate and the $f_2$ anyon. Operators which pull a $g$ out of the condensate and fuse it with $f_2$ again act as Pauli $X$ on this configuration space.

In terms of our hom-spaces, $\mathcal{C}(f_2 A \to f_2 A)$ contains the orthogonal projections $e_1 := \mathrm{id}_{f_2} \otimes \pi_1$ and $e_g := \mathrm{id}_{f_2} \otimes \pi_g$, which become the matrices

$$e_1 = \begin{bmatrix} 1 & 0 \\ 0 & 0 \end{bmatrix}, \qquad e_g = \begin{bmatrix} 0 & 0 \\ 0 & 1 \end{bmatrix}.$$

This completely determines the isomorphism (44). Then, by definition, $e_1 \lhd 2\pi_g = e_g$ and $e_g \lhd 2\pi_g = e_1$, so $2\pi_g$ acts as Pauli $X$ as promised.

When interpreting (44) as the sum of spaces of tunneling operators, we have a right action of $(\mathcal{C}(A \to A), \star)$; the action of $\pi_g$ is to apply a short $g$-string operator between the two domain walls in parallel to the tunneling operator. As before, since this string could be fused with the $f_2$ string crossing the $\mathcal{C}$ bulk, the action of $\pi_g$ is therefore conjugation by Pauli $X$, i.e. by the matrix $\begin{bmatrix} 0 & 1 \\ 1 & 0 \end{bmatrix}$.

This has two consequences. First, if we interpret $\mathcal{C}(f_2 A \to f_2 A)$ as a space of local operators acting on an $f_2$ anyon near one of the domain walls, then by Lemma 4.8, the subspace

$$SU(3)_1(\alpha \oplus \alpha^2 \to \alpha \oplus \alpha^2) \cong \mathcal{C}_A(f_2 A_A \to f_2 A_A) \cong \mathbb{C} \oplus \mathbb{C} \tag{45}$$

of (44), which is the space of operators stable under fusion with the $A$ condensate, is the space of matrices stable under conjugation by Pauli $X$. Just as in § 5.2, minimal projections in this subspace are given by the following matrices.

$$\mathrm{id}_\alpha := \frac{1}{2}\begin{bmatrix} 1 & 1 \\ 1 & 1 \end{bmatrix} \quad \text{and} \quad \mathrm{id}_{\alpha^2} := \frac{1}{2}\begin{bmatrix} 1 & -1 \\ -1 & 1 \end{bmatrix}.$$

Of course, we could swap the roles of $\alpha$ and $\alpha^2$, which amounts to composing with an invertible domain wall which exchanges the two.

Second, now that we have identified projections onto the $\alpha$ and $\alpha^2$ summands near each domain wall, we can resume our interpretation of (44) as the space of operators bringing an $f_2$ anyon from one domain wall to the other, in which case the subspace (45) consists of tunneling operators in the $\mathcal{W}_1$ sector. The above projections are thus also tunneling operators

$$T_{\alpha \to \alpha} := \frac{1}{2}\begin{bmatrix} 1 & 1 \\ 1 & 1 \end{bmatrix} \quad \text{and} \quad T_{\alpha^2 \to \alpha^2} := \frac{1}{2}\begin{bmatrix} 1 & -1 \\ -1 & 1 \end{bmatrix},$$

since a projection is a partial isometry from itself to itself. Thus, we see that $\mathrm{End}^A(\mathcal{W}_1)$ is the trivial Witt autoequivalance.

Meanwhile, tunneling operators through the $\mathcal{W}_{-1}$ sector form the orthogonal complement of (45) in (44), i.e. the space of matrices which obtain a phase of $-1$ when conjugated by Pauli $X$. Thus, they are spanned by the tunneling operators

$$T_{\alpha \to \alpha^2} = \frac{1}{2}\begin{bmatrix} 1 & 1 \\ -1 & -1 \end{bmatrix}, \qquad T_{\alpha^2 \to \alpha} = \frac{1}{2}\begin{bmatrix} 1 & -1 \\ 1 & -1 \end{bmatrix}.$$

This confirms that the anyons $\alpha$ and $\alpha^2$ are exchanged in the $\mathcal{W}_{-1}$ sector.

(5) The anyons $f_1$ and $f_3$ in $Z^{\mathcal{A}}(\mathcal{Y}) \cong SU(2)_4$ become confined in the condensate. These differ by fusion with the condensing $g$ anyon. Thus if we put our model on a capped sphere, there is a single operator $L_f$, corresponding to an $f_1$ or $f_3$ string encircling the uncondensed strip, that acts non-trivially on the ground state Hilbert space. (In this case, $f_1$ and $f_3$ strings become identified at the boundaries of the strip, as $f_1 A_A \cong f_3 A_A$).

From the $S$-matrix in § 2.2, we can see that the short string operator $\pi_g$ and an extended string operator $L_f$ anticommute, so that $L_f$ exchanges the images of the projections $w_1$ and $w_{-1}$. Moreover, we see that for the trivial domain wall $\mathcal{W}_1$, which is the image of $w_1$, the strip must contain an even number of extended $f_1$ or $f_3$ loops. The non-trivial domain wall $\mathcal{W}_{-1}$, in contast, is encircled by an odd number of such loops.

Finally, a pair of $f_1$ or $f_3$ anyons in the strip are point defects separating $\mathcal{W}_1$ and $\mathcal{W}_{-1}$ wall regions. An $\alpha$ particle that is sent through the domain wall on one side of such a point defect, and brought back on the other, returns as $\alpha^2$.

In summary: as in § 2.2.1, by setting $\mathcal{A} = \overline{SU(3)_1}$ and $\mathcal{X} \cong SU(3)_1$, we begin with a Walker-Wang model associated to the UMTC $SU(3)_1$ in the (3+1)D bulk, and cut off along a plane to obtain a (2+1)D boundary with $Z^{\mathcal{A}}(\mathcal{X}) \cong SU(3)_1$ topological order. As a fusion category, $\mathcal{X} = SU(3)_1$ includes as (the full subcategory spanned by) the invertible objects of our chosen $\mathcal{Y} = \mathcal{TY}_{3,-}$, so we can extend the boundary lattice model to allow the additional simple object $\sigma \in \mathrm{Irr}(\mathcal{TY}_{3,-})$. This strip then exhibits $Z^{\mathcal{A}}(\mathcal{Y}) \cong SU(2)_4$ topological order, and the algebra $(1 \oplus g) \in SU(2)_4$ condenses at each edge of the strip.

Similar to the situation in § 5.2, the short $g$-string operator between the two domain walls splits the boundary into two superselection sectors, which differ by the autoequivalence of $Z^{\mathcal{A}}(\mathcal{X}) \cong SU(3)_1$ which exchanges $\alpha$ and $\alpha^2$. Applying the short $g$-string operator can be interpreted as counting the number of $f_1$ and $f_3$ lines in the uncondensed strip modulo 2, and the confined $f_1$ and $f_3$ anyons become point defects between the two summands of composite domain wall.

## 5.4 Nonabelian condensate example: dihedral groups

This example concerns a condensate in the double of $\mathcal{D}(D_n)$ of the dihedral group $D_n$. Because a complete description of $\mathcal{D}(D_n)$ is lengthy and the details are known to many in the community, we defer a full treatment to Appendix C. Definitions, notation, and basic results regarding the double $\mathcal{D}(G)$ of a finite group, as well as a thorough analysis of the special case $\mathcal{D}(D_n)$ where this example arises, appear there. In particular, we write $\mathbb{C}[H]$ for the group algebra and $\mathbb{C}^H$ for the algebra of functions on a group $H$.

(1) In this example, we consider

$$
\begin{aligned}
\mathcal{A} &= \mathsf{Hilb}\,, \\
\mathcal{X} &= \mathsf{Hilb}[D_a]\,, \\
\mathcal{Y} &= \mathsf{Hilb}[D_n]\,, \\
{}_{\mathcal{X}}\mathcal{M}_{\mathcal{Y}} &= \mathsf{Hilb}[D_a]\,,
\end{aligned}
$$

where $n$ is odd, and $a$ is a divisor of $n$. Here, $D_k$ is the dihedral group of order $2k$, with the presentation $D_k \cong \langle r, f \,|\, r^k, f^2, (rf)^2 \rangle$. Anyons in $\mathcal{D}(D_k) \cong Z(\mathsf{Hilb}_{\mathsf{fd}}[D_k])$ are irreducible $D_k$-graded $D_k$-representations, which we denote by $(g, \rho)$, where $g \in D_k$ and $\rho \in \mathrm{Irr}(\mathrm{Rep}(\mathrm{Stab}_{D_k}(g)))$.

At the domain wall, we condense the subgroup algebra

$$
A = \mathbb{C}[\langle r^a \rangle] \cong \bigoplus_{k=1}^{\frac{n/a-1}{2}} (r^k, 1)\,.
$$

The resulting fusion category $\mathcal{X} \cong \mathcal{Y}_A$ is equivalent to $\mathrm{Hilb}[D_a]$, because $D_n/\langle r^a \rangle \cong D_a$ (note $\langle r^a \rangle \cong \mathbb{Z}/\frac{n}{a}$). Thus, we have $Z(\mathcal{X}) \cong \mathcal{D}(D_a)$. The Witt equivalence $\mathrm{End}_{\mathcal{X}-\mathcal{Y}}(\mathcal{X})$ of wall excitations is given by $\mathcal{D}(D_a, D_n)$, where objects are $D_a$-graded $D_n$-representations; see Definition C.3 and Example C.4. As for the bimodule tensor category structure, the action $\mathcal{D}(D_a) \to \mathcal{D}(D_a, D_n)$ involves inducing each $D_a$ representation to all of $D_n$, while the action $\mathcal{D}(D_n) \to \mathcal{D}(D_a, D_n)$ involves applying the quotient map $D_n \to D_a$ to the grading.

Before doing any computations, we will summarize the results of our analysis. The composite domain wall $\mathcal{W}$ has $\frac{n/a+1}{2}$ summands. One of these $\mathcal{W}_0$, is the identity domain wall of $\mathcal{D}(D_a)$. The remaining $\frac{n/a-1}{2}$ summands all correspond to the Lagrangian algebra $L(B, B, \mathrm{id}) \in \mathcal{D}(D_a) \boxminus \overline{\mathcal{D}(D_a)}$, where $B = \mathbb{C}^{D_a/\langle f \rangle} \cong 1 \oplus (1, \epsilon)$. The anyon $(f, 1) \in \mathcal{D}(D_n)$ splits as a direct sum of several wall excitations at each $\mathcal{D}(D_n)-\mathcal{D}(D_a)$ domain wall, one of which becomes the anyon $(f, 1) \in \mathcal{D}(D_a)$, and the rest of which are distinct and confined to the wall. Each summand of $\mathcal{W}$ gives a different identification of the summands of $(f, 1)$ on the left and right domain walls, with only the identity summand of $\mathcal{W}$ identifying the mobile summands $(f, 1) \in \mathcal{D}(D_a)$ on both sides. We will see that the summands other than $\mathcal{W}_0$ are all equivalent as Witt equivalences $\mathcal{D}(D_a) \to \mathcal{D}(D_a)$.

A description of string-net models for a class of similar boundaries appears in Appendix C.2; this boundary is the case where $G = D_n$, $H = \langle r^a \rangle \subseteq D_n$.

(2) The algebra $\mathcal{C}(A \to A)$ is generated under composition by projections $\pi_{r^{ka}} = \pi_{r^{-ka}}$ onto the simple summands of $A$, where $k$ runs from 0 to $\frac{n/a-1}{2}$. We also adopt the notation $\pi_1 := \pi_{r^0}$. Observe that $\mathrm{End}(A) \cong (\mathbb{C}^{\langle r^a \rangle})^f$, the part of the algebra $\mathbb{C}^{\langle r^a \rangle}$ of functions on $\langle r^a \rangle \cong \mathbb{Z}/\frac{n}{a}$ stable under the action of $f$, which exchanges $r^a$ and $r^{-a}$. Along the same lines, the algebra $(\mathcal{C}(A \to A), \star)$ is just $\mathbb{C}[\langle r^a \rangle]^f$, the part of the group algebra of $\langle r^a \rangle$ stable under the action of $f$. Explicitly, the identity for $\star$ is $\frac{n}{a}\pi_1$, and the convolution product is given by

$$\pi_{r^{ka}} \star \pi_{r^{ja}} = \frac{a}{n}\left(\pi_{r^{(k+j)a}} + \pi_{r^{(k-j)a}}\right).$$

Minimal projections for the convolution product are as follows.

$$w_0 := \sum_{j=0}^{\frac{n/a-1}{2}} \pi_{r^{ja}},$$

$$w_k := \sum_{j=0}^{\frac{n/a-1}{2}} \left(e^{2\pi ijk(a/n)} + e^{-2\pi ijk(a/n)}\right)\pi_{r^{ja}}.$$

We again denote the summand of $\mathcal{W}$ corresponding to $w_k$ by $\mathcal{W}_k$. The minimal projections correspond to indecomposable $D_n$-subrepresentations of $\mathbb{C}[\langle r^a \rangle]$, i.e. subspaces of $\mathbb{C}[\langle r^a \rangle]$ which are preserved by the $D_n$-action. As usual, $w_0$ is proportional to $\mathrm{id}_A$, and $\mathcal{W}_0$ is the identity/trivial domain wall from $Z(\mathcal{X}) \cong \mathcal{C}_A^{\mathrm{loc}} \cong \mathcal{D}(D_a)$ to itself.

(3) Most of the anyons $d \in \mathcal{D}(D_a) \cong \mathcal{D}(D_n)_A^{\mathrm{loc}}$ are free $A$-modules. Similar to the the previous two examples, distinct free modules partition those simple objects of $\mathcal{D}(D_n)$ which have the form $(r^x, \omega^j)$, including the objects $(1, \sigma_j)$. (Here $r^x$ represents a conjugacy class of $D_n$, while $\omega^j$ and $\sigma^j$ are associated with irreducible representations of $D_n$; see Appendix C.3 for details). For each such anyon $c \in \mathcal{D}(D_n)$, either $c$ is a summand of $A$, or $cA$ is the direct sum of $\frac{n}{a}$ distinct simple objects. In fact, in this case we have $\mathcal{C}(cA \to cA) \cong \mathbb{C}[\langle r^a \rangle]$ as a $\mathcal{C}(A \to A) \cong \mathbb{C}[\langle r^a \rangle]^f$-module. Therefore, for anyons $d$ of the form $(1, \sigma_k)$ or $(r^k, \omega^j)$, there are 1-dimensional spaces of tunneling operators $d \to d$ for the summand $\mathcal{W}_0$, and 2-dimensional spaces of tunneling operators $d \to d$ for $\mathcal{W}_k$ when $k \neq 0$.

The remaining free modules can be divided into two overlapping pairs: $A$ and $(1, \epsilon)A$; and $(f, 1)A$ and $(f, -1)A$. We begin with the latter. Something more interesting happens to the anyons $(f, \pm 1) \in \mathcal{D}(D_a)$, because the anyons $(f, \pm 1) \in \mathcal{D}(D_n)$ split into several summands at the domain wall, only one of which is local. Since $(f, 1) \cong (f, -1)(1, \epsilon)$, and the Abelian anyon $(1, \epsilon)$ remains an Abelian anyon when $A$ is condensed, we focus solely on the fate of $(f, 1)$. As described in Appendix C.3, the free module $(f, 1)A_A$ decomposes as follows.

$$(f, 1)A_A \cong (f, 1)_A \oplus \left( \bigoplus_{k=1}^{\frac{n/a-1}{2}} (f, \sigma_k)_A \right).$$

Here $(g, \rho)A_A$ denotes a free module over an object $(g, \rho) \in \mathrm{Irr}(\mathcal{D}(D_n))$, where $g \in D_n$ and $\rho \in \mathrm{Irr}(\mathrm{Rep}(\mathrm{Stab}_{D_n}(g)))$, while $(h, \lambda)_A$ denotes an object in $\mathrm{Irr}(\mathcal{D}(D_n)_A \cong \mathcal{D}(D_a, D_n))$, where $h \in D_a$, and $\lambda \in \mathrm{Irr}(\mathrm{Rep}(\mathrm{Stab}_{D_n}(h)))$, with $D_n$ acting on $D_a \cong D_n/\langle r^a \rangle$ by conjugacy. In particular, when we write $(f, \rho)_A$, $\rho$ is an irreducible representation of

$$\mathrm{Stab}_{D_n}(f) = \langle f, r^a \rangle \cong D_{n/a}.$$

The underlying object of $(f, 1)_A$ is the simple object $(f, 1)$, while the underlying object of $(f, \sigma_k)_A$ is $(f, 1) \oplus (f, -1)$. Consequently, there are 1-dimensional spaces of tunneling operators $(f, 1)_A \to (f, 1)_A$.

(4) In this case, we will be able to identify the domain wall summands $\mathcal{W}_i$ by the numbers of tunneling channels alone. Since

$$\mathcal{W} \cong \bigoplus_{k=0}^{\frac{n-1}{2a}} \mathcal{W}_k$$

has many summands, yet $\mathcal{C}_A((f, 1)_A, (f, 1)_A) \cong \mathcal{C}((f, 1), (f, 1))$ is only 1-dimensional, we can immediately see that there *cannot* be nonzero tunneling operators for the anyon $(f, 1)_A \in Z(\mathcal{X})$ through all summands of $\mathcal{W}$ - in fact, $(f, 1)$ must be confined in every summand but $\mathcal{W}_0$.

The fate of $(1, \epsilon)$ is closely related to that of $(f, 1)$. The anyon $(1, \epsilon)$ in the condensed $\mathcal{D}(D_a)$ topological order is just the free module $(1, \epsilon)A_A$. However, since $(1, \epsilon)(r^{ak}, 1) \cong (r^{ak}, 1)$, there are many nonzero tunneling operators $1 \to (1, \epsilon)$ and $(1, \epsilon) \to 1$.

Explicitly, we have $\dim(\mathcal{C}(A \to A)) = \dim(\mathcal{C}((1, \epsilon)A \to (1, \epsilon)A)) = \frac{n/a+1}{2}$, while $\dim(\mathcal{C}(A \to (1, \epsilon)A)) = \dim(\mathcal{C}((1, \epsilon)A \to A)) = \frac{n/a-1}{2} = \frac{n/a+1}{2} - 1$. Thus, we see that the anyon $(1, \epsilon) \in \mathcal{C}_A^{\mathrm{loc}}$ from the condensed region *condenses* on at least some summands of the domain wall.

We can use these observations to give a more precise description of the various summands in terms of a Lagrangian algebra $L(B, B, \Phi)$ of the folded theory $\mathcal{D}(D_a) \boxtimes \overline{\mathcal{D}(D_a)}$. We know that $(f, 1)$ is confined by all but the trivial summand $\mathcal{W}_0$. Thus some anyon which braids nontrivially with $(f, 1)$ must condense on each of the remaining $\frac{n/a+1}{2} - 1$ summands. This can only be $(1, \epsilon)$, since no other anyons of $\mathcal{C}_A^{\mathrm{loc}}$ have nonzero tunneling operators to the vacuum. Thus, with the exception of $\mathcal{W}_0$, all summands of $\mathcal{W}$ must correspond to a Lagrangian algebra $L(B, B, \Phi)$, where $B \cong 1 \oplus (1, \epsilon)$ is the condensable algebra in $\mathcal{D}(D_a)$ described above, and $\Phi$ is an outer automorphism of $\mathcal{D}(D_a)_B^{\mathrm{loc}} \cong \mathcal{D}(\mathbb{Z}/a)$. In other words, the indecomposable domain walls in these superselection sectors correspond to a strip of $\mathbb{Z}/a$ toric code containing an invertible boundary corresponding to the autoequivalence $\Phi$, separating the two $\mathcal{D}(D_a)$ bulk regions. Since $\mathbb{Z}/a \subseteq D_a$ is the subgroup generated by

$r \in D_a$, this is an example of the 'smooth' boundary $\mathcal{D}(G) - \mathcal{D}(K)$ for $K \subseteq G$ for which a lattice model is given in § C.2.

As shown in Appendix C, $\mathcal{D}(D_a)_B^{\mathrm{loc}} \cong \mathcal{D}(\mathbb{Z}/a)$, the $\mathbb{Z}/a$ toric code, with $(r^x, \omega^j) \in \mathcal{D}(D_a)$ splitting as $e^x m^j \oplus e^{-x} m^{-j}$. The possible autoequivalences of $\mathcal{D}(\mathbb{Z}/a)$ are computed in [9, 55]. Because each $(r^x, \omega^j)$ only tunnels to itself through each $\mathcal{W}_k$, the autoequivalence $\Phi : \mathcal{D}(\mathbb{Z}/a) \to \mathcal{D}(\mathbb{Z}/a)$ must be either the identity, or the autoequivalence $\Psi$ which maps $e \to \bar{e}$, $m \to \bar{m}$. However, because $B \cong \mathbb{C}^{D_n/\langle r \rangle} \cong \mathbb{C}^{\mathbb{Z}/2}$, there is a nontrivial automorphism $\psi$ of $B$, which acts as $1$ on the $1$ component and $-1$ on the $(1, \epsilon)$ component. This automorphism $\psi$ induces the automorphism $\Psi$ of $\mathcal{D}(D_n)_B^{\mathrm{loc}}$, so as explained in Remark 3.6, $L(B, B, \mathrm{id})$ and $L(B, B, \Psi)$ are isomorphic Lagrangian algebras, meaning that the two possible autoequivalences $\mathrm{id}$ and $\Psi$ give equivalent domain walls.

(5) Anyons of the form $(1, \sigma_k)$ where $\frac{n}{a}$ does not divide $k$ are confined to the middle $\mathcal{D}(D_n)$ strip. From the half-braiding (C.1) on $\mathcal{D}(D_n)$ and the idempotents in $\mathcal{C}(A \to A)$ computed above, we can see that the operator $L_{\sigma_k}$ that inserts an extended $(1, \sigma_k)$-string operator in the middle strip satisfies $L_{\sigma_k} w_0 = w_k L_{\sigma_k}$, meaning that $L_{\sigma_k}$ maps the $\mathcal{W}_0$ sector onto the $\mathcal{W}_i$ sector. More generally, $L_{\sigma_k}$ maps the $\mathcal{W}_i$ sector onto the $\mathcal{W}_{i+k}$ and $\mathcal{W}_{i-k}$ sectors.

One might interpret this by saying that the idempotents $w_i$ count (up to sign) the number of extended $m$-strings in the *uncondensed* bulk region running parallel to the domain walls region in an appropriate non-Abelian sense, where $m \oplus m^{-1} := (1, \sigma_1)$. Of course, just as we saw in the toric code example of § 5.1, the exact number of $m$-strings in a particular superselection sector is not well-defined (even up to sign), because of the possibility of different microscopic realizations of the condensation domain walls. This is again reflected mathematically by the fact that the summands $\mathcal{W}_i$ for $i \neq 0$ are equivalent $\mathcal{X} - \mathcal{X}$ bimodule categories, even though they are different as summands of $\mathcal{W}$.

The description of point defects between different summands in terms of $\mathcal{D}(D_n)$ bulk anyons is exactly analogous to the examples discussed above: the anyon $(1, \sigma_k)$ becomes a point defect between $\mathcal{W}_i$ and $\mathcal{W}_{i+k} \oplus \mathcal{W}_{i-k}$. When an $(r^x, 1)$ particle from one of the condensed regions with $\mathcal{D}(D_a)$ topological order is brought into the bulk, braids around a $(1, \sigma_k)$ anyon, and returns to the boundary, it acquires a phase of $\omega^{\pm kx}$, with the sign determined by the choice of superselection sectors above and below the defect.

Similar to the previous examples, we can view an $(f, 1)$ anyon near a region where $A$ is condensed as having a configuration space corresponding to the object $(f, 1)A \in \mathcal{D}(D_n)$. When an $(f, 1)$ anyon braids around a $(1, \sigma_k)$ anyon, summands of $((f, 1)A)(1, \sigma_k) \cong (1, \sigma_k)((f, 1)A)$ acquire different phases, so that the summands of $(f, 1)A_A$ are permuted. One of these summands becomes the anyon $(f, 1)_A$ in the condensed phase $\mathcal{D}(D_a)$, while the others are confined excitations on the domain wall. Thus, the $(f, 1)$ particle in the $\mathcal{D}(D_a)$ condensate is unable to tunnel through at least one of the wall segments adjacent to a $(1, \sigma_k)$ point defect, and hence its braiding with such defects cannot be defined.

# 6 Conclusions and outlook

In this article, we introduced enriched UFCs as a new categorical framework of for describing (2+1)D topologically ordered phases and topological domain walls between them. We propose that particle mobility through domain walls is characterized by tunneling operators, which appear as higher morphisms in a 3-category of (2+1)D topological orders. We used the action of tunneling operators on spherical ground states to identify the superselection sectors

of composite domain walls, analyzing each sector from a particle mobility perspective, and explicitly demonstrated our methods in several examples.

In particular, we saw that when performing anyon condensation in the complement of a strip, the composite domain wall between condensed bulks has several distinct superselection sectors. If our condensate is associated with deequivariantization, then different superselection sectors of the domain wall act differently on anyons in the condensed phase which arise from the splitting of an anyon in the uncondensed phase. In examples which go beyond deequivariantization, such as the dihedral example of § 5.4, summands of the composite boundary may differ in terms of particle mobility: anyons from the condensed phase may become condensed or confined at the domain wall in some superselection sectors and not others.

We also saw that, when two parallel domain walls are placed on a sphere, non-contractible loop operators in the uncondensed strip exchange the superselection sectors for the domain wall. One direction for future investigation would be to uncover the general story of non-local operators in the middle strip which map between superselection sectors of the composite domain wall. Loop operators associated with confined anyons act transitively on superselection sectors, and point defects between the domain walls which arise as summands of a composite domain wall come from confined anyons in the middle bulk region, but such point defects may also have a simpler description in terms of wall excitations on the original domain walls.

Another possible direction involves using a more general notion of boundary between (2+1)D topological orders. In this article, we considered only (1+1)D domain walls which live on the boundary of Walker-Wang models, which do not extend into the (3+1)D bulk. One could also consider domain walls which do extend into the bulk, i.e., (1+1)D domain walls on the boundary which are attached to a (2+1)D domain wall between Walker-Wang bulks. This could potentially include gapless domain walls between topological orders with a different Witt class as the anomaly, which were studied extensively in [92,93].

The ground state degeneracy associated with domain walls in (2+1)D which we have analyzed has potential uses in quantum computation [13]. Also, there has been much recent work on the machinery of fusion 2-categories [40,45] and nondegenerate braided fusion 2-categories [72], which provide a mathematical description of (3+1)D topological order [88,89,98]. Some of the theory of condensable algebras which underlies our results has already been lifted to this setting [38,39,60], and further lifting our results to the case of fusion 2-categories would be a natural approach to understanding the composition of (2+1)D domain walls between (3+1)D topological orders.

## Acknowledgements

The authors would like to thank David Aasen, Jacob Bridgeman, and Dominic Williamson for helpful conversations. We especially thank David Reutter for many conversations and ideas from the initial phases of this article.

**Funding information**    This project began at the 2020 MSRI semester on Quantum Symmetries supported by NSF grant DMS 1440140. PH and DP were supported by NSF grants DMS 1654159, 1927098, and 2051170. CJ was supported by NSF grant DMS 1901082/210. FJB is supported by NSF grant DMR 1928166.

# A Fusion and modular categories

Recall that a multifusion category $\mathcal{X}$ is a finite semisimple tensor category where all objects are dualizable [48, Def. 4.1.1]. If the tensor unit $1_{\mathcal{X}}$ is a simple object then $\mathcal{X}$ is called a fusion category. As noted in the introduction, we will omit the tensor product symbol and simply write $ab$ for the tensor product of two objects in $\mathcal{X}$. We denote by $\mathrm{Irr}(\mathcal{X})$ a set of representatives for the simple objects in $\mathcal{X}$, so that every object is isomorphic to one of the form $\bigoplus_{f \in \mathrm{Irr}(\mathcal{X})} N_f f$. The *associator* is a natural isomorphism

$$\alpha_{a,b,c} : (ab)c \to a(bc),$$

which will be suppressed whenever possible. For the description in terms of $F$-matrices, we refer the reader to [124].

Given a fusion category $\mathcal{X}$, one can construct the fusion category $\mathcal{X}^{\mathrm{mp}}$ where the order of tensor product is reversed, i.e. $x \otimes_{\mathcal{X}^{\mathrm{mp}}} y := y \otimes_{\mathcal{X}} x$, replacing $\alpha$ with $\alpha^{-1}$.

By [115], if a fusion category is unitarizable, then it is uniquely unitarizable. This means being unitary is a property of a fusion category, which manifests as the existence of a set $\mathrm{Irr}(\mathcal{X})$ such that the $F$-matrices are unitary.

In particular, a UMTC is a unitary fusion category equipped with a unitary nondegenerate braiding, where nondegenerate means that the $S$-matrix is invertible. If $\mathcal{C}$ is a UMTC, objects in $\mathrm{Irr}(\mathcal{C})$ classify anyon types in a $\mathcal{C}$ topological order.

We make heavy use of the graphical calculus for UMTCs [7, 68]. Strings are labelled by objects in a UMTC, and correspond to the worldlines of direct sums of anyons [21, 26, 78, 124]. We denote the braiding of $\mathcal{C}$ by $\beta$, which we depict graphically by a crossing:

$$\beta_{c,d} = \underset{c \quad d}{\diagup\!\!\!\!\diagdown} \;\; : cd \to dc. \tag{A.46}$$

For the description of the braiding in terms of unitary $R$-matrices, we refer the reader to [124]. In general, the objects $c$ and $d$ in the above diagram can be direct sums of simple objects; the braiding between two direct sums is determined by the braidings between each pair of summands.

By [63, 73], the collection of fusion categories forms a 3-category named UFC, whose objects are fusion categories, 1-morphisms are finitely semisimple bimodule categories, 2-morphisms are bimodule functors, and 3-morphisms are bimodule natural transformations. We refer the reader to [46] for more details. This 3-category has a symmetric monoidal structure given by Deligne tensor product $\boxtimes$. Given fusion categories $\mathcal{X}, \mathcal{Y}$, the simple objects of $\mathcal{X} \boxtimes \mathcal{Y}$ are exactly $f \boxtimes g$ such that $f \in \mathrm{Irr}(\mathcal{X})$ and $g \in \mathrm{Irr}(\mathcal{Y})$, with the obvious tensor product fusion rules.

A braided fusion category $\mathcal{C}$ can act on a fusion category $\mathcal{X}$ via a braided tensor functor $\Phi : \mathcal{C} \to Z(\mathcal{X})$ into the Drinfeld center of $\mathcal{X}$. A fusion category $\mathcal{X}$ equipped with such a $\mathcal{C}$-action is called a *module tensor category* for $\mathcal{C}$ [67, 90]. If $U : Z(\mathcal{X}) \to \mathcal{X}$ is the functor which forgets the half-braiding, then the action $\mathcal{C} \boxtimes \mathcal{X} \to \mathcal{X}$ given by $c \boxtimes f \mapsto U(\Phi(c))f$ makes $\mathcal{X}$ an ordinary module category for the underlying fusion category of $\mathcal{C}$. The full data of $\Phi$ can be thought of as compatibility between this action, the braiding in $\mathcal{C}$, and the tensor product in $\mathcal{X}$. Given two braided fusion categories $\mathcal{C}, \mathcal{D}$, a *bimodule tensor category* is a module tensor category for the Deligne product $\mathcal{C} \boxtimes \overline{\mathcal{D}}$, where $\overline{\mathcal{D}}$ denotes taking the reverse braiding.[41]

Similarly, braided fusion categories form a symmetric monoidal 4-category called UBFC, whose objects are braided fusion categories, 1-morphisms are bimodule multifusion categories, 2-morphisms are finitely semisimple bimodule categories with appropriate coherences,

---

[41]We use the same labels for objects in $\mathcal{D}$ and $\overline{\mathcal{D}}$, so that $\overline{d}$ denotes the dual of an object $d$, rather than the corresponding object in $\overline{\mathcal{D}}$.

3-morphisms are bimodule functors, and 4-morphisms are natural transformations. We refer the reader to [14,74] for more details. Again, UBFC is symmetric monoidal under the Deligne product. Physically, the Deligne product $\mathcal{C} \boxtimes \mathcal{D}$ corresponds to stacking two decoupled layers of (2+1)D phases. The composition of 1-morphisms will be discussed in § 3.1 below.

## B  Condensable algebras

In this appendix, we recall the mathematical notions necessary to understand anyon condensation, most of which appear in [42], along with their physical interpretation, which mostly follows [85]. The basic idea of anyon condensation is that some collection of anyons, the condensate, becomes identified with the vacuum. In order to consistently identify a condensate with the vacuum, the condensate must come equipped with additional data: the structure of a *condensable algebra*, also known as a unitarily separable étale algebra. We also define the category of modules over a condensable algebra, which can be used to describe the topological order in and at the boundary of regions where the condensable algebra has been condensed.

An algebra object $A$ in the UMTC $\mathcal{C}$ consists of the data $(A, m_A, i_A)$, where $A$ is an object in $\mathcal{C}$, i.e. a direct sum of anyons, the multiplication operator $m_A : AA \to A$ is denoted by a trivalent vertex, and the unit operator $i_A : 1_{\mathcal{C}} \to A$ is denoted by a univalent vertex, and $m_A$ and $i_A$ satisfy the identities below. In the remainder of this subsection, unlabelled black strings refer to the object $A$.

(associative)    (unital).

We denote the adjoints of $m_A$ and $i_A$ by their vertical reflections. By composing $m_A$ with members of an orthonormal basis for $\bigoplus_{x \in \mathrm{Irr}(\mathcal{C})} \mathcal{C}(x, A)$ (or $\bigoplus_{x \in \mathrm{Irr}(\mathcal{C})} \mathcal{C}(A, x)$) on each strand, one can see that the choice of $m_A$ is equivalent to the choice of vertex lifting coefficients [51] for the condensate $A$.

An algebra $(A, m_A, i_A) \in \mathcal{C}$ is called *condensable* if it is also:

- *commutative*: $\quad = m_A \circ \beta_{A,A} = m_A = \quad$

- *unitarily separable*: $m_A^{\dagger}$ is an $A-A$ bimodule map and $m_A \circ m_A^{\dagger} = \mathrm{id}_A$.

$\quad = \quad = \quad \qquad \qquad \quad = \quad$

Typically, we also assume $A$ is *connected*, i.e., $\dim(\mathcal{C}(1_{\mathcal{C}} \to A)) = 1$. In this case, by [110, Rem. 5.6.3], we get standard duality pairings for $A$ by

$$\mathrm{ev}_A := \quad \qquad \qquad \mathrm{coev}_A := \quad .$$

Explicitly, as an important consequence of unitary separability of a condensable algebra $A$, the multiplication map $m_A$ determines an orthogonal projection $m_A^{\dagger} m_A$, selecting specific fusion channels between the objects in the direct sum $A$. Likewise, the unit map selects the unique vacuum channel. The physical interpretation of all these conditions is that, when the images of these projections are energetically favored, strings labelled by $A$ behave like the vacuum string [51], so that any 2D network of $A$-strands only depends on the connectivity, and not the genus. Thus, we may replace a single $A$-strand by an 'A-strand mesh' which behaves like a 2D foam/defect [35, 36, 60, 85], making $A$ the new vacuum in this 2D region.

Given a condensable algebra $A \in \mathcal{C}$, we would like to define the UMTC describing excitations in a (2+1)D bulk region where $A$ is condensed. However, it is more convenient to first define the UFC $\mathcal{C}_A$, which describes wall excitations at the boundary between a region with $\mathcal{C}$ bulk region, and a region where $A$ has been condensed. This $\mathcal{C}_A$ is the category of right $A$-modules $M_A = (M, r_M)$ in $\mathcal{C}$ [57] (see also [24] and [42, §3.3]).

The category $\mathcal{C}_A$ is a UFC derived from $\mathcal{C}$ with fusion rules consistent with the identification of $A$ with the vacuum. To a first approximation, modules $M_A = (M, r_M)$ are the equivalence classes of anyons which are identified by condensing $A$. More precisely, $M$ is a direct sum of anyons and the right action $r_M : MA \to M$, is a choice of fusion channels between $M$ and the condensate $A$, allowing $M$ to absorb $A$. We denote $M_A$ by a green string and the right action $r_M$ by a green trivalent vertex. The choice of $r_M$ must make $M$ stable under repeated fusion with the condensate, leading to the following associativity and unitality conditions.

$$\underbrace{\begin{array}{c}M\end{array} = \begin{array}{c}M\end{array}}_{\text{associative}} \qquad \underbrace{\begin{array}{c}M \quad M\end{array} = \begin{array}{c}M \quad M\end{array}}_{\text{unital}}.$$

**Remark B.1.** Note that this graphical definition of a category of $A$-modules makes sense much more generally: if $\mathcal{X}$ is a (multi)fusion category, $A$ is an algebra in $\mathcal{X}$, and $\mathcal{M}$ is a right $\mathcal{X}$-module category, the same diagrams define a notion of right $A$-module in $\mathcal{M}$, and hence a category $\mathcal{M}_A$ of such modules. Such a category appears in the definition of Notation 3.7.

Since $A$ is a condensable algebra, the category $\mathcal{C}_A$ of $A$-modules has a tensor product $\otimes_A$, which describes the fusion of two excitations living on the domain wall. The tensor product of two $A$-modules is the image of the projection

$$p_{M,N} := \begin{array}{c}\end{array} \in \text{End}_{\mathcal{C}}(MN),$$

where the orange string denotes $N$, and the crossing is the braiding in $\mathcal{C}$. One can interpret the tensor product on $\mathcal{C}_A$ as being defined by embedding $\mathcal{C}_A \to {}_A\mathcal{C}_A$, where each right $A$-module is equipped with a left action defined via the right action and the half-braiding on $A$. The category ${}_A\mathcal{C}_A$ of $A{-}A$-bimodules has a natural monoidal structure for any algebra $A$ in a fusion category [48, 7.8.25].

The image of the projection $p_{M,N}$ is the largest subobject of $MN$ where the effects of fusing a copy of $A$ into the $M$ strand agrees with the effect of fusing $A$ into the $N$ strand. Thus, fusion channels in $MN$ which are preserved by $p_{M,N}$ are stable under fusion with the condensate, whereas fusion channels which are killed by $p_{M,N}$ are also killed when fusing with the condensate. In particular, $A_A$ is the tensor unit of $\mathcal{C}_A$; when $A$ is condensed, $A$ becomes the new vacuum.

The condensed region has topological order described by $\mathcal{C}_A^{\text{loc}}$, the subcategory of $\mathcal{C}_A$ which consists of wall excitations which braid trivially with the condensate, and hence can be pulled off the wall into the bulk region where $A$ is condensed [85, § 2.4]. The UMTC $\mathcal{C}_A^{\text{loc}}$ consists of the *local* right $A$-modules [42, Def 3.12], which satisfy

$$\bullet \ \textit{local:} \quad \begin{array}{c}\end{array} = r_M \circ \beta_{A,M} \circ \beta_{M,A} = r_M = \begin{array}{c}\end{array}.$$

The braiding on $\mathcal{C}_A^{\text{loc}}$ is inherited from $\mathcal{C}$, and one checks that it is compatible with the tensor product in $\mathcal{C}_A$. It is known that $\mathcal{C}_A^{\text{loc}}$ is again a modular tensor category [83, Thm 4.5] [42,

Cor 3.30]. Since local modules braid trivially with the condensate, they can still move adiabatically about the system where $A$ is condensed; modules that do not satisfy the locality condition braid nontrivially with $A$, and are therefore confined to the domain wall. In the special case where $\mathcal{C}_A^{\mathrm{loc}} \cong \mathsf{Hilb}_{\mathsf{fd}}$, i.e. the condensed region has trivial topological order, we say that $A$ is a *Lagrangian algebra*.

Often, we also consider inclusions $A \subseteq B$ of condensable algebras in $\mathcal{C}$. Restricting the multiplication map $BB \to B$ to $BA$ gives $B$ the structure of an $A$-module, and the commutativity of $B$ implies that this $A$-module is local. Hence, condensable algebras $B$ which contain $A$ as a subalebra are condensable algebras in $\mathcal{C}_A^{\mathrm{loc}}$. Conversely, if $B_A$ is a condensable algebra in $\mathcal{C}_A^{\mathrm{loc}}$, then forgetting the module action makes $B$ a condensable algebra in $\mathcal{C}$, with the unit map $A_A \to B_A$ giving an inclusion of algebras $A \to B$ [42, 3.6].

Categories of modules over an algebra also play an important role in understanding anyon condensation from the perspective of enriched fusion categories, as outline in Example 3.2. Observe that the definition of the tensor product $\otimes_A$ does not use the fact that $\mathcal{C}$ is braided, but only that $A$ can half-braid under objects in $\mathcal{C}$, i.e. that $A \in Z(\mathcal{C})$. Therefore, given a unitary fusion category $\mathcal{X}$ and a condensable algebra $A \in Z(\mathcal{X})$, we can define the unitary fusion category $\mathcal{X}_A$ of right $A$-modules in $\mathcal{X}$. We then have $Z(\mathcal{X}_A) \cong Z(\mathcal{X}_A)^{\mathrm{loc}}$ [42, Thm. 3.20], so that $\mathcal{X}_A$ provides the data for a lattice model realization for both the domain wall and the bulk region where $A$ is condensed. As we outlined in Example 3.2, this generalizes straightforwardly to the case where $\mathcal{X}$ is $\mathcal{A}$-enriched and $A \in Z^{\mathcal{A}}(\mathcal{X})$.

# C $\quad \mathcal{D}(D_n)$

In this appendix, we provide some background on the UMTC $\mathcal{D}(D_n)$, where $D_n$ is the dihedral group with $2n$ elements, with presentation $D_n = \langle r, f \,|\, r^n, f^2, (rf)^2 \rangle$. We begin in § C.1 by establish general facts and notation for $\mathcal{D}(G)$, where $G$ is a finite group. In § C.2, we describe a commuting projector lattice model for $\mathcal{D}(G)$ topological order. Finally, in § C.3, we work out the specific case $G = D_n$ in detail.

## C.1 $\quad \mathcal{D}(G)$ **for a non-Abelian group** $G$

Anyons in $\mathcal{D}(G)$ correspond mathematically to irreducible $G$-graded $G$-representations [16, §3.2] Explicitly, a $G$-graded Hilbert space is a Hilbert space $V$ together with a decomposition $V = \bigoplus_{g \in G} V_g$. The subspace $V_g$ is called the *$g$-graded component* of $V$, and states $v \in V_g$ are said to be *$g$-graded*. Maps between $G$-graded Hilbert spaces must preserve the grading, sending $g$-graded vectors to $g$-graded vectors. A $G$-graded $G$-representation consists of a $G$-graded Hilbert space $V$, together with an action $\phi : G \to \mathrm{End}(V)$ of $G$ by unitary operators, satisfying $\phi_h V_g \cong V_{hgh^{-1}}$.

This algebraic description comes from the fact that $\mathcal{D}(G) \cong Z(\mathsf{Hilb}[G])$, the center of the category of finite-dimensional $G$-graded Hilbert spaces. The tensor product on $\mathsf{Hilb}[G]$ is the usual tensor product of Hilbert spaces, and the grading is defined by

$$(V \otimes W)_g = \bigoplus_h V_h \otimes W_{h^{-1}g}$$

the grading of a pure tensor is the product of the two gradings. A half-braiding amounts to the data of a $G$-representation. Explicitly, if $\mathbb{C}_h$ is the 1-dimensional $h$-graded Hilbert space spanned by $e_h$, then the half-braiding $\mathbb{C}_h \otimes V \to V \otimes \mathbb{C}_h$ is given by

$$e_h \otimes v \mapsto \phi_h(v) \otimes e_h\,.$$

The requirement that the half-braiding preserves the $G$-grading means the representation $\phi$ must conjugate the grading, as described above. Thus, if $V$ and $W$ are objects in $\mathcal{D}(G)$, and $v^g \in V$ and $w^h \in W$ are graded vectors, then the braiding has the form

$$\beta_{V,W} : v^g \otimes w^h \mapsto \phi_g^W(w^h) \otimes v^g . \tag{C.1}$$

An irreducible $G$-graded $G$-representation is determined by a pair $(g, \rho)$ where $g \in G$ and $\rho \in \mathrm{Irr}(\mathrm{Rep}(\mathrm{Stab}_G(g)))$ is an action of $\mathrm{Stab}_G(g)$ on a Hilbert space $V^\rho$. Consequently, we will generally label simple objects in $\mathcal{D}(G)$ by such pairs. (Often, we just speak of the action $\rho$ and suppress the Hilbert space $V^\rho$.) We can define a $G$-graded $G$-representation $W$ by setting $W_g := V^\rho$, which gives an irreducible $G$-graded $\mathrm{Stab}_G(g)$-representation, and inducing to get a $G$-representation. The final Hilbert space $W$ will still have $W_g \cong V^\rho$, $W_{hgh^{-1}} \cong W_g$ for every $h \in G$, and $W_k \cong 0$ if $k$ is not conjugate to $g$. For completeness, we give an explicit description of the construction of the $G$-graded $G$-representation $(g, \rho)$. Suppose $\{v_i\}$ is a basis for $V^\rho$. Then the induced representation will have the basis $\{v_i^h | h \in [g]\}$, where $[g]$ is the conjugacy class of $g$. Choose $R \subseteq G$ so that each $h \in [g]$ satisfies $h = rgr^{-1}$ for precisely one $r \in R$. Then every $h \in G$ can be written as $rs$ for some $r \in R$ and $s \in \mathrm{Stab}_G(g)$. Setting $\widetilde{\rho}_s(v_i^g) := (\rho_h(v_i))^g$ and $\widetilde{\rho}_r(v_i^g) := (v_i^{rgr^{-1}})$ completely determines the representation $\widetilde{\rho}$.

**Example C.1.** For an Abelian group $A$, the theory of $A$-graded $A$-representations is straightforward: an $A$-graded $A$-representation is just a pair $(a, \rho)$ where $a \in A$ and $\rho \in \mathrm{Rep}(A)$, with the entire representation graded by $a$. As a fusion category, $\mathcal{D}(A) \cong \mathrm{Hilb}[A] \boxtimes \mathrm{Rep}(A)$.

We now provide a fully explicit example of a $G$-graded $G$-representation where $G$ is not Abelian. The theory of dihedral groups will be analyzed in much more generality in Appendix C.3 below, but we still use the dihedral group $G = D_3 \cong S_3$ here, because it provides the simplest possible non-Abelian example.

**Example C.2.** Consider

$$G = D_3 = \langle r, f | r^3 = 1, rf, fr^{-1} \rangle .$$

Another way of understanding $G$ is that $G \cong S_3$, with $r = (123)$ and $f = (23)$. The group $D_3$ has a single 2-dimensional representation $\sigma$ given by

$$\sigma : r \mapsto \begin{bmatrix} e^{2\pi i/3} & 0 \\ 0 & e^{-2\pi i/3} \end{bmatrix}, f \mapsto \begin{bmatrix} 0 & 1 \\ 1 & 0 \end{bmatrix}.$$

Here, $\sigma$ acts on the Hilbert space $V \cong \mathbb{C}^2$; a basis of $V$ is $\{e_1 = (1,0)^T, e_2 = (0,1)^T\}$.

There are 3 possible ways to define a $D_3$-grading on $\sigma$. First, we could set $V_1 := V$ (and $V_g := 0$ for $g \neq 1$). Since $D_3 = \mathrm{Stab}_{D_3}(1)$, this is the representation $(1, \sigma)$.

Second, we could set $V_r := \mathrm{span}\, e_1$ and $V_{r^{-1}} := \mathrm{span}\, e_2$. Since $r^{-1} = frf^{-1}$, the $D_3$-grading is compatible with the $D_3$-action. Since $\mathrm{Stab}_{D_3}(r) = \langle r \rangle$, this is the representation $(r, e^{2\pi i/3})$. Here, we denote a representation of the cyclic group $\langle r \rangle$ by the eigenvalue of $r$, which uniquely determines the action of $\langle r \rangle$ on the $r$-graded component, since all irreducible representations of Abelian groups are 1-dimensional. The formulae for the induced representation described above then force $\widetilde{e^{2\pi i/3}} \cong \sigma$. This representation could also be called $(r^{-1}, e^{-2\pi i/3})$. In general, whenever $g \notin Z(G)$, we should expect $(g, \rho)$ to have multiple equivalent labels in this way. Finally, we could set $V_{r^{-1}} := \mathrm{span}\, e_1$ and $V_r := \mathrm{span}\, e_2$. This is the representation $(r, e^{-2\pi i/3}) \cong (r^{-1}, e^{2\pi i/3})$.

We also make some preliminary observations about anyons which must braid trivially with each other. If $g \in Z(G)$, then $(g, 1)$ is a pure flux excitation. Since $g \in \mathrm{Stab}_G(h)$ for every $h$, the anyons $(g, 1)$ and $(k, 1)$ braid trivially for every $k$. Anyons of the form $(1, \rho)$ are pure

charge excitations, which braid trivially since $\rho(1) = 1$ for every $\rho$. Other anyons, including $(g,1)$ for $g \notin Z(G)$, are dyonic, and determining whether they braid trivially involves actually computing the braiding (C.1). In general, the double braiding between representations $(g,\lambda)$ and $(h,\rho)$ is

$$v^k \otimes w^\ell \mapsto \widetilde{\lambda}_{k\ell k^{-1}}(v^k) \otimes \widetilde{\rho}_k(w^\ell),$$

where $\widetilde{\cdot}$ refers to the induced representation, $k \in [g]$, and $\ell \in [h]$. In particular $(g,\lambda)$ and $(1,\rho)$ commute precisely when $\rho_g = \mathrm{id}$, since the double braiding simplifies to

$$v^g \otimes w^1 \mapsto \rho_g(v^g) \otimes w^1.$$

We now briefly describe some condensable algebras over $\mathcal{D}(G)$, along with the categories of modules; full details, as well as a description of all condensable algebras in $\mathcal{D}(G,\omega)$, appear in [37]. We make use of the modified quantum double construction [18].

**Definition C.3.** If $L$ and $K$ are finite groups, and $L$ is equipped with a $K$-action $\phi : K \to \mathrm{End}(L)$, then the **modified quantum double** $\mathcal{D}(L,K)$ is the category of $L$-graded $K$-representations, where the action of $k \in K$ takes $\ell$-graded vectors to $(\phi(k)(\ell))$-graded vectors.

**Example C.4.** If $G$ is a finite group, $H$ is a normal subgroup of $G$, and $K$ is any subgroup of $G$, then we have the fusion category $\mathcal{D}(G/H,K)$, where $K$ acts on $G/H$ by conjugacy.

In particular, the usual quantum double $\mathcal{D}(G)$ is a modified quantum double, namely $\mathcal{D}(G,G)$. The category $\mathcal{D}(G/H,K)$ then becomes $\mathcal{D}(G)$ module tensor category, with the action simply applying the quotient map $G \to G/H$ to the grading and restricting the representation from $G$ to $K$.

If $H \lhd G$ is a normal subgroup, then the group algebra $A := \mathbb{C}[H]$ is a condensable algebra in $\mathcal{D}(G)$. This algebra is spanned by $H$, and the generator $h$ is an $h$-graded vector; as an object in $\mathcal{D}(G)$, we have $A \cong \bigoplus_{r \in R}(r,1)$, where $R$ is a set containing one representative of each conjugacy class in $G$ which is contained in $H$. We have $\mathcal{D}(G)_A \cong \mathcal{D}(G/H,G)$, and $\mathcal{D}(G)_A^{\mathrm{loc}} \cong \mathcal{D}(G/H)$.

If $K \subseteq G$ is any subgroup, then the algebra $B := \mathbb{C}^{G/K}$ of functions on $G/K$ (which need not be a group, but carries a left action of $G$) is a condensable algebra in $\mathcal{D}(G)$. This algebra is a $G$-representation, and so can be viewed as an object of $\mathcal{D}(G)$ which is entirely graded by 1. We have $\mathcal{D}(G)_B \cong \mathcal{D}(G,K)$, and $\mathcal{D}(G)_B^{\mathrm{loc}} \cong \mathcal{D}(K)$.

## C.2 String-net realization

We now describe a string-net model adapted to the special case of $\mathcal{D}(G)$. We begin with a model dual to Kitaev's quantum double model [77], built on a square lattice. This model can also be viewed as a variant of [103], taking advantage of the fact that $\mathrm{Hilb}[G]$ is multiplicity free. The special case of $\mathcal{D}(S_3) \cong \mathcal{D}(D_3)$, appears explicitly in [114, § 5.4].

To produce the $\mathcal{D}(G)$ bulk, we assign to each link the Hilbert space $\mathbb{C}^G$. An orthonormal basis of this Hilbert space is $\{|g\rangle : g \in G\}$, i.e. $\langle g|h\rangle = \delta_{g,h}$. We define the operators $\lambda_g$ and $\rho_g$ by $\lambda_g|h\rangle = |gh\rangle$ and $\rho_g|h\rangle = |hg\rangle$. Notice that $\lambda_g^\dagger = \lambda_{g^{-1}}$ and $\rho_g^\dagger = \rho_{g^{-1}}$; also, for every $g$ and $h$, we have $[\lambda_g, \rho_h] = 0$.

For each plaquette $p$, we define the operator

$$B_p = \frac{1}{|G|} \sum_{g \in G} \rho_g \boxed{\begin{matrix} & \rho_g & \\ & p & \\ & \lambda_g^\dagger & \end{matrix}} \lambda_g^\dagger \ .$$

For each vertex $v$, we define

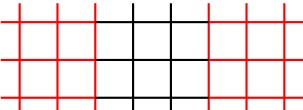

$$A_v \left| a \xrightarrow{d^\dagger}_{\substack{v \\ b}} c^\dagger \right\rangle = -1^{1-\chi_1(abc^{-1}d^{-1})} \left| a \xrightarrow{d^\dagger}_{\substack{v \\ b}} c^\dagger \right\rangle,$$

where $\chi_1(1) = 1$ and $\chi_1(x) = 0$ for $x \neq 1$. The Hamiltonian

$$H = -\sum_v A_v - \sum_p B_p,$$

is then a sum of commuting projectors.

We can describe a model for a 'rough' boundary, where $A = \mathbb{C}[H] \in \mathcal{D}(G)$ is condensed, by replacing a strip of black links with red ones, where each red link carries a Hilbert space spanned by $G/H = \{gH : g \in G\}$.

We redefine the operators $A_v$ and $B_p$ at vertices and plaquettes with red links as follows.

$$A_v \left| a \xrightarrow{dH}_{\substack{v \\ bH}} cH \right\rangle = (-1)^{1-\chi_H(abc^{-1}d^{-1}H)} \left| a \xrightarrow{dH}_{\substack{v \\ bH}} cH \right\rangle,$$

$$A_v \left| aH \xrightarrow{dH}_{\substack{v \\ bH}} cH \right\rangle = (-1)^{1-\chi_H(abc^{-1}d^{-1}H)} \left| aH \xrightarrow{dH}_{\substack{v \\ bH}} cH \right\rangle,$$

$$B_p = \frac{1}{|G|} \sum_{g \in G} \rho_g \boxed{\,p\,} \lambda^\dagger_{gH}, \quad \text{or} \quad \frac{1}{|G|} \sum_{g \in G} \rho_{gH} \boxed{\,p\,} \lambda^\dagger_{gH}.$$

Again, $\chi_H(H) = 1$ and $\chi_H(xH) \neq 1$ if $xH$ is not the identity coset $H$.

For $h \in H$, we have $\chi_H(hH) = \chi_H(H) = 1$. Consequently, anyons in $\mathcal{D}(G)$ of the form $(h, \rho)$ for $h \in H$ will not excite $A_v$ terms at vertices $v$ incident to a red link, explaining why $A$ becomes condensed.

We can also create a horizontal 'smooth' boundary, where $B = \mathbb{C}^{G/K}$ is condensed, by replacing a strip of black links with blue ones, where each blue link carries a Hilbert space spanned by $K$.

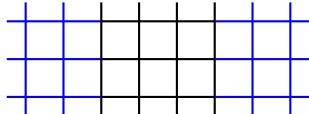

We then redefine the operators $A_v$ and $B_p$ at vertices and plaquettes with blue links as follows.

$$A_v \left| a \xrightarrow{d}_{\substack{v \\ b}} k \right\rangle = (-1)^{1-\chi_1(abk^{-1}d^{-1})} \left| a \xrightarrow{d}_{\substack{v \\ b}} k \right\rangle,$$

$$A_v \left| i \xrightarrow{\ell}_{\substack{v \\ j}} k \right\rangle = (-1)^{1-\chi_1(ijk^{-1}\ell^{-1})} \left| i \xrightarrow{\ell}_{\substack{v \\ j}} k \right\rangle,$$

$$B_p = \frac{1}{|K|} \sum_{k \in K} \rho_k \boxed{p} \lambda_k^\dagger \Big|_{\lambda_k^\dagger} , \quad \text{or} \quad \frac{1}{|K|} \sum_{k \in K} \rho_k \boxed{p} \lambda_k^\dagger \Big|_{\lambda_k^\dagger} .$$

## C.3 Dihedral groups

We begin with a brief discussion of the UMTC $\mathcal{D}(D_n)$, the quantum double of the dihedral group, where $n$ is odd. The dihedral group $D_n$ has $2n$ elements and is given by the presentation

$$D_n = \langle r, f \,|\, r^n = f^2 = 1, rf = fr^{-1} \rangle .$$

If $n$ is even, we have $D_n \cong D_{n/2} \times \mathbb{Z}/2$, and hence $\mathcal{D}(D_n) \cong \mathcal{D}(D_{n/2}) \boxtimes \mathcal{D}(\mathbb{Z}/2)$. We are interested in behavior specific to non-Abelian topological orders, so we always choose $n$ to be odd in order to avoid the unnecessary complication of keeping track of one or more extra layers of toric code.

The one dimensional irreducible representations of $D_n$ are the trivial representation 1 and the sign representation $\epsilon : r \mapsto 1, f \mapsto -1$. The 2-dimensional irreducible representations of $D_n$ are given by

$$\sigma_j : r \mapsto \begin{bmatrix} \omega^j & 0 \\ 0 & \omega^{-j} \end{bmatrix}, \quad f \mapsto \begin{bmatrix} 0 & 1 \\ 1 & 0 \end{bmatrix},$$

where $\omega = e^{2\pi i/n}$ and $j$ ranges from 1 to $\lfloor \frac{n}{2} \rfloor$. We sometimes also mention the reducible representation $\sigma_0 \cong 1 \oplus \epsilon$, or irreducible representations $\sigma_{-j} \cong \sigma_{n-j} \cong \sigma_j$.

The conjugacy classes in $D_n$ are $[1] = \{1\}$, $[r^k] = \{r^k, r^{-k}\}$ for each $k \neq 0$, and $[f] = \{r^k f \,|\, 0 \leq k < n\}$. The stabilizer of $r^k$ is $\langle r \rangle \cong \mathbb{Z}/n$, with irreducible representations determined by an eigenvalue $\omega^j$ of $r$, while the stabilizer of $r^k f$ is $\langle r^k f \rangle \cong \mathbb{Z}/2$, with two possible irreducible representations $\pm 1$. We have

$$\mathrm{Ind}_{\langle r \rangle}^{D_n}(\omega^j) \cong \sigma_j ,$$

$$\mathrm{Ind}_{\langle r^k f \rangle}^{D_n}(1) \cong 1 \oplus \left( \bigoplus_{k=1}^{\lfloor \frac{n}{2} \rfloor} \sigma_k \right) ,$$

$$\mathrm{Ind}_{\langle r^k f \rangle}^{D_n}(-1) \cong \epsilon \oplus \left( \bigoplus_{k=1}^{\lfloor \frac{n}{2} \rfloor} \sigma_k \right) .$$

The fusion rules of $\mathcal{D}(D_n)$ are abelian, and are summarized in the following table.

| $X$ | $Y$ | $X \otimes Y$ |
|---|---|---|
| $(1, \epsilon)$ | $(1, \epsilon)$ | $(1, 1)$ |
| $(1, \epsilon)$ | $(1, \sigma_k)$ | $(1, \sigma_k)$ |
| $(1, \sigma_k)$ | $(1, \sigma_j)$ | $(1, \sigma_{k+j}) \oplus (1, \sigma_{k-j})$ |
| $(r^a, \omega^k)$ | $(r^b, \omega^j)$ | $(r^{a+b}, \omega^{j+k}) \oplus (r^{a-b}, \omega^{j-k})$ |
| $(1, \epsilon)$ | $(f, 1)$ | $(f, -1)$ |
| $(r^a, \omega^k) k \neq 0$ | $(f, 1)$ | $(f, 1) \oplus (f, -1)$ |
| $(f, 1)$ | $(f, 1)$ | $(1, 1) \oplus \left( \bigoplus_{a=1}^{\lfloor \frac{n}{2} \rfloor} \bigoplus_{k=0}^{n-1} (r^a, \omega^k) \right)$ |

We note the isomorphisms $(r^a, \omega^k) \cong (r^{-a}, \omega^{-k})$ and define by convention $(r^0, \omega^k) := (1, \sigma_k)$. Most entries of the table involving $(f, -1)$ are omitted, but since $(f, -1) \cong (f, 1) \otimes (1, \epsilon)$ and $(1, \epsilon)$ is an Abelian anyon, they can be easily computed.

The fusion graph for $\mathcal{D}(D_n)$ with respect to $(f, 1)$ is given by

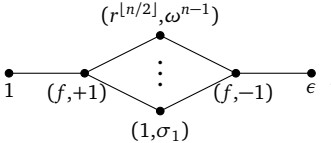

where the middle vertices range over all remaining simple objects.

Observe that $D_n$ has the index 2 normal subgroup $\mathbb{Z}/n\mathbb{Z} = \langle r \rangle$, and thus $\mathsf{Hilb}(D_n)$ is a $\mathbb{Z}/2\mathbb{Z} = \langle f \rangle$-graded extension of $\mathsf{Hilb}(\mathbb{Z}/n\mathbb{Z})$. By [61], the *relative center* $Z_{\mathsf{Hilb}(\mathbb{Z}/n\mathbb{Z})}(\mathsf{Hilb}(D_n))$ is a $\mathbb{Z}/2\mathbb{Z}$-crossed braided extension of $\mathcal{D}(\mathbb{Z}/n\mathbb{Z})$, and the $\mathbb{Z}/2\mathbb{Z}$-equivariantization of this relative center is $\mathcal{D}(D_n)$. Physically, this tells us that anyons with a flux other than $[f]$ can be thought of as direct sums of the abelian anyons in $\mathbb{Z}/n\mathbb{Z}$-toric code, resulting from the inclusion of loops labelled $f$ in the ground state. If we condense the algebra $(1,1) \oplus (1, \epsilon) \cong \mathbb{C}[\mathbb{Z}/2]$, confining $f$, we end up with $\mathcal{D}(\mathbb{Z}/n)$ topological order, and the 2-dimensional anyons in $\mathcal{D}(D_n)$ all split. Explicitly, $(1, \sigma_k) \cong m^k \oplus m^{-k}$, $(r^a, 1) \cong e^a \oplus e^{-a}$, and in general, $(r^a, \omega^k) \cong e^a m^k \oplus e^{-a} m^{-k}$. Since $f r f^{-1} = r^{-1}$, lines labelled by $f$ correspond to the automorphism $e \mapsto e^{-1}$, $m \mapsto m^{-1}$ of $\mathbb{Z}/n$-toric code.

Thus, $\mathcal{D}(D_n)$ is an $\langle f \rangle$-graded tensor category, with $\mathcal{D}(D_n)_1 \cong \mathcal{D}(\mathbb{Z}/n)^{\langle f \rangle}$ as the trivial graded component, and $(f, \pm 1)$ as the simple objects in the $f$-graded component. As a $\mathcal{D}(D_n)_1$ module, the component $\mathcal{D}(D_n)_f$ is the category of modules over the algebra $\mathbb{C}[\langle r \rangle] \otimes \mathbb{C}^{D_n/\langle f \rangle}$. Note that $\mathbb{C}^{D_n/\langle f \rangle} \cong 1 \oplus \left( \bigoplus_{k=1}^{\lfloor n/2 \rfloor} \sigma_k \right)$ is isomorphic to the standard representation of $D_n$, where $D_n$ acts on (a vector space spanned by) the vertices of a regular $n$-gon.

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
