# Peer review of "Composing topological domain walls and anyon mobility"

_SciPost Physics, doi:SciPost Phys. 15, 076 (2023)_

## Round 1 · Author Response

We would like to thank the referees for their careful reading of the manuscript and their suggestions to improve the paper. We have addressed all comments of the referees. Please find our responses below in the list of changes below.

---

## Round 1 · List of Changes

Warnings issued while processing user-supplied markup:

  • Inconsistency: Markdown and reStructuredText syntaxes are mixed. Markdown will be used.
    Add "#coerce:reST" or "#coerce:plain" as the first line of your text to force reStructuredText or no markup.
    You may also contact the helpdesk if the formatting is incorrect and you are unable to edit your text.

Referee 1:

In some aspects, the presentation of the work does not make the intended audience clear. Some concepts are explained extensively, including an appendix on algebra objects, but other seem to be deemed obvious. I feel that a subset of the audience, particularly in a physics journal, may benefit from some clearer distinction/direction as to whether some parts are intended for the more mathematical audience.

We have included more sign posting to indicate when certain parts are intended for a more mathematical audience, together with introducing the environment "Remark (Mathematical)" to explicitly signify this for certain remarks.

Remark III.13 seems to be functioning to expand on Remark II.4, however this is not made clear by the authors. Both of these deal with categorical intricacies which may be beyond many of the physically inclined readers. As such, I recommend these sections be reviewed and clarified.

We have added some language to clarify the connection between these two remarks. In both cases, we have created a new environment called Remark (Mathematical) to emphasize that these remarks are intended for a more mathematical audience. This will serve a sign post to more physically inclined readers that they may skip these remarks. We have included text in the introduction about this new notation and our intentions for our readers.

A couple of times, notation/concepts are used extensively before being introduced/defined. Some particular examples include: 'free module functor', used on page 16, but defined on page 18. In example VD (pages 46-47), this happens extensively.

We have added Footnote 21 introducing the free module functor where it first appears, along with a reference to [BN11]. Section VD now begins with explicit sign posting to Appendix C, where this new notation is introduced and defined. We have also introduced the notation for the group algebra and the function algebra on the group here to clarify any confusion.

Requested changes 1. The explanation of the stacking Witt equivalences on page 12-13 is quite confusing. A figure should be added to clarify where everything sits.

We have now added Figure 6 to help with clarity.

  1. An in text definition of a Witt equivalence should be given. This is central to the work, but only appears in a figure caption.

    It is already defined in the text in the first paragraph of IIC, where "Witt equivalence" appears in italics.

  2. I recommend the authors attempt to re-order/clarify some sections as mentioned above.

    We have attempted to address this by explicitly flagging those remarks that focus on category theory as “Remark (Mathematical)." We have also tried to make sure we define our notation before using it, and we have increased sign posting to appendices when appropriate.

  3. Due to the length of the SU(2)_4 example, I think it would benefit from explicitly reminding the reader at the end how the various categories relate to the notation A, X, and Z^A(X) used throughout.

    We have reminded the reader at the end of IIB about which categories correspond to A, Z(X), and Z^A(X).

  4. Page 46, column 2, part (2). This section seems somewhat scrambled. The notation is used, the reintroduced after the equations. Some of the notation has already been used in the first column. This should be reviewed.

    We have included a paragraph at the beginning of Section VD making explicit reference to Appendix C, and we have introduced the notation for the group algebra and function algebra there to make things less confusing. We also have deleted repeated text in (2), which hopefully clears things up.

Very minor/Typos:

We have corrected all the minor typos pointed out below. We reply about a particular one below.

Typo page 1, column 1: 'vaccuum' Typo page 2, column 2: 'bimdoule' Typo page 12, column 2: Should 'K_B' be 'K_{\overline{B}}'?

Since B\boxtimes \overline{B} is its own reverse, K_B is the same as K_{\overline{B}}. We have added a comment to this effect.

Typo page 21, column 1: Same as above for K_{Z(Y)} Typo page 21, column 2.5: End_{X−Y}(M)should be A enriched Typo page 39, column 2.75: This seems to be a definition of C Typo page 40, column 1.99: '[L_e,em^2]' Typo page 40, column 2.75: 'geometrythis' Typo page 42, column 2: 'T_{m→m}' Typo page 45, column 1.5: 'is contains' Typo page 45, column 2: missing 1/2 factors Typo page 46, column 1: \rho \in Irr(Rep(Stab_{D_n}(g))) should be D_k

=====================================================

Referee 2:

  1. The notation “2D bulk” in Fig. 1 is confusing. This “2D” must be space dimension, right? but 2+1D is clearly a spacetime dimension. Maybe a declaration of the convention is needed.

    We have now included our conventions for this figure in its caption below.

  2. Page 2, the mathematical term “Morita 4-category UBFC of unitary braided fusion categories” appears suddenly. I think that it is better to explain its definition. Otherwise, I do not see why “objects in individual UMTCs, do not appear as higher morphisms in the 4-category UBFC.”

    We include a more detailed explanation of UBFC here along with an explanation of why the anyons do not appear at the correct categorical level in UBFC to describe topological order.

  3. Page 2, the notion of a module tensor category over a braided tensor category also appeared in Definition 2.6.1 in [KZ18a].

    Here we are only citing the original reference that we are aware of, although we acknowledge the same notion has appeared in many interesting articles since.

  4. Page 7, what does the term “\Omega tensor” mean? Is it the \Omega in Eq. (3)? If so, it is better to refer to (3) here.

    The \Omega-tensor is the 6j description of the half-braiding as described in Equations (42) and (43) of [LLB21]. We have changed the sentence by including "the \Omega-tensor (3) giving the 6j-description" to make this more clear.

  5. Page 8, is C in Remark II.5 a typo?

    Fixed!

  6. Page 11, “also that Witt equivalent UMTCs can appear as surface topological orders of the same invertible bulk.” I have a question here. Is that true that Witt inequivalent UMTCs can not realized as surface topological orders the same invertible bulk? If so, the invertible 3+1d TQFT are classified by the Witt group of UMTC, right? Two surface topological orders associated to two Witt inequivalent UMTCs can be connected by 1+1D gapless domain wall. Does this gapless wall produces a 2+1D ‘relative bulk’ in 3+1D?

    This is a very important and interesting question that we specifically wanted to avoid in this article, as it is beyond the scope of the present work. We were careful to only focus on topological domain walls and commuting projector local Hamiltonians.

  7. Page 12. is the “UBFC” in Definition II.7 a typo for UMTC? Appendix B only discuss UMTC.

    Yes, since we are claiming K_A is Lagrangian. Fixed!

  8. Page 13, the “M” at the bottom of the left column should be a typo.

    Fixed!

  9. Page 14, A in the footnote is a typo.

    Fixed!

  10. Page 15, “we now turn to the question of central interest: namely, what can happen when we compose two or more domain walls by stacking them? This question is very important, since any domain wall can be obtained by such compositions.” The reason provided here does not sound very convincing. In addition to the reasons from condensed matter physics, maybe the authors can also add that computing the “fusion rules” of topological defects in various dimensions is becoming a fashion especially after the new wave of studying the non-invertible symmetries from the high energy community.

    We have added to the wording here to explicitly discuss non-invertible symmetries, along with including many new references.

  11. Page 20, I am not sure if the notation (M \boxtimes N)_{S_Y} has been explained anywhere in the paper. Is it the category of left or right S_y-modules in (M \boxtimes N)? If so, this notion is not so obvious to physics oriented journal. Maybe an explanation is needed.

    We now include Remark B1 in the appendix defining this, and we include a forward reference to this remark.

  12. Page 20, “In this alternative definition, a y string emanating from the left (X,Y) boundary cannot end in the bulk, and must cross to the right (Y, Z) boundary.” This sentence is hard to visualize without a picture.

    We have revised our explanation of the Deligne product in this context and also included a picture.

  13. Page 22, Example III.12 and more examples can be found in Table 1 in arXiv:2205.05565.

    We have included a citation to this reference.

  14. Page 41-42, some mathematical details in the discussion of double Ising might have appeared in arXiv:1903.12334.

    We have added a citation to this reference, as well as an additional new reference.

---

## Editorial Decision

published